# Enhancing Instance-Level Image Classification with Set-Level Labels

**Renyu Zhang**
Department of Computer Science
University of Chicago
zhangr@uchicago.edu

**Aly A. Khan**
Department of Pathology and Family Medicine
University of Chicago
aakhan@uchicago.edu

**Yuxin Chen**
Department of Computer Science
University of Chicago
chenyuxin@uchicago.edu

**Robert L. Grossman**
Department of Computer Science and Medicine
University of Chicago
rgrossman1@uchicago.edu

## Abstract

Instance-level image classification tasks, e.g., few-shot learning and transfer learning, have traditionally relied on single-instance labels to train models. However, set-level coarse-grained labels that capture relationships among instances can also provide rich information in real-world scenarios. In this paper, we present a novel approach to enhance instance-level image classification by leveraging set-level labels. We provide a theoretical analysis of the proposed method, including recognizing conditions for fast excess risk rate, shedding light on the theoretical foundations of our approach. We conducted experiments on two distinct categories of datasets: natural image datasets and histopathology image datasets. Our experimental results demonstrate the effectiveness of our approach, showcasing improved classification performance compared to traditional single-instance label-based methods. Notably, our algorithm achieves 13% improvement in classification accuracy compared to the strongest baseline on the histopathology image classification benchmarks. Importantly, our experimental findings align with the theoretical analysis, reinforcing the robustness and reliability of our proposed method. This work bridges the gap between instance-level and set-level image classification, offering a promising avenue for advancing the capabilities of image classification models with set-level coarse-grained labels.

## 1 Introduction

A large amount of labeled data is typically required in traditional machine learning approaches, e.g., few-shot learning (FSL) and transfer learning (TL), to learn a robust model. However, procuring sufficient labeled data for each task is often challenging or infeasible in real-world scenarios. In this paper, we consider a novel problem setting where, similar to FSL, we have a limited number of fine-grained labels in the target domain. In the source domain, though, we have a large amount of coarse-grained set-level labels, which are easier to obtain and relevant to fine-grained labels. For example, in a digital library, there are coarse-grained set-level labels indicating the general content of photo albums, such as "beach vacation", "nature landscapes", or "picnic". However, within each of these albums, there are numerous individual images, each with its own unique details and characteristics that are not explicitly labeled. In the downstream task, for instance, we care about the object classification such as "tree", "beach", or "mountain". Similarly, in the medical domain, it is often useful to predict fine-grained labels of tissues, while only set-level annotations of histopathology slides are available for training at scale. We seek to enhance the downstream classification tasks with the coarse-grained set-level labels.

An effective approach to addressing the overreliance on abundant training data is FSL—a paradigm that has gained significant attention in recent years (Vinyals et al., 2016; Wang & Hebert, 2016; Triantafillou et al., 2017; Finn et al., 2017; Snell et al., 2017; Sung et al., 2018; Wang et al., 2018; Oreshkin et al., 2018; Rusu et al., 2018; Ye et al., 2018; Lee et al., 2019b; Li et al., 2019). FSL

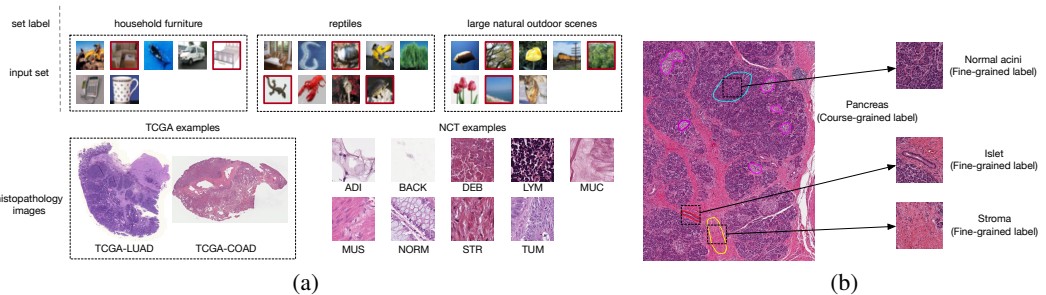

Figure 1: (a) A collection of image sets sampled from CIFAR-100 are in the upper row. The coarse-grained label of a set is the most frequent superclass of images inside the set. WSI examples from TCGA and patches from NCT dataset are in the lower row. (b) Hierarchy of coarse- and fine-grained labels for histopathology images.

pretrains a model that can quickly adapt to new tasks using only a few labeled examples. Recent studies (Chen et al., 2019; Tian et al., 2020; Shakeri et al., 2022; Yang et al., 2022) have shown that pretraining, coupled with fine-tuning on a new task, outperforms more sophisticated episodic training methods. This involves initially training a base model on a diverse set of tasks using abundant labeled data from a source domain, and subsequently fine-tuning the model using only a small number of labeled examples specific to the target task. Despite their promising performance, existing FSL models typically depend on finely labeled source data for predicting fine-grained labels.

As an illustrative example, we consider histopathology image classification where acquiring a substantial number of fine-grained labels for individual patches (e.g., tissue labels shown in the lower row of figure 1a) is challenging. Conversely, a wealth of coarse-grained labels (e.g. the site of origin of the tumors associated with whole slide images (WSIs) from TCGA shown on the left-hand side of the upper row of figure 1a) are easily available. This motivates us to leverage these abundant coarse-grained labels and hierarchical relationships, such as between organs and tissues (as depicted in figure 1b), to enhance representation learning. Tissues consist of cellular assemblies with shared functionalities, while organs are comprised of multiple tissues. This hierarchical relationship serves as a conceptual foundation for our representation learning and provides significant contextual information for facilitating representation learning. By using coarse-grained information within this hierarchy, our goal is to learn efficiently fine-grained tissue representations within WSIs. Another example is shown in the upper row of figure 1a. We emulate a programmatic labeler that use heuristics such as keywords, regular expression, or knowledge bases to solicit sets of images. The coarse-grained labels, e.g., the most frequent superclass of images in the set, can be used to facilitate representation learning for downstream tasks such as instance-level image classification.

**Our contribution** In this paper, we introduce Fine-grAined representation learning from Coarse-graIned LabEls (FACILE), a novel generic representation learning framework that uses easily accessible coarse-grained annotations to quickly adapt to new fine-grained tasks. Distinct from existing practices in FSL and TL, our approach utilizes coarse-grained labels in the source domain. This sets our methodology apart from conventional FSL and TL techniques, which typically rely on meticulously labeled source data to train models.

We provide an initial theoretical analysis to motivate the empirical success of FACILE and examine the convergence rate for the excess risk of downstream tasks under a novel Lipschitzness condition on the loss function concerning the fine-grained labels. Our study reveals that the availability of coarse-grained labels can lead to a substantial acceleration in the excess risk rate for fine-grained label prediction tasks, achieving a fast rate of $\mathcal{O}(1/n)$, where $n$ represents the number of fine-grained data points. This analysis highlights the significant potential for leveraging coarse-grained labels to enhance the learning process in fine-grained label prediction tasks.

In our experiments, we thoroughly investigate the effectiveness of FACILE through a series of extensive experiments on natural image datasets and histopathology image datasets. For natural image datasets, we sample input sets from training data from CIFAR-100 and use the unique superclass number and most frequent superclass as coarse-grained labels. The generated datasets are used to evaluate different models. We also evaluate models by fine-tuning the fully connected layer ap-

pended to ViT-B/16 (Dosovitskiy et al., 2020) of CLIP (Radford et al., 2021) in an anomaly detection dataset based on CUB200 (He & Peng, 2019). For histopathology applications, we leverage two large datasets with coarse-grained labels to pretrain our models. Subsequently, we evaluate the performance of these trained models on a diverse collection of histopathology datasets. Our algorithm achieves strong performance on 4 downstream datasets. Notably, when tested on LC25000 (Borkowski et al., 2021), our model achieves roughly 90% average ACC with 1,000 randomly sampled tasks which only have 5 fine-grained labeled data points for each of the 5 classes, outperforms the strongest baseline by roughly 13% with logistic regression fine-grained classifier. We further evaluate various models by fine-tuning the fully connected layer appended to ViT-B/14 (Dosovitskiy et al., 2020) of DINO V2 (Oquab et al., 2023). These models can leverage the capability of "foundation" models and enhance the model performance on target tasks. Our experiments provide compelling evidence of the efficacy and generalizability of FACILE across various datasets, highlighting its potential as a robust representation learning framework.

## 2    FINE-GRAINED REPRESENTATION LEARNING FROM COARSE-GRAINED LABELS

**Notation**   Our model pre-trains on a collection of samples, denoted by $\{(s_i, w_i)\}_{i=1}^m$. Each $s_i$ is an input set of instances $\{x_j\}_{j=1}^a$, where $a$ represents the variable input set size. $\{w_i\}$ are the set-level coarse-grained labels. The space of all instances is $\mathcal{X}$ and the space of all instance labels, which we call fine-grained labels, is $\mathcal{Y}$. The space of pre-training data is $\mathcal{S} \times \mathcal{W}$, where $\mathcal{S} = \{\{x_1, \ldots, x_a\} : x_j \in \mathcal{X} \text{ for } \forall j \in [a]\}$ and $\mathcal{W}$ denotes the space of coarse-grained labels. We receive $(X, Y)$ from product space $\mathcal{X} \times \mathcal{Y}$ and corresponding $(S, W)$ from product space $\mathcal{S} \times \mathcal{W}$. The goal is to predict the fine-grained labels $y \in \mathcal{Y}$ from the instance features $x \in \mathcal{X}$.

### 2.1   THE FACILE ALGORITHM

We study the model in a FSL setting where we have three datasets: (1) pre-training coarse-grained datasets $\mathcal{D}_m^{\text{cg}} = \{(s_i, w_i)\}_{i=1}^m$ sampled i.i.d. from $P_{S,W}$ (2) fine-grained support dataset $\mathcal{D}_n^{\text{fg}} = \{(x_i, y_i)\}_{i=1}^n$ sampled i.i.d., from $P_{X,Y}$, and (3) query set $\mathcal{D}^{\text{query}}$. The support set $\mathcal{D}_n^{\text{fg}}$ contains $c$ classes and $k$ samples $x$ in each class (i.e., $n \equiv kc$). We assume a latent space $\mathcal{Z}$ for embedding $Z$. We define instance feature maps $\mathcal{E} = \{e : \mathcal{X} \to \mathcal{Z}\}$, set-input functions $\mathcal{G} = \{g : \mathcal{M} \to \mathcal{W}\}$ where $\mathcal{M} = \{\{z_1, \ldots, z_a\} : z_j \in \mathcal{Z} \text{ for } j \in [a]\}$, and fine-grained label predictors $\mathcal{F} = \{f : \mathcal{Z} \to \mathcal{Y}\}$. The corresponding set-input feature map of an instance feature map $e$ is defined as $\phi^e : \mathcal{S} \to \mathcal{M}$. We assume the class of $f$ is parameterized and identify $f$ with parameter vectors for theoretical analysis. We then learn feature map $e$, fine-grained label predictor $f$, and predict fine-grained label with $f \circ e$. The schema of our model is illustrated in figure 2.

We assume two loss functions: $\ell^{\text{fg}} : \mathcal{Y} \times \mathcal{Y} \to \mathbb{R}$ for fine-grained label prediction and $\ell^{\text{cg}} : \mathcal{W} \times \mathcal{W} \to \mathbb{R}$ for coarse-grained label prediction. $\ell^{\text{fg}}$ measures the loss of the fine-grained label predictor. We assume this loss is differentiable in its first argument. $\ell^{\text{cg}}$ measures the loss of pre-training with coarse-grained labels. For theoretical analysis, we are interested in two particular cases of $\ell^{\text{cg}}$: i) $\ell^{\text{cg}}(w, w') = \mathbb{1}\{w \neq w'\}$ where $\mathcal{W}$ is a categorical space; and ii) $\ell^{\text{cg}}(w, w') = \|w - w'\|$ (for some norm $\|\cdot\|$ on $\mathcal{W}$) where $\mathcal{W}$ is a continuous space. We can also measure the loss of a feature map $e$ by $\ell_e^{\text{cg}} = \ell^{\text{cg}}(g_e \circ \phi^e(s), w)$, where $g_e \in \arg\min_g \mathbb{E}_{P_{S,W}} \ell^{\text{cg}}(g \circ \phi^e(S), W)$. We assume there is an unknown "good" embedding $M = \phi^{e_0}(S) \in \mathcal{M}$, by which a set-input function $g_{e_0}$ can determine $W$, i.e., $g_{e_0}(M) = g_{e_0} \circ \phi^{e_0}(S) = W$. The strict assumption of equality can be relaxed by incorporating an additive error term into our risk bounds of $g_{e_0} \circ \phi^{e_0}$.

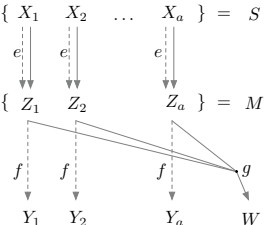

Figure 2: Schema of the FACILE model. The dotted lines represent the flow of fine-grained data, and the solid lines denote the flow of coarse-grained labels

Our primary goal is to learn an instance label predictor or fine-grained label predictor $\hat{f} \circ \hat{e}$ that achieves low risk $\mathbb{E}_{P_{X,Y}}[\ell^{\text{fg}}(\hat{f} \circ \hat{e}(X), Y)]$ and we can bound the excess risk:

$$\mathbb{E}_{P_{X,Y}}[\ell^{\text{fg}}(\hat{f} \circ \hat{e}(X), Y) - \ell^{\text{fg}}(f^* \circ e^*(X), Y)] \tag{1}$$

---

**Algorithm 1** FACILE algorithm

1: **Input:** loss functions $\ell^{\mathrm{fg}}$, $\ell^{\mathrm{cg}}$, predictors $\mathcal{E}$, $\mathcal{G}$, $\mathcal{F}$, datasets $\mathcal{D}_m^{\mathrm{cg}}$ and $\mathcal{D}_n^{\mathrm{fg}}$
2: obtain feature map $\hat{e} \leftarrow \mathcal{A}(\ell^{\mathrm{cg}}, \mathcal{D}_m^{\mathrm{cg}}, \mathcal{E})$
3: create dataset $\mathcal{D}_n^{\mathrm{fg,aug}} = \{(z_i, y_i) : z_i = \hat{e}(x_i), (x_i, y_i) \in \mathcal{D}_n^{\mathrm{fg}}\}_{i=1}^n$
4: obtain fine-grained label predictor $\hat{f} \circ \hat{e}$, where $\hat{f} \leftarrow \mathcal{A}(\ell^{\mathrm{fg}}, \mathcal{D}_n^{\mathrm{fg,aug}}, \mathcal{F})$
5: **Return:** $\hat{f} \circ \hat{e}$

---

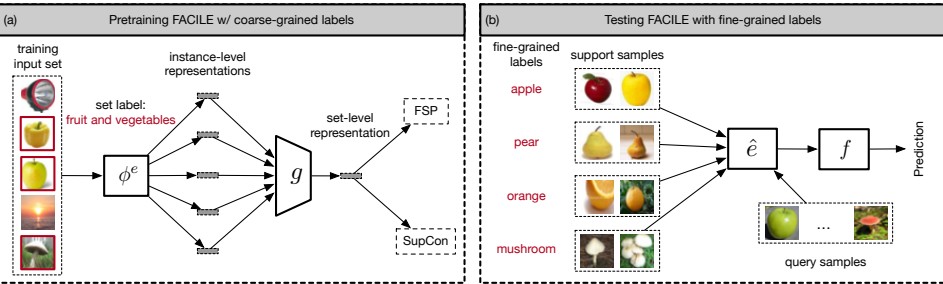

Figure 3: An overview of the FACILE algorithm. (a) Pre-training step of FACILE with coarse-grained labels. The input is a set of images and the target is set-level coarse-grained label. $e$ is an instance feature map and $\phi^e$ is the corresponding set-input feature map. $g$ is the set-input model. We can instantiate the $\mathcal{A}(\ell^{\mathrm{cg}}, \mathcal{D}_m^{\mathrm{cg}}, \mathcal{E})$ with any supervised learning algorithms, e.g., fully supervised pre-training (FSP) with cross-entropy loss and the SupCon model. (b) Fine-grained learning of FACILE with fine-grained labels. The learned instance feature map $\hat{e}$ extracts instance-level features from patches of the support set and query set. $f$ is the fine-grained label predictor.

where $e^* \in \arg\min_{e \in \mathcal{E}} \mathbb{E}_{P_{S,W}} \ell_e^{\mathrm{cg}}(S, W)$ and $f^* \in \arg\min_{f \in \mathcal{F}} \mathbb{E}_{P_{X,Y}}[\ell^{\mathrm{fg}}(f \circ e^*(X), Y)]$.

The pseudocode for FACILE is provided in Algorithm 1, and we further illustrate the FACILE algorithm in figure 3. Given an input set $s_i$ comprising instances $x_1, \ldots, x_a$, the feature map $e$ is employed to extract instance-level features for all the instances within the input set. Subsequently, a set-input model $g$ is utilized to generate set-level features based on the instance-level features. Our FACILE framework is designed to be a generic algorithm that is compatible with any supervised learning method in its pre-training stage. We chose SupCon (Supervised Contrastive Learning) (Khosla et al., 2020) and FSP as they are representative of the two main approaches within supervised learning: contrastive and non-contrastive (traditional supervised) learning, respectively. During testing, we extract the pre-trained feature map $\hat{e}$ and fine-tune a classifier $f$ using the generated embeddings from $\hat{e}$ and the fine-grained labels of the support set. The performance of the classifier $\hat{f}$ is then reported for the query set. Note that Algorithm 1 is generic since the two learning steps can use any supervised learning algorithm.

## 2.2 THEORETICAL ANALYSIS

We denote the underlying distribution of $\mathcal{D}_m^{\mathrm{cg}}$ as $P_{S,W}$ and the underlying distribution of $\mathcal{D}_n^{\mathrm{fg}}$ as $P_{X,Y}$. We assume the joint distribution of $Z$ and $Y$ is $P_{Z,Y}$. After we learn the feature map $\hat{e}$, we can define a new distribution $\hat{P}_{Z,Y} = P(Z, Y)\mathbb{1}\{Z = \hat{e}(X)\}$, where $\mathbb{1}$ is the indicator function. The $\mathcal{D}_n^{\mathrm{fg,aug}}$ is i.i.d. samples from $\hat{P}_{Z,Y}$. In order to include the underlying distribution of $\mathcal{D}_m^{\mathrm{cg}}$, and $\mathcal{D}_n^{\mathrm{fg}}$ into analysis, with a slight abuse of notation we use $\mathcal{A}_m(\ell^{\mathrm{cg}}, P_{S,W}, \mathcal{E})$ to denote $\mathcal{A}(\ell^{\mathrm{cg}}, \mathcal{D}_m^{\mathrm{cg}}, \mathcal{E})$ and use $\mathcal{A}_n(\ell^{\mathrm{fg}}, \hat{P}_{Z,Y}, \mathcal{F})$ to denote $\mathcal{A}(\ell^{\mathrm{fg}}, \mathcal{D}_n^{\mathrm{fg,aug}}, \mathcal{F})$. The two learning algorithms are described as follows.

**Definition 1.** (Coarse-grained learning; pretraining) Let $\mathrm{Rate}_m(\ell^{\mathrm{cg}}, P_{S,W}, \mathcal{E}; \delta)$ (abbreviated to $\mathrm{Rate}_m(\ell^{\mathrm{cg}}, P_{S,W}, \mathcal{E})$) be the rate of $\mathcal{A}_m(\ell^{\mathrm{cg}}, P_{S,W}, \mathcal{E})$ which takes $\ell^{\mathrm{cg}}$, $\mathcal{E}$ and $m$ i.i.d. observations from $P_{S,W}$ as input, and return a feature map $\hat{e} \in \mathcal{E}$ such that

$$\mathbb{E}_{P_{S,W}} \ell_{\hat{e}}^{\mathrm{cg}}(S, W) \leq \mathrm{Rate}_m(\ell^{\mathrm{cg}}, P_{S,W}, \mathcal{E}; \delta)$$

with probability at least $1 - \delta$.

**Definition 2.** (Fine-grained learning; downstream task learning) Let $\text{Rate}_n(\ell^{\text{fg}}, P_{Z,Y}, \mathcal{F}; \delta)$ (abbreviated to $\text{Rate}_n(\ell^{\text{fg}}, P_{Z,Y}, \mathcal{F})$) be the excess risk rate of $\mathcal{A}_n(\ell^{\text{fg}}, P_{Z,Y}, \mathcal{F})$ which take $\ell^{\text{fg}}$, $\mathcal{F}$, and $n$ i.i.d. observations from a distribution $P_{Z,Y}$ as input, and returns a fine-grained predictor $\hat{f} \in \mathcal{F}$ such that $\mathbb{E}_{P_{Z,Y}}\left[\ell^{\text{fg}}_{\hat{f}}(Z,Y) - \ell^{\text{fg}}_{f^*}(Z,Y)\right] \leq \text{Rate}_n(\ell^{\text{cg}}, P_{Z,Y}, \mathcal{F}; \delta)$ with probability at least $1 - \delta$.

Next, we introduce our relative Lipschitz assumption and the central condition for quantifying task relatedness. The Lipschitz property requires that small perturbations to the feature map $e$ that do not harm the pre-training task, do not affect the loss of downstream task much either.

**Definition 3.** We say that $f$ is $L$-Lipschitz relative to $\mathcal{E}$ if for all $s \in \mathcal{S}$, $x \in s$, $y \in \mathcal{Y}$, and $e, e' \in \mathcal{E}$,

$$|\ell^{\text{fg}}(f \circ e(x), y) - \ell^{\text{fg}}(f \circ e'(x), y)| \leq L\ell^{\text{cg}}(g_e \circ \phi^e(s), g_{e'} \circ \phi^{e'}(s))$$

The function class $\mathcal{F}$ is $L$-Lipschitz relative to $\mathcal{E}$, if every $f \in \mathcal{F}$ is $L$-Lipschitz relative to $\mathcal{E}$.

Definition 3 generalizes the definition of $L$-Lipschitzness in Robinson et al. (2020) to bound the downstream loss deviation through the loss of the set label predictions. In the special case where $s = \{x\}$, and $g$ is a classifier for the pretraining labels, our Lipschitz condition reduces to the Lipschitzness definition of Robinson et al. (2020).

The central condition is well-known to yield fast rates for supervised learning (Van Erven et al., 2015). Please refer to definition 6 for the definition of central condition. We show that our surrogate problem $(\ell^{\text{fg}}, \hat{P}_{Z,Y}, \mathcal{F})$ satisfies a central condition in proposition 7.

**Theorem 4.** *Suppose that $(\ell^{\text{fg}}, P_{Z,Y}, \mathcal{F})$ satisfies the central condition, $\mathcal{F}$ is $L$-Lipschitz relative to $\mathcal{E}$, $\ell^{\text{fg}}$ is bounded by $B > 0$, $\mathcal{F}$ is $L'$-Lipschitz in its $d$-dimensional parameters in the $l_2$ norm, $\mathcal{F}$ is contained in the Euclidean ball of radius $R$, and $\mathcal{Y}$ is compact. We also assume that $\text{Rate}_m(\ell^{\text{cg}}, P_{S,W}, \mathcal{E}) = \mathcal{O}\left(1/m^\alpha\right)$. Then when $\mathcal{A}_n(\ell^{\text{fg}}, \hat{P}_{Z,Y}, \mathcal{F})$ is ERM we obtain excess risk $\mathbb{E}_{P_{X,Y}}\left[\ell^{\text{fg}}_{\hat{f} \circ \hat{e}}(X,Y) - \ell^{\text{fg}}_{f^* \circ e^*}(X,Y)\right]$ bound with probability at least $1 - \delta$ by $\mathcal{O}\left(\frac{d\alpha\beta \log RL'n + \log \frac{1}{\delta}}{n} + \frac{B+2L}{n^{\alpha\beta}}\right)$ if $m = \Omega(n^\beta)$ and $\ell^{\text{cg}}(w, w') = \mathbb{1}\{w \neq w'\}$.*

For a typical scenario where $\text{Rate}_m(\ell^{\text{cg}}, P_{S,W}, \mathcal{E}) = \mathcal{O}(1/\sqrt{m})$, we can obtain fast rates with $m = \Omega(n^2)$. Similarly, in the scenario where $\mathcal{A}_m(\ell^{\text{cg}}, P_{S,W}, \mathcal{E})$ achieves fast rate, i.e., $\text{Rate}_m(\ell^{\text{cg}}, P_{S,W}, \mathcal{E}) = \mathcal{O}(1/m)$, one can obtains fast rates when $m = \Omega(n)$. More generally, if $\alpha\beta \geq 1$, we observe fast rates.

We prove our theorem by first showing that the excess risk of $\hat{f} \circ \hat{e}$ can be bounded by $2L\text{Rate}_m(\ell^{\text{cg}}, P_{S,W}, \mathcal{E}) + \text{Rate}_n(\ell^{\text{fg}}, \hat{P}_{Z,Y}, \mathcal{F})$ in proposition 5. Then, we show that $(\ell^{\text{fg}}, \hat{P}_{Z,Y}, \mathcal{F})$ also satisfies the weak central condition in proposition 7. Thus, $\text{Rate}_n(\ell^{\text{fg}}, \hat{P}_{Z,Y}, \mathcal{F})$ is also bounded by proposition 8. We refer interested readers to section H for full details of the proof.

In the next section, we first aim to empirically study the relationship between generalization error, coarse-grained dataset size, and fine-grained dataset size that our theoretical analysis predicts in section 3.3 and section 3.4. We also demonstrate the exceptional efficacy of the proposed algorithm compared to baseline models on natural image datasets and histopathology image datasets.

## 3 EMPIRICAL STUDY

### 3.1 BASELINE MODELS AND ALGORITHM INSTANTIATION

We consider two sets of baseline models: self-supervised models (Bachman et al., 2019; He et al., 2020; Chen et al., 2020; Caron et al., 2020; Grill et al., 2020; Chen & He, 2021) and weakly supervised models (Donahue et al., 2014; Sun et al., 2017; Zeiler & Fergus, 2014; Robinson et al., 2020).

**Self-supervised models** Given pre-training data $(S, W)$, self-supervised learning models ignore the labels $W$ and learn $\hat{e}$ from $S$. Then, we can test $\hat{e}$ with a new task, which consists of a support set and a query set. A new model that leverages the learned $\hat{e}$ is fine-tuned on the support set and tested on the query set. We performed two self-supervised learning models in two categories, e.g., SimCLR (Chen et al., 2020) for contrastive learning and SimSiam (Chen & He, 2021) for non-contrastive learning. Details of these self-supervised learning algorithms can be found in section G.

**Weakly supervised models** We assign each instance, from the pre-training dataset, a label of the input set to which it belongs. We train feature map $\hat{e}$ appended with a linear classifier on the pre-training dataset. We call this model FSP-Patch, where FSP stands for fully supervised pre-training and the model is trained with the assigned instance-level labels. For a new task with a support set and a query set, we use the $\hat{e}$ to extract features for both sets, train a classifier on the support set features, and test the classifier on the query set features.

Following previous work in FSL (Tian et al., 2020; Chen et al., 2019), we use $l_2$-normalized features for downstream tasks. Unless otherwise specified, we evaluate methods with 1,000 randomly sampled meta-tasks from each dataset. All meta-tasks use 15 samples per class as the query set. The average F1/ACC and 95% confidence interval (CI) are reported. We follow the test setting of Yang et al. (2022) and use NearestCentroid (NC), LogisticRegression (LR), and RidgeClassifier (RC).

## 3.2 PRETRAIN WITH UNIQUE CLASS NUMBER OF INPUT SETS

In order to show the advantages of using the coarse-grained labels, we introduce a new task of pre-training with the unique class number of input sets. Inspired by Lee et al. (2019a), we use the CIFAR-100 (Krizhevsky et al., 2009) dataset, which contains 100 classes grouped into 20 super-classes. We generate input sets by sampling between 6 and 10 images from CIFAR-100 training data. The targets of the input sets are the unique superclass number of the input sets. In our downstream tasks, we perform few-shot classifications of fine-grained classes. Despite being distinct from the downstream fine-grained labels, the coarse-grained labels offer useful information for learning useful representations for downstream tasks.

| pretraining method | unique superclass number | | | most frequent superclass | | |
|---|---|---|---|---|---|---|
| | NC | LR | RC | NC | LR | RC |
| SimCLR | $76.07 \pm 0.97$ | $75.88 \pm 1.01$ | $75.50 \pm 1.02$ | $75.91 \pm 1.00$ | $75.82 \pm 1.01$ | $75.91 \pm 1.02$ |
| SimSiam | $78.15 \pm 0.93$ | $79.44 \pm 0.92$ | $79.03 \pm 0.95$ | $78.80 \pm 0.93$ | $79.44 \pm 0.95$ | $79.43 \pm 0.93$ |
| FSP-Patch | N/A | N/A | N/A | $73.21 \pm 0.97$ | $73.92 \pm 0.98$ | $73.40 \pm 0.98$ |
| FACILE-SupCon | N/A | N/A | N/A | $79.54 \pm 0.92$ | $79.54 \pm 0.96$ | $79.12 \pm 0.95$ |
| FACILE-FSP | $\mathbf{86.25 \pm 0.79}$ | $\mathbf{85.42 \pm 0.82}$ | $\mathbf{85.84 \pm 0.81}$ | $\mathbf{82.04 \pm 0.84}$ | $\mathbf{81.70 \pm 0.91}$ | $\mathbf{81.75 \pm 0.90}$ |

Table 1: Pretraining on input sets from CIFAR-100. Testing with 5-shot 5-way meta-test sets; average F1 and CI are reported.

The ResNet18 (He et al., 2016) is used as feature maps $\hat{e}$. For FACILE-FSP, we pre-train the feature map $\hat{e}$ from these input sets and targets with $\ell_1$ loss. The features of CIFAR-100 test images are extracted with $\hat{e}$. Training settings of SimSiam, SimCLR and FACILE-FSP can be found in section A.1. We then test $\hat{e}$ in a few-shot manner. We random sample 5 classes, 5 examples from each class, for each meta-test dataset. The fine-grained label predictor $\hat{f}$ is trained on the support examples and tested on the query examples. The performance of these models is reported in table 1. We can see that FACILE-FSP outperforms self-supervised learning models by a large margin.

## 3.3 PRETRAIN WITH MOST FREQUENT CLASS LABEL

We sample input sets randomly from training data of CIFAR-100. The targets are the most frequent superclass of the input sets. If there is a tie in an input set, we choose a random top frequent superclass as the target of the input set. Training settings are similar to section 3.2 and can be found in section A.1. The performances of all models are reported in table 1. We can see that FACILE-FSP obtains better results compared to other models.

Note that the excess risk bound of the form $b = C/n^\gamma$ implies a log-linear relationship $\log b = \log C - \gamma \log n$ between the error and the number of fine-grained labels. We can visually interpret the learning rate $\gamma$. We study two cases: when the number of coarse-grained labels $m$ grows linearly with the number of fine-grained labels, and when the number of coarse-grained labels $m$ grows quadratically with the number of fine-grained labels. In order to show the generalization error rate of FACILE-FSP w.r.t. fine-grained label number on CIFAR-100 test dataset, we randomly sample 5 classes (i.e., 5-way testing) for each task. We then sample $n/5$ fine-grained examples in each class for the support set and sample 15 examples for each class for the query set. The curves are shown in figure 4. The figure shows the log-linear relationship of FACILE-FSP's generalization error on downstream tasks w.r.t. fine-grained label number. This visualization effectively captures how coarse-grained label number $m$ impacts the model's generalization capabilities.

## 3.4 EVALUATION ON HISTOPATHOLOGY IMAGES

**Datasets and data extraction**   We pretrain our models using two independent sources of WSIs. First, we downloaded data from The Cancer Genome Atlas (TCGA) from the NCI Genomic Data Commons (GDC) (Heath et al., 2021). Two collections of non-overlapping patches with different patch sizes, i.e., $224 \times 224$ and $1,000 \times 1,000$ at 20X magnification. Background patches with high or low intensity were removed. Because the number of patches generated with size $224 \times 224$ at 20X magnification is very large, at most $1,000$ randomly selected patches are kept for each slide. The names of the tumors/organs, from which slides are collected, are used as coarse-grained labels. Second, we downloaded all clinical slides from the Genotype-Tissue Expression (GTEx)

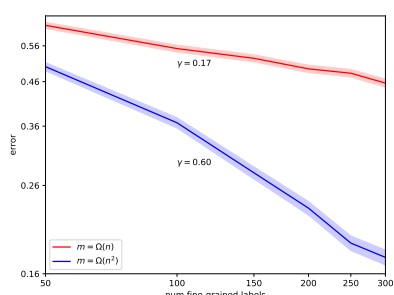

Figure 4: Generalization error (with two growth rates) of FACILE-FSP on CIFAR-100 test dataset as a function of the number of coarse-grained labels $m$.

project (Lonsdale et al., 2013), which provides a resource for studying human gene expression and regulation in relation to genetic variation. We extracted non-overlapping patches with size $1,000 \times 1,000$ at 20X magnification and patches with intensity larger than 0.1 and smaller than 0.85 are kept. For these slides, we used the organs from which the tissues were extracted as coarse-grained labels. Examples and class distributions for the two datasets can be found in section C.

We test models on 3 public datasets: LC (Borkowski et al., 2021), PAIP (Kim et al., 2021), NCT (Kather et al., 2018) and 1 private dataset PDAC. Details of these datasets are deferred to section C. Note that the TCGA and GTEx have meticulously categorized an extensive array of cancer types and organs, covering a diverse range of tissues as outlined in the LC, PAIP, and NCT. The strategic use of WSI-level labels is rooted in their potential to enrich tissue-level classification. While these labels may appear broad, they encapsulate a wealth of underlying heterogeneity inherent to different cancer regions and tissue types.

**Pretrain on TCGA with patch size** $224 \times 224$ We first train models on TCGA patches with size $224 \times 224$ at 20X magnification. After the models are trained, we test the feature map in these models on LC, PAIP, and NCT. Full details about FACILE-FSP, FACILE-SupCon, and baseline models' training settings can be found in section A.3. Latent augmentation (LA) has been shown to improve FSL performance for histopathology images (Yang et al., 2022). We use faiss (Johnson et al., 2019) to perform k-means clustering. Following the setting of Yang et al. (2022), the number of prototypes in the base dictionary is 16. Each sample is augmented 100 times by LA. We refer readers to section E for details of LA.

**Main results** The test result is shown in table 2. In order to show the performance improvement over models pre-trained on natural image datasets, we report the performance of the FSP model pre-trained on ImageNet. We can see from table 2 that our model FACILE-FSP performs the best, with a large margin compared to other models. The contrastive learning model SimCLR performs worse than non-contrastive learning model SimSiam. A possible reason could be the small batch size we used for SimCLR. SimSiam maintains high performance even with small batch sizes. FSP-Patch achieves better performance compared to self-supervised learning models and the ImageNet pretrained model, which shows the usefulness of the coarse-grained labels for downstream tasks. More experiment results about test ACC on LC, PAIP and NCT datasets can be found in section B.2.1. Test result with larger shot number is in section B.2.2.

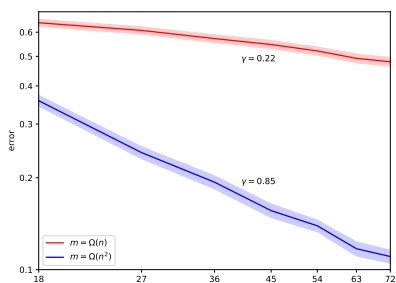

Figure 5: Generalization error on NCT dataset. The FACILE-FSP trains on TCGA dataset with $m$ coarse-grained labels. We show the error curve with two growth rates of $m$.

We further pre-train models on GTEx and TCGA with patch size $1,000 \times 1,000$ and test the models on our private dataset PDAC. We refer readers to section B.5 for experiment results on PDAC dataset.

We show the generalization error of FACILE-FSP w.r.t. fine-grained label number in figure 5. The figure reveals a pronounced log-linear relationship. A larger growth rate of coarse-grained labels implies a faster rate of excess risk.

| pretraining method | NC | LR | RC | LR+LA | RC+LA |
|---|---|---|---|---|---|
| 1-shot 5-way test on LC dataset | | | | | |
| ImageNet (FSP) | $63.26 \pm 1.46$ | $63.13 \pm 1.41$ | $63.24 \pm 1.40$ | $64.51 \pm 1.41$ | $64.95 \pm 1.39$ |
| SimSiam | $65.83 \pm 1.32$ | $66.52 \pm 1.31$ | $66.24 \pm 1.32$ | $67.21 \pm 1.29$ | $67.83 \pm 1.33$ |
| SimCLR | $64.57 \pm 1.36$ | $63.85 \pm 1.37$ | $64.16 \pm 1.37$ | $65.78 \pm 1.33$ | $66.81 \pm 1.40$ |
| FSP-Patch | $66.73 \pm 1.29$ | $66.25 \pm 1.29$ | $66.59 \pm 1.28$ | $68.01 \pm 1.24$ | $68.28 \pm 1.26$ |
| FACILE-SupCon | $74.91 \pm 1.25$ | $\mathbf{76.23 \pm 1.16}$ | $75.01 \pm 1.19$ | $75.60 \pm 1.19$ | $\mathbf{75.64 \pm 1.18}$ |
| FACILE-FSP | $\mathbf{77.39 \pm 1.21}$ | $76.14 \pm 1.25$ | $\mathbf{75.18 \pm 1.30}$ | $\mathbf{77.55 \pm 1.17}$ | $73.72 \pm 1.34$ |
| 5-shot 5-way test on LC dataset | | | | | |
| ImageNet (FSP) | $82.82 \pm 0.75$ | $80.13 \pm 0.82$ | $80.23 \pm 0.83$ | $84.70 \pm 0.70$ | $84.42 \pm 0.74$ |
| SimSiam | $85.12 \pm 0.68$ | $82.69 \pm 0.75$ | $82.80 \pm 0.76$ | $87.45 \pm 0.63$ | $87.50 \pm 0.66$ |
| SimCLR | $83.45 \pm 0.77$ | $81.93 \pm 0.83$ | $81.40 \pm 0.89$ | $85.69 \pm 0.73$ | $84.93 \pm 0.79$ |
| FSP-Patch | $84.96 \pm 0.64$ | $84.10 \pm 0.69$ | $84.45 \pm 0.68$ | $86.31 \pm 0.65$ | $86.29 \pm 0.68$ |
| FACILE-SupCon | $91.09 \pm 0.47$ | $90.34 \pm 0.48$ | $90.25 \pm 0.48$ | $91.32 \pm 0.47$ | $\mathbf{90.94 \pm 0.50}$ |
| FACILE-FSP | $\mathbf{91.67 \pm 0.45}$ | $\mathbf{90.64 \pm 0.50}$ | $\mathbf{90.52 \pm 0.52}$ | $\mathbf{92.07 \pm 0.48}$ | $89.81 \pm 0.61$ |
| 1-shot 3-way test on PAIP dataset | | | | | |
| ImageNet (FSP) | $45.96 \pm 1.22$ | $47.82 \pm 1.29$ | $47.43 \pm 1.29$ | $46.38 \pm 1.24$ | $44.90 \pm 1.24$ |
| SimSiam | $46.43 \pm 1.21$ | $47.93 \pm 1.24$ | $47.74 \pm 1.23$ | $47.20 \pm 1.21$ | $46.31 \pm 1.22$ |
| SimCLR | $44.51 \pm 1.16$ | $46.44 \pm 1.14$ | $45.59 \pm 1.15$ | $45.40 \pm 1.14$ | $45.04 \pm 1.16$ |
| FSP-Patch | $\mathbf{48.85 \pm 1.21}$ | $\mathbf{49.44 \pm 1.26}$ | $\mathbf{50.27 \pm 1.22}$ | $\mathbf{49.76 \pm 1.20}$ | $\mathbf{48.44 \pm 1.21}$ |
| FACILE-SupCon | $46.60 \pm 1.20$ | $48.63 \pm 1.22$ | $48.46 \pm 1.21$ | $47.13 \pm 1.20$ | $47.87 \pm 1.22$ |
| FACILE-FSP | $45.40 \pm 1.24$ | $46.71 \pm 1.20$ | $46.60 \pm 1.21$ | $46.36 \pm 1.22$ | $45.49 \pm 1.20$ |
| 5-shot 3-way test on PAIP dataset | | | | | |
| ImageNet (FSP) | $60.73 \pm 1.02$ | $61.21 \pm 1.12$ | $61.04 \pm 1.11$ | $61.66 \pm 0.91$ | $59.30 \pm 0.93$ |
| SimSiam | $62.88 \pm 0.97$ | $62.59 \pm 1.08$ | $63.48 \pm 1.04$ | $65.01 \pm 0.88$ | $63.22 \pm 0.89$ |
| SimCLR | $60.99 \pm 0.93$ | $61.38 \pm 1.00$ | $61.62 \pm 1.02$ | $62.39 \pm 0.91$ | $61.29 \pm 0.90$ |
| FSP-Patch | $64.45 \pm 0.92$ | $64.60 \pm 0.98$ | $64.49 \pm 0.99$ | $64.08 \pm 0.89$ | $62.79 \pm 0.89$ |
| FACILE-SupCon | $\mathbf{64.74 \pm 0.91}$ | $\mathbf{65.63 \pm 0.97}$ | $\mathbf{65.93 \pm 0.97}$ | $66.68 \pm 0.86$ | $\mathbf{66.48 \pm 0.82}$ |
| FACILE-FSP | $63.90 \pm 0.94$ | $64.59 \pm 0.96$ | $65.43 \pm 0.96$ | $\mathbf{66.77 \pm 0.86}$ | $66.34 \pm 0.85$ |
| 1-shot 9-way test on NCT dataset | | | | | |
| ImageNet (FSP) | $57.35 \pm 1.68$ | $56.39 \pm 1.64$ | $56.08 \pm 1.64$ | $57.78 \pm 1.66$ | $55.85 \pm 1.64$ |
| SimSiam | $63.60 \pm 1.62$ | $64.43 \pm 1.54$ | $64.79 \pm 1.53$ | $65.26 \pm 1.56$ | $65.39 \pm 1.53$ |
| SimCLR | $59.73 \pm 1.57$ | $59.61 \pm 1.57$ | $59.34 \pm 1.56$ | $60.57 \pm 1.57$ | $60.99 \pm 1.53$ |
| FSP-Patch | $60.08 \pm 1.46$ | $61.55 \pm 1.50$ | $62.32 \pm 1.50$ | $61.99 \pm 1.42$ | $60.62 \pm 1.38$ |
| FACILE-SupCon | $\mathbf{68.10 \pm 1.29}$ | $\mathbf{69.63 \pm 1.25}$ | $\mathbf{69.81 \pm 1.24}$ | $\mathbf{69.54 \pm 1.25}$ | $\mathbf{69.77 \pm 1.22}$ |
| FACILE-FSP | $66.38 \pm 1.38$ | $67.03 \pm 1.34$ | $67.56 \pm 1.32$ | $68.35 \pm 1.33$ | $\mathbf{69.77 \pm 1.30}$ |
| 5-shot 9-way test on NCT dataset | | | | | |
| ImageNet (FSP) | $74.59 \pm 1.11$ | $73.21 \pm 1.13$ | $74.60 \pm 1.07$ | $76.68 \pm 1.04$ | $74.39 \pm 1.09$ |
| SimSiam | $79.97 \pm 1.05$ | $79.81 \pm 1.03$ | $80.84 \pm 0.98$ | $83.45 \pm 0.92$ | $83.61 \pm 0.90$ |
| SimCLR | $76.80 \pm 1.09$ | $76.95 \pm 1.07$ | $78.25 \pm 1.03$ | $80.54 \pm 0.97$ | $81.13 \pm 0.95$ |
| FSP-Patch | $79.50 \pm 0.94$ | $79.54 \pm 0.95$ | $81.00 \pm 0.88$ | $82.42 \pm 0.81$ | $81.33 \pm 0.79$ |
| FACILE-SupCon | $\mathbf{86.79 \pm 0.61}$ | $\mathbf{87.89 \pm 0.58}$ | $\mathbf{89.10 \pm 0.52}$ | $\mathbf{89.53 \pm 0.52}$ | $\mathbf{88.58 \pm 0.54}$ |
| FACILE-FSP | $84.68 \pm 0.74$ | $85.47 \pm 0.72$ | $87.44 \pm 0.64$ | $88.00 \pm 0.63$ | $87.51 \pm 0.66$ |

Table 2: Test result on LC, PAIP, and NCT dataset; average F1 and CI are reported.

**Benefits of pre-training on Large Pathology Datasets** In order to show the benefits of pre-training on large pathology datasets, we pre-train different models on the NCT training dataset and test the performance on the LC dataset, following the setting of Yang et al. (2022). Instead of separating the mixture-domain and out-domain tasks, we directly report the average F1 and CI of LR models over all 5 classes of the LC dataset. Training details of the models can be found in section B.4. The test result on the LC dataset is shown in table 3.

We can see from table 3 the best model pre-trained on NCT, i.e., FSP with strong augmentation, performs worse than our model FACILE-FSP in table 2. Our method get roughly 13% improvement compared to Yang et al. (2022) on the LC dataset. The large margin between the two best models pre-trained on two different datasets shows the importance of pre-training with a large number of coarse-grained labels. More results on LC and PAIP can be found in figure 8. Note that SimSiam model, trained with a batch size of 55, maintains competitive performance to MoCo v3 which needs a large batch size.

| pretraining method | NC | LR | RC | LR+LA | RC+LA |
|---|---|---|---|---|---|
| SimSiam | $76.21 \pm 0.87$ | $74.05 \pm 1.10$ | $74.59 \pm 1.10$ | $77.87 \pm 0.87$ | $76.03 \pm 0.94$ |
| MoCo v3 ((Yang et al., 2022)) | $72.82 \pm 1.25$ | $70.29 \pm 1.43$ | $71.31 \pm 1.40$ | $78.72 \pm 1.00$ | $79.71 \pm 0.95$ |
| FSP (simple aug; (Yang et al., 2022)) | $56.44 \pm 1.50$ | $52.27 \pm 1.81$ | $55.62 \pm 1.74$ | $63.47 \pm 1.37$ | $63.47 \pm 1.46$ |
| FSP (strong aug) | $\mathbf{83.53 \pm 0.79}$ | $\mathbf{80.81 \pm 1.01}$ | $\mathbf{80.27 \pm 1.08}$ | $\mathbf{85.57 \pm 0.77}$ | $\mathbf{84.06 \pm 0.89}$ |
| SupCon | $81.51 \pm 0.85$ | $78.77 \pm 1.03$ | $78.65 \pm 1.08$ | $83.51 \pm 0.84$ | $83.31 \pm 0.91$ |

Table 3: Pretraining on NCT and 5-shot 5-way testing on LC dataset; average F1 and CI are reported.

**More experiments and ablation study** We refer interested readers to section F for ablation studies about set size, training procedures, and set-input models. We provide additional insights through

fine-tuning experiments. In Appendix B.1, we detail the fine-tuning of the ViT-B/16 (Dosovitskiy et al., 2020) from CLIP (Radford et al., 2021) using CUB200-based anomaly detection data (He & Peng, 2019). Similarly, Appendix B.3 discusses fine-tuning the ViT-B/14 model of DINO V2 (Oquab et al., 2023) on TCGA dataset. These experiments extend our analysis to specialized tasks, showcasing the adaptability of FACILE to foundation models.

# 4 RELATED WORK

**Weakly supervised learning** The concept of weakly supervised learning is introduced as a means to alleviate the annotation bottleneck in the training of machine learning models. There has been a large body of existing work in learning with only weak labels. A comprehensive survey about weakly supervised learning is provided in Zhou (2018); Zhang et al. (2022). We study a novel form of weak supervision which is provided by set-level coarse-grained labels. Among weakly supervised learning methods, Robinson et al. (2020) studied the generalization properties of weakly supervised learning and proposed a generic learning algorithm that can learn from weak and strong labels and can be proved to achieve a fast rate. The authors consider a different setting where each instance has a weak label and a strong label, and the strong label predictor learns to predict the strong labels from the instances and their corresponding embeddings learned with weak labels. We consider the setting where we have some coarse-grained labels of some sets, rather than instances and the downstream classifiers only use the learned embeddings to train and test on the downstream tasks.

**Multiple-instance learning for WSIs** WSI classification and regression are formulated based on multiple-instance learning (MIL) (Campanella et al., 2019; Xu et al., 2022; Ilse et al., 2018; Sharma et al., 2021; Hashimoto et al., 2020; Shao et al., 2021; Yao et al., 2020; Lu et al., 2021b;a; Chen et al., 2021b; Li et al., 2021; Chen et al., 2021a; Myronenko et al., 2021; Xiang & Zhang, 2022; Javed et al., 2022). These MIL models employ two procedures: i) feature extraction for patches cropped from a WSI and ii) aggregation of features from the same WSI. ImageNet pretrained backbones, self-supervised backbones pretrained on histopathology images, or backbones fine-tuned during training are used to extract features from patches. Deep attention pooling, graph neural networks, or sequence models, adapted for WSIs, are used for feature aggregation. In this paper, we consider a different problem setting where we enhance patch-level classification with related set-level labels. In the application of histopathology images, line 2 of our generic algorithm can be instantiated with any MIL models that have the backbones with trainable modules to extract patch-level features, e.g., Ilse et al. (2018). A complete comparison of MIL models for WSIs is out of the scope of this paper.

**Learning from coarsely-labeled data** Another related line of research is Phoo & Hariharan (2021), where the authors assume a taxonomy of classes with two levels, i.e., a set of fine-grained classes that are more challenging to annotate and a set of coarse-grained classes that are easier to annotate. In our paper, we do not assume a taxonomy of classes for the coarse-grained and fine-grained labels. The coarse-grained and fine-grained labels are closely related via a hierarchy. Also, the inputs that are fed to models to predict the coarse-grained or fine-grained labels are different, i.e., set input for coarse-grained labels and instances for fine-grained labels.

# 5 CONCLUSION AND DISCUSSION

**Summary** We introduce FACILE, a representation learning framework that leverages coarse-grained labels for model training and enhances model performance for downstream tasks. Our theoretical analysis highlights the significant potential of leveraging set-level labels to benefit the learning process of fine-grained label prediction tasks. To demonstrate the effectiveness of FACILE, we conduct pre-training on CIFAR-100-based datasets and two large public histopathology datasets using coarse-grained labels and evaluate our model on a diverse collection of datasets with fine-grained labels.

**Limitation and future work** In this paper, we consider a novel problem setting where we enhance downstream fine-grained label classification with easily available coarse-grained labels and propose a generic algorithm that contains two supervised learning steps. It is important to note that the separate utilization of loosely related coarse-grained labels and fine-grained labels can be expensive. Specifically, the pre-training of our proposed algorithm could be expensive given large amounts of coarse-grained data and the nature of the set-input data. For this reason, we are investigating methods of selecting a subset of the coarse-grained dataset to accelerate pre-training.

## 6 Acknowledgement

The research was funded in part by the Center for Translational Data Science at the University of Chicago, the Research Computing Center at the University of Chicago, and the National Science Foundation under Grant No. 2313130. We extend our deepest gratitude to Christopher R. Weber for his generous provision of the PDAC dataset, which was invaluable to the completion of this research.

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

# A  TRAINING DETAILS

## A.1  PRETRAIN WITH UNIQUE CLASS NUMBER AND MOST FREQUENT CLASS OF INPUT SETS

In our study, an epoch refers to going through all the input sets in the dataset once. SimSiam is trained for 2,000 epochs using a batch size of 512. SGD is employed with a learning rate of 0.1, weight decay of 1e-4, and momentum of 0.9. The training process incorporates a cosine scheduler. Similarly, SimCLR is trained for 2,000 epochs with a batch size of 256 and a temperature of 0.07. SGD is used with a learning rate of 0.05, weight decay of 1e-4, and momentum of 0.9. The training also utilizes a cosine scheduler.

We train FSP-Patch for 800 epochs with a batch size of 256. The SGD is used with a weight decay of 1e-4, momentum of 0.9, and cosine scheduler.

FACILE-FSP is trained for 800 epochs with a batch size of 64. SGD is used with a learning rate of 0.0125, weight decay of 1e-4, and momentum of 0.9. $\ell_1$ loss is optimized for pretraining with unique class numbers of input sets. For FACILE-SupCon, we train the model with 2,000 epochs and a batch size of 256. An additional temperature parameter is set to 0.07. The SGD is used with a learning rate of 0.05, weight decay of 1e-4, and momentum of 0.9.

## A.2  FINE-TUNE VIT-B/16 OF CLIP WITH CUB200

SimSiam is trained for 400 epochs using a batch size of 64. SGD is used with an initial learning rate of 0.0125, weight decay of 1e-4, and momentum of 0.9. The cosine scheduler is used for the optimizer. SimCLR is also trained for 400 epochs with a batch size of 64. An additional temperature parameter is set to 0.07. SGD is used with a learning rate of 0.0125, weight decay of 1e-4, and momentum of 0.9. The training also uses a cosine scheduler.

FACILE-FSP is trained for 200 epochs with a batch size of 64. SGD is used with a learning rate of 0.0125, weight decay of 1e-4, and momentum of 0.9. For FACILE-SupCon, we train the model with 800 epochs and a batch size of 64. An additional temperature parameter is set to 0.07. The SGD is used with an initial learning rate of 0.0125, weight decay of 1e-4, and momentum of 0.9. Both models' training utilized a cosine annealing scheduler.

## A.3  PRETRAIN RESNET18 WITH TCGA AND GTEX DATASET

In SimSiam, SimCLR, and FSP-Patch models, the data loader samples one patch for each slide. In FACILE-FSP and FACILE-SupCon, the data loader samples a set of $a$ patches for each slide.

SimSiam is trained for 5,000 epochs using a batch size of 55. SGD is employed with a learning rate of 0.01, weight decay of 1e-4, and momentum of 0.9. The training process incorporates a cosine scheduler. Similarly, SimCLR is trained for 5,000 epochs with a batch size of 32. An additional temperature parameter is set to 0.07. SGD is used with a learning rate of 0.006, weight decay of 1e-4, and momentum of 0.9. The training also utilizes a cosine scheduler.

FSP-Patch is trained for 1,000 epochs with a batch size of 64. We employ SGD with a learning rate of 0.05, weight decay of 1e-4, and momentum of 0.9. The training process includes the utilization of a cosine scheduler.

FACILE-FSP is trained for 3,000 epochs with batch size 32. The input set size is 5 by default. We employ SGD with a learning rate of 0.0125, weight decay of 1e-4, and momentum of 0.9. The training process includes the utilization of a cosine scheduler. Set Transformer with 3 inducing points and 4 attention heads is used for the set-input model $g$. Similarly, for our FACILE-SupCon model, we use the same input set size and set-input model. The training process is configured with a batch size of 32 and extends over 3,000 epochs. An additional temperature parameter is set to 0.07. We use SGD with a learning rate of 0.00625, weight decay of 1e-4, and momentum of 0.9. We use an MLP as a projection head with two fc layers, a hidden dimension of 512, and an output dimension of 512.

## A.4 FINE-TUNE VIT-B/14 OF DINO V2 WITH TCGA

SimSiam is trained for 400 epochs with a batch size of 64, utilizing Stochastic Gradient Descent (SGD) with an initial learning rate of 0.0125, a weight decay of 1e-4, and a momentum of 0.9. A cosine scheduler was employed. SimCLR underwent a similar training regimen for 400 epochs and a batch size of 64, with an additional temperature parameter set at 0.07 and identical SGD parameters, including the use of a cosine scheduler for learning rate adjustments.

FSP-Patch also completed 400 epochs of training with a batch size of 64. The model employed SGD with a learning rate of 0.0125, a weight decay of 1e-4, and a momentum of 0.9, along with a cosine scheduler to modulate the learning rate.

For FACILE-FSP, training spanned 200 epochs with a batch size of 64, using SGD with the same learning rate, weight decay, and momentum settings. FACILE-SupCon extended its training to 800 epochs with a batch size of 64, including an additional temperature setting of 0.07 and the same SGD configuration. Both FACILE-FSP and FACILE-SupCon models utilized a cosine annealing scheduler.

# B ADDITIONAL RESULT

## B.1 FINE-TUNE CLIP MODEL WITH ANOMALY DETECTION DATASET

In this experiment, we sought to enhance model performance with coarse-grained labels of the anomaly detection datasets (Zaheer et al., 2017; Lee et al., 2019a). A total of 11,788 input sets of size 10 are constructed from the CUB200 (He & Peng, 2019) training dataset by including one example that lacks an attribute common to the other examples in the input set. The coarse-grained labels are the positions of the anomalies. This setup creates a challenging scenario for models, as they must identify the outlier among otherwise similar instances. In downstream tasks, we evaluate the fine-tuned feature encoder composed of the fixed CLIP (Radford et al., 2021) image encoder ViT-B/16 and appended fully-connected layer on the classification of species of the CUB200 test dataset. The batch normalization (Ioffe & Szegedy, 2015) and ReLU are applied to the fully-connected layer.

Following this experiment setup, the rationale behind utilizing coarse-grained labels is grounded in their potential to enhance model discernment in downstream tasks. By training the model to identify anomalies in sets where one item diverges from the rest, we essentially teach it to focus on subtle differences and critical attribute features. This enhanced focus is particularly beneficial for fine-grained classification tasks in the CUB200 test dataset, where distinguishing between closely related species requires the model to recognize and prioritize minute, yet significant, differences.

The model training approach in this experiment centered around the CLIP image encoder, enhanced with an additional fully-connected layer. FACILE-FSP and FACILE-SupCon incorporate this setup, utilizing the CLIP-based feature encoder and focusing on finetuning the fully-connected layer through the FACILE pretraining step. In contrast, the SimSiam approach leverages the CLIP image encoder as a backbone while finetuning the projector and predictor components. Similarly, the SimCLR method also uses the CLIP encoder as its foundation but focuses solely on finetuning the projector. These varied strategies reflect our efforts to optimize the feature encoder for accurately identifying anomalies and improving classification performance in related tasks. The training details can be found in section A.2.

| pretraining method | NC | LR | RC |
|---|---|---|---|
| CLIP (ViT-B/16) | $83.84 \pm 1.10$ | $81.01 \pm 1.23$ | $82.75 \pm 1.17$ |
| SimCLR | $84.03 \pm 1.08$ | $83.49 \pm 1.14$ | $86.30 \pm 1.03$ |
| SimSiam | $84.02 \pm 1.10$ | $83.90 \pm 1.13$ | $85.68 \pm 1.07$ |
| FACILE-SupCon | $87.49 \pm 0.99$ | $86.57 \pm 1.07$ | $88.01 \pm 0.99$ |
| FACILE-FSP | $\mathbf{88.74 \pm 0.94}$ | $\mathbf{88.45 \pm 0.96}$ | $\mathbf{88.36 \pm 0.95}$ |

Table 4: Pretraining on input sets from CUB200. Testing with 5-shot 20-way meta-test sets; average F1 and CI are reported.

Note that table 4 clearly demonstrates that all models tested benefit from incorporating data from the target domain. Notably, both FACILE-SupCon and FACILE-FSP exhibit superior performance

compared to other baseline models. This observation underscores the effectiveness of our models in leveraging coarse-grained labels to enhance their anomaly detection capabilities.

## B.2 PRETRAIN RESNET18 WITH TCGA

### B.2.1 ACC ON LC, PAIP, AND NCT DATASETS

We pretrain the models on TCGA datasets with patches size $224 \times 224$ at 20X magnification. Then, these pretrained models are tested on LC, PAIP, and NCT datasets. The average ACC and CI on the LC, PAIP, and NCT datasets are shown in table 5.

| pretraining method | NC | LR | RC | LR+LA | RC+LA |
|---|---|---|---|---|---|
| | | | 1-shot 5-way test on LC dataset | | |
| ImageNet (FSP) | $65.64 \pm 0.49$ | $66.06 \pm 0.46$ | $65.92 \pm 0.48$ | $66.60 \pm 0.48$ | $67.09 \pm 0.47$ |
| SimSiam | $68.88 \pm 0.51$ | $68.53 \pm 0.48$ | $68.27 \pm 0.48$ | $68.81 \pm 0.49$ | $70.24 \pm 0.47$ |
| SimCLR | $66.41 \pm 0.48$ | $66.52 \pm 0.46$ | $66.10 \pm 0.46$ | $67.70 \pm 0.45$ | $68.71 \pm 0.46$ |
| FSP-Patch | $68.56 \pm 0.46$ | $68.51 \pm 0.45$ | $68.68 \pm 0.46$ | $69.38 \pm 0.46$ | $69.63 \pm 0.46$ |
| FACILE-SupCon | $76.64 \pm 0.50$ | $77.88 \pm 0.47$ | $76.77 \pm 0.47$ | $77.15 \pm 0.48$ | $\mathbf{77.16 \pm 0.48}$ |
| FACILE-FSP | $\mathbf{79.01 \pm 0.49}$ | $\mathbf{78.16 \pm 0.48}$ | $\mathbf{77.43 \pm 0.50}$ | $\mathbf{79.15 \pm 0.47}$ | $75.81 \pm 0.48$ |
| | | | 5-shot 5-way test on LC dataset | | |
| ImageNet (FSP) | $82.79 \pm 0.32$ | $81.31 \pm 0.31$ | $81.13 \pm 0.30$ | $84.50 \pm 0.30$ | $84.73 \pm 0.28$ |
| SimSiam | $85.12 \pm 0.30$ | $83.39 \pm 0.32$ | $83.85 \pm 0.30$ | $87.74 \pm 0.27$ | $87.90 \pm 0.26$ |
| SimCLR | $83.75 \pm 0.30$ | $82.38 \pm 0.30$ | $82.32 \pm 0.31$ | $86.12 \pm 0.28$ | $85.40 \pm 0.30$ |
| FSP-Patch | $85.15 \pm 0.29$ | $84.38 \pm 0.31$ | $85.01 \pm 0.29$ | $86.71 \pm 0.28$ | $86.24 \pm 0.27$ |
| FACILE-SupCon | $91.16 \pm 0.24$ | $90.48 \pm 0.24$ | $90.40 \pm 0.24$ | $91.39 \pm 0.24$ | $\mathbf{91.03 \pm 0.22}$ |
| FACILE-FSP | $\mathbf{91.77 \pm 0.21}$ | $\mathbf{90.85 \pm 0.23}$ | $\mathbf{90.77 \pm 0.24}$ | $\mathbf{92.19 \pm 0.22}$ | $90.02 \pm 0.24$ |
| pretraining method | NC | LR | RC | LR+LA | RC+LA |
| | | | 1-shot 3-way test on PAIP dataset | | |
| ImageNet (FSP) | $48.44 \pm 0.65$ | $50.34 \pm 0.65$ | $50.21 \pm 0.62$ | $48.90 \pm 0.62$ | $47.51 \pm 0.59$ |
| SimSiam | $49.42 \pm 0.65$ | $50.25 \pm 0.65$ | $49.76 \pm 0.65$ | $49.51 \pm 0.62$ | $49.09 \pm 0.63$ |
| SimCLR | $47.39 \pm 0.59$ | $48.35 \pm 0.59$ | $47.97 \pm 0.58$ | $47.77 \pm 0.59$ | $47.65 \pm 0.60$ |
| FSP-Patch | $\mathbf{51.61 \pm 0.68}$ | $\mathbf{51.61 \pm 0.67}$ | $\mathbf{52.06 \pm 0.67}$ | $\mathbf{51.74 \pm 0.66}$ | $\mathbf{51.38 \pm 0.66}$ |
| FACILE-SupCon | $49.65 \pm 0.61$ | $51.32 \pm 0.66$ | $51.16 \pm 0.63$ | $50.00 \pm 0.62$ | $50.81 \pm 0.65$ |
| FACILE-FSP | $48.91 \pm 0.61$ | $49.57 \pm 0.63$ | $49.68 \pm 0.63$ | $49.42 \pm 0.65$ | $48.60 \pm 0.64$ |
| | | | 5-shot 3-way test on PAIP dataset | | |
| ImageNet (FSP) | $62.46 \pm 0.52$ | $62.48 \pm 0.48$ | $63.14 \pm 0.50$ | $62.11 \pm 0.51$ | $60.52 \pm 0.49$ |
| SimSiam | $63.05 \pm 0.52$ | $64.44 \pm 0.49$ | $64.66 \pm 0.50$ | $65.44 \pm 0.53$ | $64.64 \pm 0.55$ |
| SimCLR | $61.48 \pm 0.52$ | $61.84 \pm 0.53$ | $62.75 \pm 0.51$ | $63.03 \pm 0.52$ | $61.70 \pm 0.52$ |
| FSP-Patch | $65.29 \pm 0.49$ | $65.81 \pm 0.51$ | $65.98 \pm 0.48$ | $65.70 \pm 0.50$ | $64.01 \pm 0.52$ |
| FACILE-SupCon | $\mathbf{65.44 \pm 0.51}$ | $\mathbf{66.75 \pm 0.52}$ | $\mathbf{67.11 \pm 0.51}$ | $67.24 \pm 0.53$ | $\mathbf{67.06 \pm 0.52}$ |
| FACILE-FSP | $64.68 \pm 0.53$ | $65.75 \pm 0.49$ | $66.58 \pm 0.51$ | $\mathbf{67.42 \pm 0.53}$ | $\mathbf{67.06 \pm 0.53}$ |
| pretraining method | NC | LR | RC | LR+LA | RC+LA |
| | | | 1-shot 9-way test on NCT dataset | | |
| ImageNet (FSP) | $58.75 \pm 0.35$ | $58.66 \pm 0.36$ | $58.48 \pm 0.34$ | $58.83 \pm 0.36$ | $57.32 \pm 0.36$ |
| SimSiam | $64.76 \pm 0.40$ | $66.09 \pm 0.39$ | $66.09 \pm 0.39$ | $66.54 \pm 0.40$ | $67.05 \pm 0.41$ |
| SimCLR | $60.47 \pm 0.41$ | $61.17 \pm 0.38$ | $61.43 \pm 0.39$ | $61.65 \pm 0.40$ | $62.48 \pm 0.38$ |
| FSP-Patch | $61.03 \pm 0.42$ | $63.53 \pm 0.40$ | $63.26 \pm 0.42$ | $62.75 \pm 0.43$ | $61.57 \pm 0.42$ |
| FACILE-SupCon | $\mathbf{68.99 \pm 0.45}$ | $\mathbf{70.76 \pm 0.40}$ | $\mathbf{70.89 \pm 0.41}$ | $\mathbf{70.45 \pm 0.45}$ | $70.63 \pm 0.44$ |
| FACILE-FSP | $67.43 \pm 0.44$ | $68.45 \pm 0.4$ | $68.97 \pm 0.42$ | $69.53 \pm 0.43$ | $\mathbf{70.89 \pm 0.42}$ |
| | | | 5-shot 9-way test on NCT dataset | | |
| ImageNet (FSP) | $74.82 \pm 0.26$ | $74.35 \pm 0.26$ | $75.20 \pm 0.26$ | $77.11 \pm 0.23$ | $74.89 \pm 0.26$ |
| SimSiam | $80.59 \pm 0.23$ | $80.51 \pm 0.23$ | $81.54 \pm 0.21$ | $83.68 \pm 0.22$ | $83.85 \pm 0.22$ |
| SimCLR | $77.30 \pm 0.25$ | $77.64 \pm 0.24$ | $79.17 \pm 0.24$ | $80.99 \pm 0.24$ | $81.71 \pm 0.23$ |
| FSP-Patch | $79.61 \pm 0.25$ | $79.89 \pm 0.24$ | $81.71 \pm 0.23$ | $82.92 \pm 0.24$ | $81.67 \pm 0.24$ |
| FACILE-SupCon | $\mathbf{86.89 \pm 0.22}$ | $\mathbf{88.06 \pm 0.20}$ | $\mathbf{89.26 \pm 0.19}$ | $\mathbf{89.62 \pm 0.19}$ | $\mathbf{88.67 \pm 0.21}$ |
| FACILE-FSP | $84.83 \pm 0.24$ | $85.78 \pm 0.23$ | $87.68 \pm 0.20$ | $88.16 \pm 0.20$ | $87.67 \pm 0.20$ |

Table 5: Models tested on LC, PAIP, and NCT dataset; average ACC and CI are reported.

### B.2.2 TEST WITH LARGE SHOT NUMBER

We further test the trained models with a larger shot number $k$. The result is shown in table 6

| pretraining method | NC | LR | RC | LR+LA | RC+LA |
|---|---|---|---|---|---|
| | | 10-shot 5-way on LC | | | |
| ImageNet (FSP) | $78.76 \pm 0.94$ | $78.92 \pm 0.92$ | $80.45 \pm 0.87$ | $82.25 \pm 0.83$ | $80.20 \pm 0.89$ |
| SimSiam | $88.52 \pm 0.55$ | $87.20 \pm 0.58$ | $87.73 \pm 0.56$ | $91.62 \pm 0.46$ | $91.88 \pm 0.47$ |
| SimCLR | $87.02 \pm 0.64$ | $86.26 \pm 0.64$ | $85.61 \pm 0.72$ | $90.28 \pm 0.52$ | $89.60 \pm 0.58$ |
| FSP-Patch | $88.41 \pm 0.53$ | $88.64 \pm 0.52$ | $89.15 \pm 0.51$ | $90.49 \pm 0.50$ | $89.88 \pm 0.54$ |
| FACILE-SupCon | $92.84 \pm 0.39$ | $92.87 \pm 0.38$ | $93.21 \pm 0.37$ | $94.25 \pm 0.36$ | $\mathbf{93.72 \pm 0.39}$ |
| FACILE-FSP | $\mathbf{93.10 \pm 0.39}$ | $\mathbf{93.11 \pm 0.38}$ | $\mathbf{93.63 \pm 0.37}$ | $\mathbf{94.52 \pm 0.35}$ | $93.07 \pm 0.45$ |
| | | 10-shot 3-way on PAIP | | | |
| ImageNet (FSP) | $65.36 \pm 0.91$ | $65.17 \pm 1.00$ | $65.40 \pm 0.99$ | $66.52 \pm 0.81$ | $64.45 \pm 0.81$ |
| SimSiam | $67.19 \pm 0.88$ | $67.35 \pm 0.98$ | $68.55 \pm 0.94$ | $70.88 \pm 0.77$ | $70.62 \pm 0.77$ |
| SimCLR | $65.77 \pm 0.85$ | $66.70 \pm 0.91$ | $67.01 \pm 0.91$ | $68.41 \pm 0.79$ | $66.96 \pm 0.82$ |
| FSP-Patch | $68.50 \pm 0.82$ | $69.12 \pm 0.85$ | $69.39 \pm 0.85$ | $70.13 \pm 0.75$ | $68.25 \pm 0.76$ |
| FACILE-SupCon | $\mathbf{70.03 \pm 0.81}$ | $\mathbf{71.24 \pm 0.84}$ | $72.17 \pm 0.83$ | $\mathbf{73.31 \pm 0.71}$ | $72.50 \pm 0.71$ |
| FACILE-FSP | $69.19 \pm 0.82$ | $71.13 \pm 0.82$ | $71.78 \pm 0.81$ | $73.22 \pm 0.73$ | $\mathbf{72.78 \pm 0.71}$ |
| | | 10-shot 9-way on NCT | | | |
| ImageNet (FSP) | $78.76 \pm 0.94$ | $78.92 \pm 0.92$ | $80.45 \pm 0.87$ | $82.25 \pm 0.83$ | $80.20 \pm 0.89$ |
| SimSiam | $82.92 \pm 0.91$ | $83.42 \pm 0.89$ | $84.76 \pm 0.81$ | $87.66 \pm 0.72$ | $88.12 \pm 0.69$ |
| SimCLR | $80.34 \pm 0.96$ | $81.67 \pm 0.90$ | $83.09 \pm 0.84$ | $85.96 \pm 0.76$ | $86.82 \pm 0.72$ |
| FSP-Patch | $83.36 \pm 0.77$ | $84.05 \pm 0.74$ | $85.93 \pm 0.65$ | $87.15 \pm 0.62$ | $86.05 \pm 0.63$ |
| FACILE-SupCon | $\mathbf{89.57 \pm 0.49}$ | $\mathbf{91.11 \pm 0.45}$ | $\mathbf{92.20 \pm 0.41}$ | $\mathbf{92.88 \pm 0.39}$ | $\mathbf{92.02 \pm 0.41}$ |
| FACILE-FSP | $87.54 \pm 0.61$ | $89.25 \pm 0.56$ | $90.77 \pm 0.49$ | $91.63 \pm 0.48$ | $91.23 \pm 0.50$ |

Table 6: Test result on LC, PAIP, and NCT dataset with shot number 10; average F1 and CI are reported.

### B.3 FINE-TUNE VIT-B/14 OF DINO V2 ON TCGA DATASET

Similar to section B.1, we fine-tune a fully-connected layer that is appended after DINO V2 Oquab et al. (2023) ViT-B/14. This methodology is applied across various models to assess their performance on histopathology image datasets. By adopting the DINO V2 architecture, known for its robustness and effectiveness in visual representation learning, we aim to harness its potential for the specialized domain of histopathology. We refer interested readers to section A.4 for details of pretraining.

Notably, our methods, FACILE-SupCon and FACILE-FSP, demonstrated markedly superior results in comparison to other baseline models when applied to histopathology image datasets as shown in table 7. This outcome highlights the effectiveness of these methods in leveraging coarse-grained labels specific to histopathology, thereby greatly enhancing the model performance of downstream tasks. Another critical insight emerged from our research: the current foundation model, DINO V2, exhibits limitations in its generalization performance on histopathology images. This suggests that while DINO V2 provides a strong starting point due to its robust visual representation capabilities, there is a clear need for further finetuning or prompt learning to optimize its performance for the unique challenges presented by histopathology datasets. This finding underscores the importance of specialized adaptation in the application of foundation models to specific domains like medical imaging.

### B.4 BENEFITS OF PRETRAINING ON LARGE PATHOLOGY DATASETS

In order to demonstrate the advantages of pretraining on large pathology datasets, we compare the performance of models pretrained on TCGA datasets with those pretrained on NCT datast, which are also studied in Yang et al. (2022).

The SimSiam model is trained for 100 epochs. SGD optimizer is used with learning rate of 0.01, weight decay of 0.0001, momentum of 0.9, and cosine learning rate decay. The batch size is 55.

For MoCo v3, similar to (Chen et al., 2021c; Yang et al., 2022), LARS optimizer (You et al., 2017) was used with an initial learning rate of 0.3, weight decay of $1.5e-6$, the momentum of 0.9, and cosine decay schedule. MoCo v3 was trained with a batch size of 256 for 200 epochs.

The FSP model with simple augmentation follows the setting of Yang et al. (2022). SGD optimizer with learning rate of 0.5, momentum of 0.9 and weight decay of 0 are used. A large batch size is used 512. The model is trained for 100 epochs with "step decay" schedule. The learning rate

| pretraining method | NC | LR | RC | LR+LA | RC+LA |
|---|---|---|---|---|---|
| 1-shot 5-way test on LC dataset | | | | | |
| DINO V2 (ViT-B/14) | $44.82 \pm 1.41$ | $47.51 \pm 1.39$ | $47.63 \pm 1.38$ | $47.36 \pm 1.39$ | $48.88 \pm 1.44$ |
| SimSiam | $48.79 \pm 1.37$ | $49.43 \pm 1.35$ | $48.43 \pm 1.36$ | $49.38 \pm 1.34$ | $49.50 \pm 1.34$ |
| SimCLR | $50.47 \pm 1.31$ | $50.52 \pm 1.33$ | $50.44 \pm 1.32$ | $51.66 \pm 1.32$ | $51.78 \pm 1.38$ |
| FSP-Patch | $49.73 \pm 1.41$ | $53.59 \pm 1.38$ | $53.07 \pm 1.41$ | $51.79 \pm 1.40$ | $51.27 \pm 1.43$ |
| FACILE-SupCon | $\mathbf{56.24 \pm 1.43}$ | $\mathbf{56.51 \pm 1.41}$ | $\mathbf{55.95 \pm 1.42}$ | $\mathbf{56.29 \pm 1.43}$ | $54.07 \pm 1.44$ |
| FACILE-FSP | $55.67 \pm 1.40$ | $56.26 \pm 1.36$ | $55.83 \pm 1.35$ | $56.01 \pm 1.38$ | $\mathbf{55.35 \pm 1.40}$ |
| 5-shot 5-way test on LC dataset | | | | | |
| DINO V2 (ViT-B/14) | $66.12 \pm 0.98$ | $64.71 \pm 1.12$ | $66.36 \pm 1.10$ | $72.95 \pm 0.93$ | $75.11 \pm 0.91$ |
| SimSiam | $67.51 \pm 0.96$ | $64.99 \pm 1.05$ | $65.39 \pm 1.05$ | $70.30 \pm 0.93$ | $71.19 \pm 0.93$ |
| SimCLR | $70.10 \pm 0.92$ | $69.28 \pm 0.96$ | $69.18 \pm 0.97$ | $72.99 \pm 0.92$ | $72.91 \pm 0.94$ |
| FSP-Patch | $71.97 \pm 0.96$ | $71.11 \pm 1.04$ | $71.19 \pm 1.03$ | $73.96 \pm 0.94$ | $73.20 \pm 0.96$ |
| FACILE-SupCon | $75.58 \pm 0.88$ | $74.26 \pm 0.94$ | $73.20 \pm 0.95$ | $75.81 \pm 0.90$ | $74.34 \pm 0.96$ |
| FACILE-FSP | $\mathbf{75.86 \pm 0.86}$ | $\mathbf{74.64 \pm 0.89}$ | $\mathbf{74.12 \pm 0.93}$ | $\mathbf{76.17 \pm 0.88}$ | $\mathbf{75.08 \pm 0.95}$ |
| 1-shot 3-way test on PAIP dataset | | | | | |
| DINO V2 (ViT-B/14) | $41.51 \pm 1.27$ | $44.37 \pm 1.26$ | $44.28 \pm 1.25$ | $42.43 \pm 1.27$ | $42.78 \pm 1.27$ |
| SimSiam | $49.42 \pm 1.28$ | $48.07 \pm 1.35$ | $48.44 \pm 1.36$ | $48.76 \pm 1.33$ | $46.48 \pm 1.37$ |
| SimCLR | $48.60 \pm 1.19$ | $48.76 \pm 1.25$ | $47.98 \pm 1.26$ | $48.94 \pm 1.23$ | $47.20 \pm 1.26$ |
| FSP-Patch | $46.09 \pm 1.17$ | $47.44 \pm 1.18$ | $48.09 \pm 1.19$ | $46.76 \pm 1.18$ | $43.68 \pm 1.22$ |
| FACILE-SupCon | $\mathbf{51.97 \pm 1.18}$ | $\mathbf{52.25 \pm 1.22}$ | $\mathbf{51.80 \pm 1.22}$ | $51.36 \pm 1.22$ | $\mathbf{50.24 \pm 1.23}$ |
| FACILE-FSP | $51.34 \pm 1.16$ | $51.18 \pm 1.19$ | $51.51 \pm 1.19$ | $\mathbf{51.50 \pm 1.16}$ | $49.77 \pm 1.22$ |
| 5-shot 3-way test on PAIP dataset | | | | | |
| DINO V2 (ViT-B/14) | $57.59 \pm 1.07$ | $58.19 \pm 1.10$ | $59.37 \pm 1.07$ | $61.84 \pm 0.85$ | $60.81 \pm 0.86$ |
| SimSiam | $61.56 \pm 0.97$ | $62.52 \pm 1.01$ | $62.81 \pm 1.01$ | $64.40 \pm 0.86$ | $62.44 \pm 0.93$ |
| SimCLR | $62.20 \pm 0.93$ | $61.78 \pm 0.99$ | $63.20 \pm 0.97$ | $63.38 \pm 0.86$ | $63.03 \pm 0.88$ |
| FSP-Patch | $63.77 \pm 0.88$ | $63.85 \pm 0.94$ | $63.85 \pm 0.93$ | $63.61 \pm 0.85$ | $60.91 \pm 0.87$ |
| FACILE-SupCon | $\mathbf{67.16 \pm 0.84}$ | $67.29 \pm 0.89$ | $66.88 \pm 0.90$ | $\mathbf{67.61 \pm 0.85}$ | $\mathbf{66.34 \pm 0.84}$ |
| FACILE-FSP | $67.14 \pm 0.85$ | $\mathbf{67.67 \pm 0.84}$ | $\mathbf{67.54 \pm 0.86}$ | $67.12 \pm 0.81$ | $66.05 \pm 0.83$ |
| 1-shot 9-way test on NCT dataset | | | | | |
| DINO V2 (ViT-B/14) | $56.03 \pm 1.62$ | $59.11 \pm 1.57$ | $60.13 \pm 1.55$ | $58.71 \pm 1.57$ | $59.06 \pm 1.55$ |
| SimSiam | $62.60 \pm 1.45$ | $61.89 \pm 1.50$ | $61.90 \pm 1.51$ | $62.27 \pm 1.47$ | $61.05 \pm 1.44$ |
| SimCLR | $65.43 \pm 1.43$ | $64.18 \pm 1.44$ | $64.15 \pm 1.46$ | $64.83 \pm 1.43$ | $62.69 \pm 1.38$ |
| FSP-Patch | $65.22 \pm 1.49$ | $65.93 \pm 1.41$ | $65.94 \pm 1.40$ | $65.26 \pm 1.45$ | $62.66 \pm 1.46$ |
| FACILE-SupCon | $71.55 \pm 1.36$ | $70.36 \pm 1.37$ | $70.52 \pm 1.35$ | $71.05 \pm 1.35$ | $\mathbf{68.85 \pm 1.40}$ |
| FACILE-FSP | $\mathbf{72.05 \pm 1.34}$ | $\mathbf{70.70 \pm 1.35}$ | $\mathbf{70.77 \pm 1.34}$ | $\mathbf{71.14 \pm 1.34}$ | $68.03 \pm 1.40$ |
| 5-shot 9-way test on NCT dataset | | | | | |
| DINO V2 (ViT-B/14) | $76.85 \pm 0.98$ | $76.51 \pm 1.02$ | $78.67 \pm 0.94$ | $82.20 \pm 0.82$ | $82.75 \pm 0.83$ |
| SimSiam | $80.81 \pm 0.85$ | $80.06 \pm 0.87$ | $81.55 \pm 0.85$ | $83.18 \pm 0.80$ | $82.39 \pm 0.83$ |
| SimCLR | $82.87 \pm 0.80$ | $81.91 \pm 0.82$ | $82.86 \pm 0.80$ | $83.92 \pm 0.77$ | $82.89 \pm 0.79$ |
| FSP-Patch | $83.63 \pm 0.83$ | $83.49 \pm 0.80$ | $84.34 \pm 0.78$ | $85.32 \pm 0.75$ | $83.03 \pm 0.79$ |
| FACILE-SupCon | $87.74 \pm 0.64$ | $87.00 \pm 0.64$ | $87.38 \pm 0.62$ | $87.82 \pm 0.63$ | $86.15 \pm 0.69$ |
| FACILE-FSP | $\mathbf{87.93 \pm 0.65}$ | $\mathbf{87.52 \pm 0.65}$ | $\mathbf{87.72 \pm 0.62}$ | $\mathbf{88.01 \pm 0.64}$ | $\mathbf{86.46 \pm 0.70}$ |

Table 7: Test result on LC, PAIP, and NCT dataset with ViT-B/14 from DINO V2; average F1 and CI are reported.

multiplied by 0.1 at 30, 60, and 90 epochs respectively. The FSP model with strong augmentation was trained for 50 epochs. The batch size is set to 64. The SGD is used with a learning rate of 0.03, momentum of 0.9, weight decay of 0.0001, and the cosine schedule. The model is trained for 50 epochs.

The SupCon model is trained with trained for 100 epochs. The batch size is set to 64. The SGD optimizer is used with a learning rate of 0.01, momentum of 0.9, weight decay of 0.0001, and the cosine schedule.

table 8 shows the performance of the pretrained models on the LC and PAIP dataset with shot numbers 1 or 5. Notably, the best-performing models on the two test datasets exhibit a significant performance gap compared to the best models pretrained on TCGA datasets as depicted in table 2.

## B.5   PRETRAIN ON TCGA AND GTEX WITH PATCH SIZE 1,000X1,000

We train the models using TCGA patches of size $1,000 \times 1,000$, which are extracted from 20X magnification and resized to $224 \times 224$. Subsequently, the pretrained models are evaluated on PDAC datasets, and the corresponding test performance is presented in Figure 9. Notably, for shot number of 1 and 5, our model significantly outperforms other models, demonstrating a substantial performance margin.

| pretraining method | NC | LR | RC | LR+LA | RC+LA |
|---|---|---|---|---|---|
| 1-shot 5-way test on LC dataset | | | | | |
| SimSiam | $59.30 \pm 1.31$ | $58.67 \pm 1.41$ | $58.58 \pm 1.40$ | $59.66 \pm 1.35$ | $59.85 \pm 1.35$ |
| MoCo v3 ((Yang et al., 2022)) | $59.38 \pm 1.62$ | $59.39 \pm 1.68$ | $59.46 \pm 1.68$ | $60.15 \pm 1.59$ | $60.54 \pm 1.58$ |
| FSP (simple aug; (Yang et al., 2022)) | $51.42 \pm 1.59$ | $46.06 \pm 1.88$ | $46.33 \pm 1.86$ | $50.53 \pm 1.65$ | $51.00 \pm 1.65$ |
| FSP (strong aug) | $\mathbf{68.00 \pm 1.29}$ | $\mathbf{66.17 \pm 1.41}$ | $\mathbf{66.18 \pm 1.46}$ | $\mathbf{68.39 \pm 1.34}$ | $\mathbf{68.02 \pm 1.40}$ |
| SupCon | $64.48 \pm 1.33$ | $63.52 \pm 1.42$ | $63.84 \pm 1.40$ | $65.43 \pm 1.33$ | $65.98 \pm 1.38$ |
| 5-shot 5-way test on LC dataset | | | | | |
| SimSiam | $76.21 \pm 0.87$ | $74.05 \pm 1.10$ | $74.59 \pm 1.10$ | $77.87 \pm 0.87$ | $76.03 \pm 0.94$ |
| MoCo v3 ((Yang et al., 2022)) | $72.82 \pm 1.25$ | $70.29 \pm 1.43$ | $71.31 \pm 1.40$ | $78.72 \pm 1.00$ | $79.71 \pm 0.95$ |
| FSP (simple aug; (Yang et al., 2022)) | $56.44 \pm 1.50$ | $52.27 \pm 1.81$ | $55.62 \pm 1.74$ | $63.47 \pm 1.37$ | $63.47 \pm 1.46$ |
| FSP (strong aug) | $\mathbf{83.53 \pm 0.79}$ | $\mathbf{80.81 \pm 1.01}$ | $\mathbf{80.27 \pm 1.08}$ | $\mathbf{85.57 \pm 0.77}$ | $\mathbf{84.06 \pm 0.89}$ |
| SupCon | $81.51 \pm 0.85$ | $78.77 \pm 1.03$ | $78.65 \pm 1.08$ | $83.51 \pm 0.84$ | $83.31 \pm 0.91$ |
| 1-shot 3-way test on PAIP dataset | | | | | |
| SimSiam | $37.13 \pm 1.14$ | $38.26 \pm 1.13$ | $37.93 \pm 1.15$ | $38.00 \pm 1.12$ | $38.67 \pm 1.12$ |
| MoCo v3 ((Yang et al., 2022)) | $43.17 \pm 1.26$ | $42.48 \pm 1.30$ | $43.02 \pm 1.31$ | $43.55 \pm 1.28$ | $44.57 \pm 1.28$ |
| FSP (simple aug; (Yang et al., 2022)) | $37.15 \pm 1.07$ | $36.69 \pm 1.13$ | $37.39 \pm 1.08$ | $37.40 \pm 1.07$ | $35.28 \pm 1.09$ |
| FSP (strong aug) | $47.67 \pm 1.18$ | $48.44 \pm 1.19$ | $48.16 \pm 1.21$ | $48.27 \pm 1.17$ | $\mathbf{49.38 \pm 1.19}$ |
| SupCon | $\mathbf{48.45 \pm 1.19}$ | $\mathbf{49.29 \pm 1.20}$ | $\mathbf{48.97 \pm 1.22}$ | $\mathbf{49.47 \pm 1.20}$ | $48.53 \pm 1.20$ |
| 5-shot 3-way test on PAIP dataset | | | | | |
| SimSiam | $47.52 \pm 1.00$ | $48.12 \pm 1.10$ | $47.04 \pm 1.11$ | $52.70 \pm 0.95$ | $54.51 \pm 1.00$ |
| MoCo v3 ((Yang et al., 2022)) | $55.43 \pm 1.00$ | $54.23 \pm 1.09$ | $54.05 \pm 1.09$ | $56.07 \pm 0.92$ | $55.73 \pm 0.93$ |
| FSP (simple aug; (Yang et al., 2022)) | $44.98 \pm 0.95$ | $45.13 \pm 0.96$ | $45.30 \pm 0.96$ | $44.34 \pm 0.87$ | $44.03 \pm 0.88$ |
| FSP (strong aug) | $62.00 \pm 0.88$ | $62.48 \pm 0.97$ | $62.04 \pm 0.98$ | $\mathbf{64.82 \pm 0.86}$ | $\mathbf{64.60 \pm 0.87}$ |
| SupCon | $\mathbf{63.62 \pm 0.91}$ | $\mathbf{64.38 \pm 0.96}$ | $\mathbf{63.61 \pm 1.00}$ | $64.37 \pm 0.87$ | $64.28 \pm 0.88$ |

Table 8: pretraining on NCT dataset and testing on LC and PAIP dataset; average F1 and CI are reported.

| pretraining method | NC | LR | RC | LR+LA | RC+LA |
|---|---|---|---|---|---|
| 1-shot 5-way test | | | | | |
| ImageNet (FSP) | $29.57 \pm 1.07$ | $31.32 \pm 1.09$ | $31.16 \pm 1.07$ | $30.88 \pm 1.08$ | $30.14 \pm 1.08$ |
| SimSiam | $30.48 \pm 1.08$ | $30.18 \pm 1.12$ | $30.19 \pm 1.13$ | $30.41 \pm 1.08$ | $31.13 \pm 1.10$ |
| SimCLR | $30.79 \pm 1.08$ | $30.93 \pm 1.13$ | $30.78 \pm 1.12$ | $31.33 \pm 1.08$ | $31.22 \pm 1.07$ |
| FSP-Patch | $34.04 \pm 1.16$ | $33.99 \pm 1.20$ | $33.69 \pm 1.20$ | $34.29 \pm 1.15$ | $34.99 \pm 1.16$ |
| FACILE-SupCon | $35.44 \pm 1.17$ | $34.94 \pm 1.20$ | $34.58 \pm 1.22$ | $35.68 \pm 1.16$ | $35.27 \pm 1.17$ |
| FACILE-FSP | $\mathbf{37.36 \pm 1.16}$ | $\mathbf{36.07 \pm 1.23}$ | $\mathbf{36.93 \pm 1.21}$ | $\mathbf{36.79 \pm 1.19}$ | $\mathbf{36.81 \pm 1.18}$ |
| 5-shot 5-way test | | | | | |
| ImageNet (FSP) | $41.83 \pm 0.96$ | $41.30 \pm 1.10$ | $41.08 \pm 1.08$ | $42.38 \pm 0.94$ | $41.29 \pm 0.93$ |
| SimSiam | $40.15 \pm 1.03$ | $37.29 \pm 1.21$ | $37.43 \pm 1.21$ | $41.87 \pm 1.00$ | $42.70 \pm 1.01$ |
| SimCLR | $40.30 \pm 1.04$ | $38.74 \pm 1.19$ | $39.02 \pm 1.16$ | $40.98 \pm 0.96$ | $40.90 \pm 0.98$ |
| FSP-Patch | $44.26 \pm 1.10$ | $42.99 \pm 1.20$ | $43.69 \pm 1.12$ | $46.32 \pm 0.97$ | $46.69 \pm 0.96$ |
| FACILE-SupCon | $45.83 \pm 1.09$ | $45.07 \pm 1.18$ | $45.93 \pm 1.13$ | $47.72 \pm 0.95$ | $47.00 \pm 0.95$ |
| FACILE-FSP | $\mathbf{48.21 \pm 1.04}$ | $\mathbf{47.62 \pm 1.12}$ | $\mathbf{47.94 \pm 1.08}$ | $\mathbf{48.84 \pm 0.95}$ | $\mathbf{48.37 \pm 0.95}$ |

Table 9: Models pretrained on TCGA and tested on PDAC dataset; average F1 and CI are reported.

Similarly, we train the models using GTEx patches with dimensions of $1,000 \times 1,000$. The patches are extracted from 20X magnification and resized to $224 \times 224$. The pretrained models are tested on PDAC datasets, revealing similar outcomes, as illustrated in Table 10.

| pretraining method | NC | LR | RC | LR+LA | RC+LA |
|---|---|---|---|---|---|
| 1-shot 5-way test | | | | | |
| SimSiam | $34.78 \pm 1.18$ | $34.57 \pm 1.25$ | $35.30 \pm 1.25$ | $35.13 \pm 1.19$ | $35.27 \pm 1.19$ |
| SimCLR | $33.68 \pm 1.14$ | $33.74 \pm 1.18$ | $33.69 \pm 1.17$ | $34.28 \pm 1.14$ | $33.84 \pm 1.12$ |
| FSP-Patch | $31.87 \pm 1.09$ | $32.90 \pm 1.13$ | $32.53 \pm 1.11$ | $32.55 \pm 1.09$ | $32.10 \pm 1.07$ |
| FACILE-SupCon | $34.36 \pm 1.06$ | $34.35 \pm 1.13$ | $34.39 \pm 1.14$ | $34.70 \pm 1.07$ | $34.35 \pm 1.07$ |
| FACILE-FSP | $\mathbf{35.62 \pm 1.10}$ | $\mathbf{35.51 \pm 1.15}$ | $\mathbf{35.40 \pm 1.13}$ | $\mathbf{35.87 \pm 1.10}$ | $\mathbf{36.16 \pm 1.09}$ |
| 5-shot 5-way test | | | | | |
| SimSiam | $46.00 \pm 1.10$ | $43.26 \pm 1.30$ | $44.19 \pm 1.26$ | $47.24 \pm 1.00$ | $\mathbf{47.85 \pm 1.00}$ |
| SimCLR | $44.44 \pm 1.08$ | $43.40 \pm 1.19$ | $43.58 \pm 1.15$ | $44.60 \pm 0.98$ | $44.17 \pm 0.96$ |
| FSP-Patch | $42.09 \pm 0.99$ | $40.15 \pm 1.15$ | $40.69 \pm 1.09$ | $42.71 \pm 0.92$ | $42.66 \pm 0.90$ |
| FACILE-SupCon | $44.85 \pm 1.02$ | $43.65 \pm 1.15$ | $44.01 \pm 1.13$ | $46.37 \pm 0.93$ | $45.10 \pm 0.92$ |
| FACILE-FSP | $\mathbf{46.91 \pm 0.97}$ | $\mathbf{46.32 \pm 1.07}$ | $\mathbf{47.10 \pm 1.02}$ | $\mathbf{48.01 \pm 0.90}$ | $47.70 \pm 0.89$ |

Table 10: Models pretrained on GTEx and tested on PDAC dataset; average F1 and CI are reported.

## C    DATASETS

### C.1    GTEX DATASET

The Genotype-Tissue Expression (GTEx) project is a pioneering initiative aimed at constructing an extensive public repository to investigate tissue-specific gene expression and regulation. The GTEx project collected samples from 54 non-diseased tissue sites across nearly 1000 individuals, with an emphasis on molecular assays such as Whole Genome Sequencing (WGS), Whole Exome Sequencing (WES), and RNA-sequencing. Additionally, the GTEx Biobank contains a plethora of unutilized samples. The GTEx portal (`https://gtexportal.org/home/`) provides unrestricted access to a plethora of data, including gene expression levels, quantitative trait loci (QTLs), and histology images, to aid the research community in advancing our understanding of human gene expression and its regulation.

We downloaded all the slides from the GTEx portal. The organs from which the slides are extracted are used for coarse-grained labels. We extract all the non-overlapping patches with size $1,000 \times 1,000$ and only keep those with intensity in $[0.1, 0.85]$ to filter out backgrounds.

The number of slides from each organ for GTEx can be found in figure 6. Thumbnails of WSI examples from the GTEx dataset can be found in figure 7.

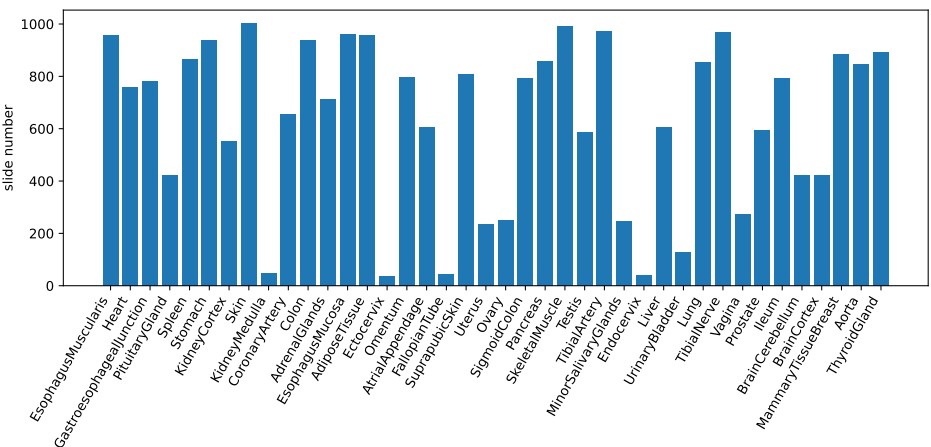

Figure 6: Slide number for each organ in GTEx

### C.2    TCGA DATASET

The Cancer Genome Atlas (TCGA; `https://www.cancer.gov/ccg/research/genome-sequencing/tcga`) is a project that aims to comprehensively characterize genetic mutations responsible for cancer using genome sequencing and bioinformatics. The TCGA dataset consists of 10,825 patient samples, including gene expression, DNA methylation, copy number variation, and mutation data, histopathology data, among others (source sites: Duke University Medical School McLendon Roger 1 Friedman Allan 2 Bigner Darrell 1 et al., 2008; 13 et al., 2012). This large-scale dataset has enabled researchers to identify numerous genomic alterations associated with cancer and has contributed to the development of new diagnostic and therapeutic approaches.

We downloaded all the diagnostic slides from GDC portal `https://portal.gdc.cancer.gov/`. The project names of the slides are used for coarse-grained labels. We extract patches at two different scales, i.e., $224 \times 224$ and $1,000 \times 1,000$ at 20X magnification, from all the slides.

The number of slides from each project for TCGA can be found in figure 8. Thumbnails of WSI examples from TCGA dataset can be found in figure 9.

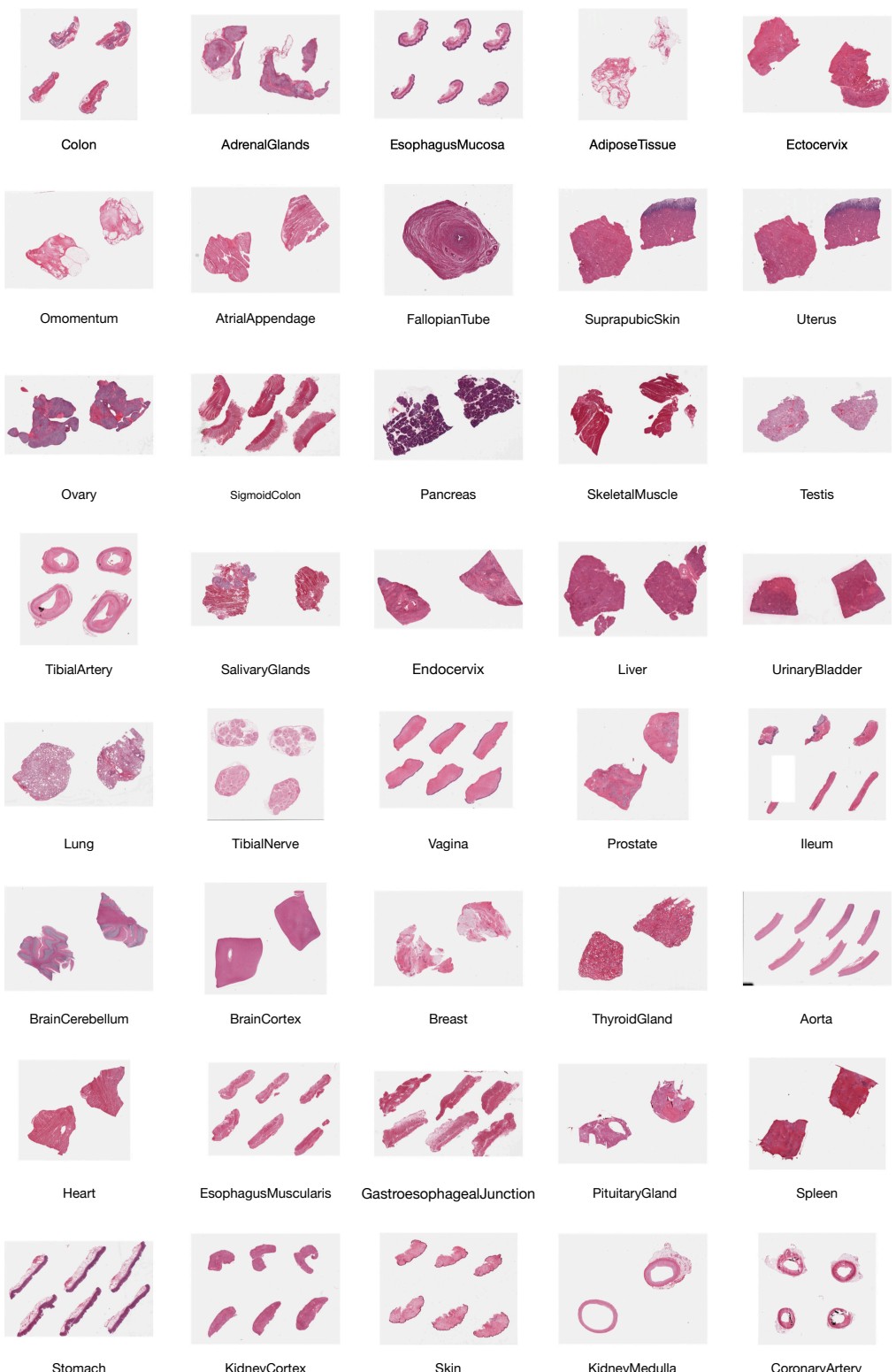

Figure 7: Randomly deleted examples from GTEx dataset

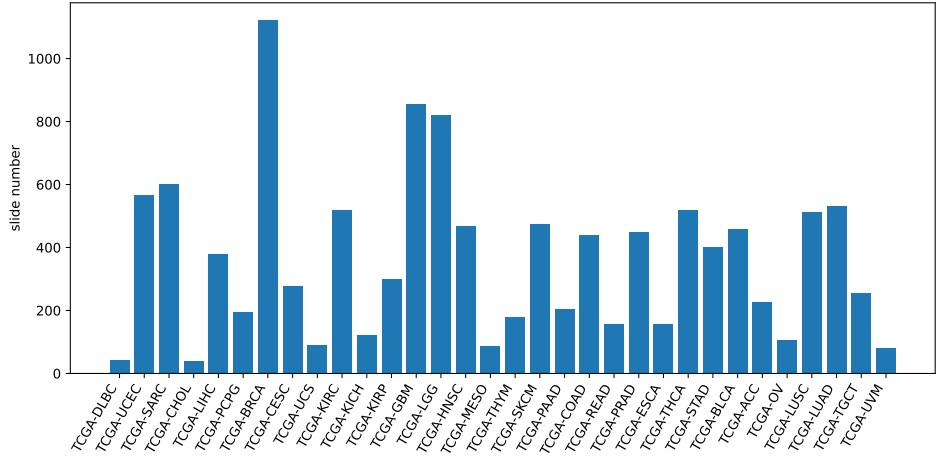

Figure 8: Slide number for each tumor in TCGA

## C.3 PDAC DATASET

To address the presence of multiple tissues within certain patches, we employ a labeling strategy that involves identifying and labeling the centered tissues within these patches. To ensure annotation accuracy, each patch undergoes labeling by a minimum of two pathologists, thereby maintaining the quality of the annotations. For the specific patch numbers corresponding to each tissue in the PDAC dataset, please refer to Figure 10. Furthermore, examples of patches from the PDAC dataset are provided in Figure 11, offering visual illustrations of the dataset.

| Coarse-Grained Dataset | Data type and annotation | WSI number | Extracted patch number |
|---|---|---|---|
| GTEx | slides; organs | 25,501 | 9,465,689 |
| TCGA | slides; tumors | 11,638 | 10,321,273 (11,588,226 w/ size 224) |
| Fine-Grained Dataset | Data type and annotation | WSI number | Extracted patch number |
| PDAC | patches; tissues | 194 | 12,250 |
| LC25000 | patches; tissues | 1,250 | 25,000 |
| PAIP19 | patches; tissues | 60 | 75,000 |
| NCT-CRC-HE-100K | patches; tissues | 86 | 100,000 |

Table 11: Dataset statistics

In order to validate our model on a real-world dataset, we generated WSIs of Pancreatic Ductal Adenocarcinoma (PDAC)[1]. PDAC, a particularly aggressive and lethal form of cancer originating in the pancreatic duct cells, presents various subtypes, each with distinct morphological characteristics. These variations underscore the need for advanced automated tools to accurately characterize and differentiate between these subtypes, thereby aiding disease studies and potentially informing treatment strategies. Examples of PDAC and class distribution are detailed in section C. There are in total 12,250 annotated patches extracted from 194 slides. The patch size used for this analysis is $1,000 \times 1,000$ at a 20X magnification. Each patch was annotated into one of 5 classes (i.e., Stroma, Normal Acini, Normal Duct, Tumor, and Islet) and confirmed by at least two pathologists.

## C.4 NCT, PAIP, AND LC

We test our models on 4 datasets with fine-grained labels. These datasets are from diverse body sites. Statistics of these datasets can be found in table 11.

NCT-CRC-HE-100K (NCT) is collected from colon (Kather et al., 2018). It consists of 9 classes with 100K non-overlapping patches. The patch size is $224 \times 224$. LC25000 (LC) is collected from lung and colon sites (Borkowski et al., 2021). It has 5 classes and each class has 5,000 patches. The patch size is $768 \times 768$. We resize the patches to $224 \times 224$. PAIP19 (PAIP) is collected from liver site (Kim et al., 2021). There are in total 50 WSIs. The WSIs are cropped into patches with size

---
[1]We will make data publicly available upon acceptance of our paper

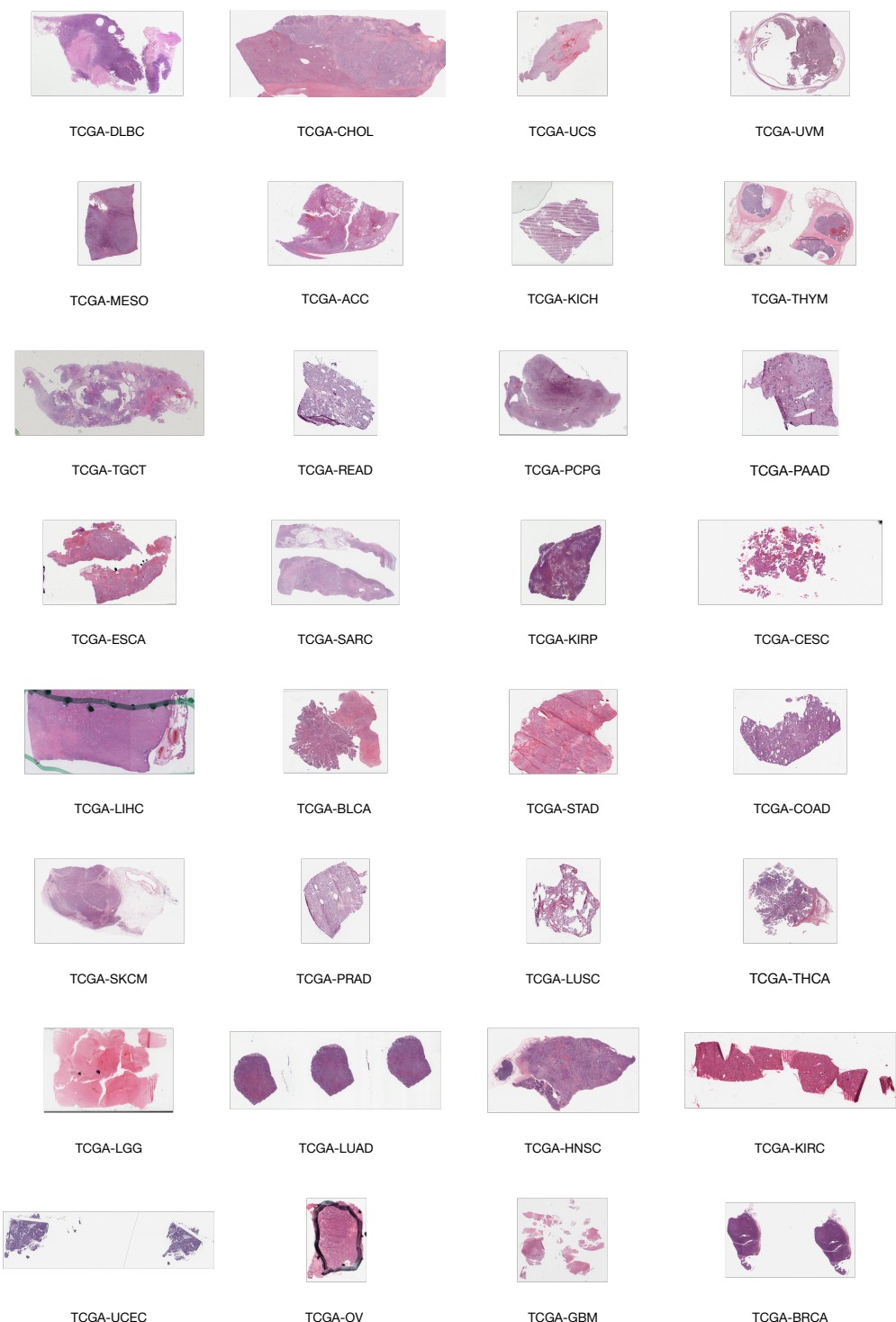

Figure 9: Randomly selected examples from TCGA dataset

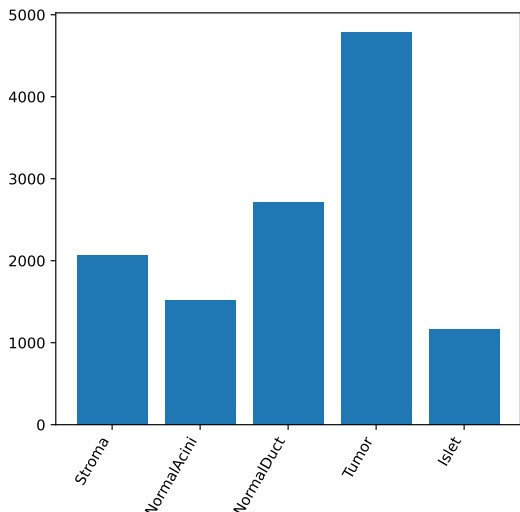

Figure 10: Patch number for each tissue for PDAC

$224 \times 224$. We only keep those patches with masks and assign labels with majority voting similar to Yang et al. (2022). We downsample these patches to 75K patches, with 25K in each class.

## D  DATA AUGMENTATION

Two data augmentation strategies are used in this paper.

**Simple augmentation**    Following Yang et al. (2022), we also used a simple augmentation policy which includes random resized cropping and horizontal flipping. In our paper, this simple augmentation policy is only used for FSP-Patch model pretraining on the NCT dataset.

**Strong augmentation**    Following previous work (Grill et al., 2020; Chen et al., 2021c; Yang et al., 2022), for SimCLR and SupCon models, we used similar strong data augmentation which contains random resized cropping, horizontal flipping, horizontal flipping, color jittering (Wu et al., 2018) with (brightness=0.8, contrast=0.8, saturation=0.8, hue=0.2, probability=0.8), grayscale conversion (Wu et al., 2018) with (probability=0.2), Gaussian blurring (Chen et al., 2020) with (kernel size=5, min=0.1, max=2.0, probability=0.5), and polarization (Grill et al., 2020) with (threshold=128, probability=0.2).

In implementing the SimSiam model, we adopted a comparable augmentation strategy, utilizing robust data augmentation techniques. Specifically, we fine-tuned parameters for color jittering, setting brightness, contrast, and saturation adjustments to 0.4, and hue to 0.1. These modifications were applied with a probability of 0.8, as informed by Chen & He (2021).

## E  LATENT AUGMENTATION

Latent augmentation (LA) was originally proposed in Yang et al. (2022) to improve the performance of the few-shot learning system in a simple unsupervised way. The pretrained feature extractor can only transfer parts of available knowledge in the pretraining datasets by the learned weights of the feature extractor. More transferable knowledge is inherent in the pretraining data representations.

In order to fully exploit the pretraining data, possible semantic shifts of clustered representations of the pretraining dataset are transferred to downstream tasks besides the pretrained feature extractor weights. The k-means clustering method is performed on the representations of pretraining datasets, which are generated by the pretrained feature extractor $\hat{e}$. Assume we obtain $C$ clusters after clustering. The base dictionary $\mathcal{B} = \{(c_i, \Sigma_i)\}_{i=1}^{C}$ is constructed, where $c_i$ is the $i$-th cluster prototype,

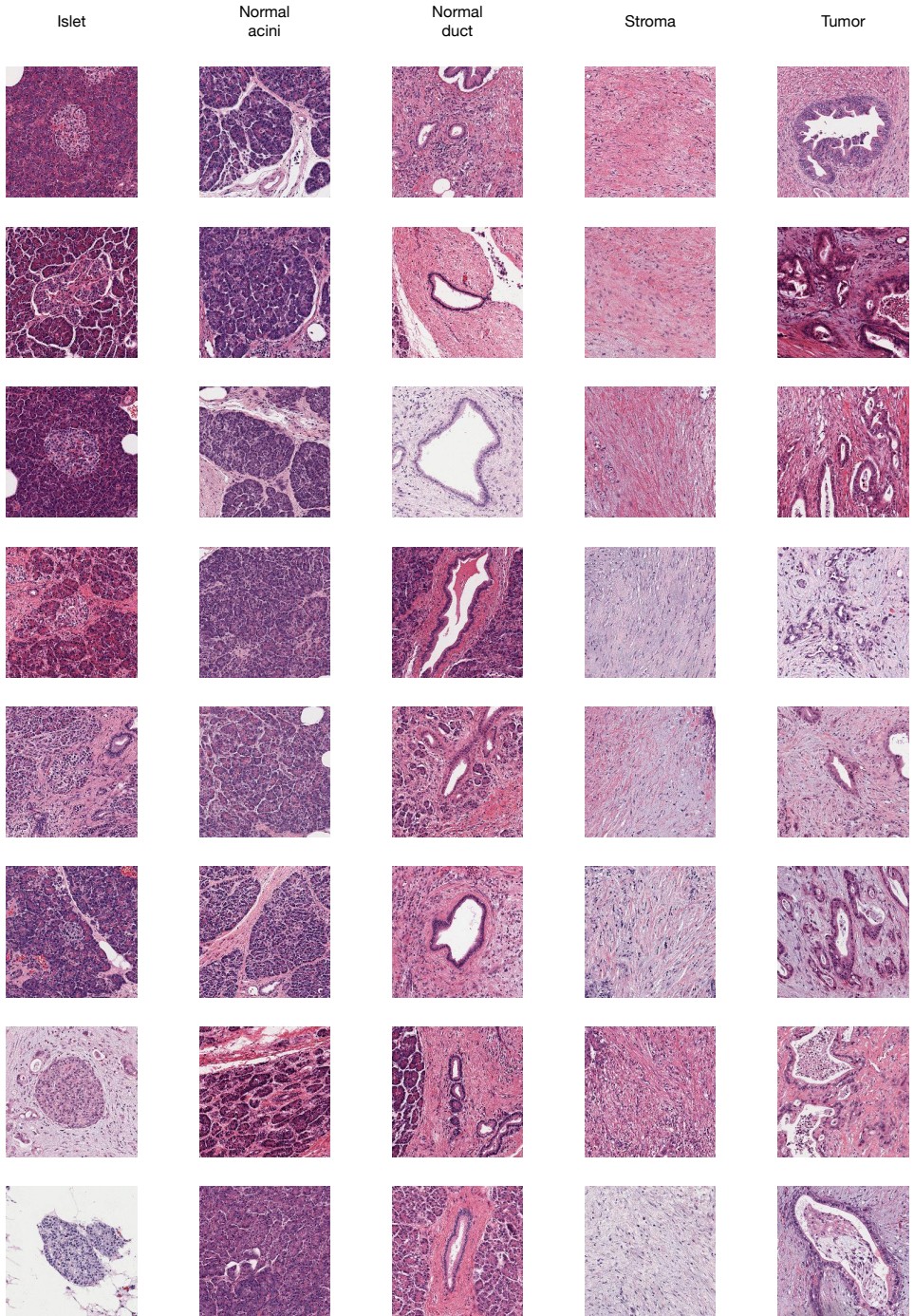

Figure 11: Randomly selected examples from each class of PDAC dataset.

i.e., mean representation of all samples in the cluster and $\Sigma_i$ is the covariance matrix of the cluster. During downstream task testing, LA uses the original representation $z$ to select the closest prototype from $\mathcal{B}$. We can get additive augmentation $\tilde{z} = z + \delta$, where $\delta$ is sampled from $\mathcal{N}(0, \Sigma_{i^*})$ and $i^*$ is the index of closest prototype of $z$. The classifier of the downstream tasks is then trained on both the original representations and the augmented representations.

# F  ABLATION STUDY

## F.1  SET-INPUT MODELS

Pooling architectures have been used in various set-input problems, e.g, 3D shape recognition (Shi et al., 2015; Su et al., 2015), learning the statistics of a set (Edwards & Storkey, 2016). Vinyals et al. (2015); Ilse et al. (2018) pool elements in a set by a weighted average with weights computed by the attention module. (Zaheer et al., 2017; Edwards & Storkey, 2016) proposed to aggregate embeddings of instances, extracted using a neural network, with pooling operations (e.g., mean, sum, max). This simple method satisfies the permutation invariant property and can work with any set size. Santoro et al. (2017) used a relational network to model all pairwise interactions of elements in a given set. Lee et al. (2019a) proposed to use the Transformer (Vaswani et al., 2017) to explicitly model higher-order interactions among the instances in a set.

We evaluate three set-input models for the FACILE-FSP model: attention-based MIL pooling (Ilse et al., 2018), Deep Set (Zaheer et al., 2017), and Set Transformer (Lee et al., 2019a). Attention-based MIL pooling uses a weighted average of instance embeddings from a set where weights are determined by a neural network. The attention-based MIL pooling corresponds to a version of attention (Lin et al., 2017; Raffel & Ellis, 2015). It has been adapted by Zhang et al. (2020b;a); Pal et al. (2021) in the context of H&E images. It uses a single fully connected layer and softmax with batch normalization and ReLU activation to predict the attention weights for instances. In the Deep Set model, each instance in a set is independently fed into a neural network that takes fixed-sized inputs. The extracted features are then aggregated using a pooling operation (i.e., mean, sum, or max). The final output is obtained by further non-linear operations. The simple architecture satisfies the permutation invariant property and can work with any set size. Set Transformer adapted the Transformer model for set data. It leverages the attention mechanism (Vaswani et al., 2017) to capture interactions between instances of the input set. It applies the idea of inducing points from the sparse Gaussian process literature to reduce quadratic complexity to linear in the size of the input set.

We train FACILE-FSP with three set-input models. The set size $a$ is set to 5. In the attention-based MIL pooling model, we implemented the simple version, and use the single fc layer with softmax to predict attention weights from ResNet18 extracted features. For Deep Set model, we use two fc layers with ReLU activation functions in between to extract instance features before set pooling. In the Set Transformer, we use 4 attention heads and 3 inducing points.

From table 12, we conclude that none of the 3 set-input models used in FACILE-FSP is consistently better than the other set-input models. The Deep Set model achieves the highest average F1 score with more tasks.

## F.2  LEARNING CURVE

To validate the adequacy of training for all models, we assess the intermediate checkpoints of each pretraining model on the LC dataset. The learning curves and confidence intervals (CI) of FACILE-FSP, FSP-Patch, and SimSiam are displayed in figure 12. Upon careful examination of the learning curves in figure 12, we observe conclusive evidence of complete training for all models, as they have reached convergence.

## F.3  INPUT SET SIZE

To examine the impact of input set size on downstream tasks, we conduct pretraining experiments using FACILE-FSP on the TCGA dataset with varying input set sizes. The resulting feature map $e$

| set-input model | NC | LR | RC | LR+LA | RC+LA |
|---|---|---|---|---|---|
| 1-shot 5-way test on LC dataset | | | | | |
| Attention-based MIL pooling | $70.53 \pm 1.32$ | $69.86 \pm 1.39$ | $69.75 \pm 1.37$ | $71.15 \pm 1.31$ | $70.31 \pm 1.34$ |
| Deep Set | $\mathbf{77.84 \pm 1.16}$ | $\mathbf{77.56 \pm 1.16}$ | $\mathbf{77.56 \pm 1.17}$ | $\mathbf{79.16 \pm 1.09}$ | $\mathbf{77.38 \pm 1.18}$ |
| Set Transformer | $75.09 \pm 1.30$ | $73.57 \pm 1.29$ | $73.16 \pm 1.33$ | $74.03 \pm 1.28$ | $72.88 \pm 1.34$ |
| 5-shot 5-way test on LC dataset | | | | | |
| Attention-based MIL pooling | $88.12 \pm 0.59$ | $81.60 \pm 1.04$ | $82.51 \pm 0.97$ | $89.18 \pm 0.57$ | $88.15 \pm 0.65$ |
| Deep Set | $90.35 \pm 0.50$ | $\mathbf{90.91 \pm 0.47}$ | $\mathbf{91.54 \pm 0.46}$ | $\mathbf{91.68 \pm 0.50}$ | $\mathbf{90.97 \pm 0.54}$ |
| Set Transformer | $\mathbf{90.67 \pm 0.54}$ | $89.18 \pm 0.61$ | $89.02 \pm 0.63$ | $90.03 \pm 0.59$ | $88.71 \pm 0.67$ |
| 1-shot 3-way test on PAIP dataset | | | | | |
| Attention-based MIL pooling | $50.98 \pm 1.37$ | $51.93 \pm 1.35$ | $51.91 \pm 1.36$ | $51.98 \pm 1.36$ | $52.39 \pm 1.35$ |
| Deep Set | $\mathbf{52.04 \pm 1.25}$ | $\mathbf{53.27 \pm 1.25}$ | $\mathbf{54.19 \pm 1.26}$ | $\mathbf{52.66 \pm 1.25}$ | $\mathbf{52.79 \pm 1.23}$ |
| Set Transformer | $48.81 \pm 1.21$ | $50.08 \pm 1.24$ | $50.75 \pm 1.23$ | $50.03 \pm 1.23$ | $49.41 \pm 1.20$ |
| 5-shot 3-way test on PAIP dataset | | | | | |
| Attention-based MIL pooling | $67.04 \pm 1.00$ | $66.06 \pm 1.17$ | $66.61 \pm 1.10$ | $\mathbf{70.19 \pm 0.87}$ | $\mathbf{70.54 \pm 0.81}$ |
| Deep Set | $\mathbf{69.42 \pm 0.85}$ | $\mathbf{69.93 \pm 0.92}$ | $\mathbf{70.52 \pm 0.87}$ | $69.96 \pm 0.84$ | $68.39 \pm 0.84$ |
| Set Transformer | $66.61 \pm 0.91$ | $67.57 \pm 0.95$ | $67.78 \pm 0.95$ | $68.24 \pm 0.85$ | $67.20 \pm 0.86$ |
| 1-shot 9-way test on NCT dataset | | | | | |
| Attention-based MIL pooling | $60.04 \pm 1.40$ | $64.53 \pm 1.29$ | $64.81 \pm 1.31$ | $64.00 \pm 1.34$ | $66.66 \pm 1.32$ |
| Deep Set | $\mathbf{68.21 \pm 1.30}$ | $68.17 \pm 1.31$ | $\mathbf{68.69 \pm 1.30}$ | $\mathbf{69.24 \pm 1.28}$ | $\mathbf{68.18 \pm 1.33}$ |
| Set Transformer | $67.76 \pm 1.31$ | $\mathbf{68.52 \pm 1.30}$ | $68.55 \pm 1.28$ | $68.33 \pm 1.28$ | $67.72 \pm 1.28$ |
| 5-shot 9-way test on NCT dataset | | | | | |
| Attention-based MIL pooling | $81.94 \pm 0.75$ | $82.40 \pm 0.72$ | $84.46 \pm 0.65$ | $86.49 \pm 0.62$ | $\mathbf{87.66 \pm 0.59}$ |
| Deep Set | $85.18 \pm 0.60$ | $85.87 \pm 0.60$ | $87.11 \pm 0.56$ | $87.06 \pm 0.61$ | $85.81 \pm 0.66$ |
| Set Transformer | $\mathbf{86.45 \pm 0.62}$ | $\mathbf{87.74 \pm 0.59}$ | $\mathbf{87.97 \pm 0.58}$ | $\mathbf{88.00 \pm 0.59}$ | $86.92 \pm 0.61$ |

Table 12: Performance of FACIEL-FSP with three different set-input models; average F1 and CI are reported.

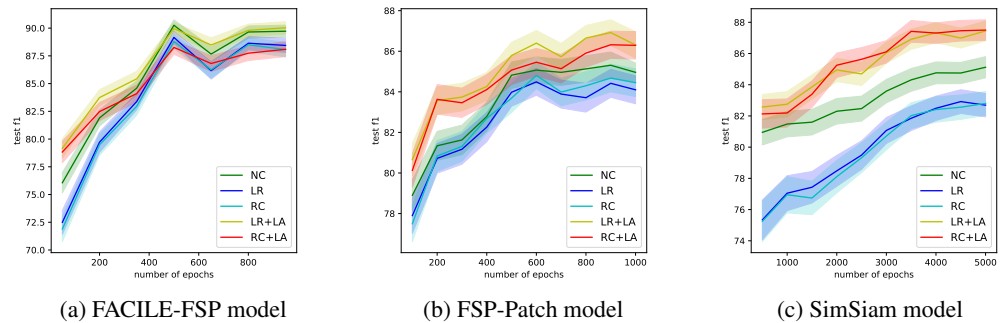

(a) FACILE-FSP model      (b) FSP-Patch model      (c) SimSiam model

Figure 12: Learning curves of FACILE-FSP model, FSP-Patch model, and SimSiam. The mean F1 score and CI of 5 few-shot models tested on the LC dataset with 5-shot are shown with curves.

from the trained FACILE-FSP is then evaluated on LC, PAIP, and NCT datasets with shot numbers 1 and 5. The corresponding performances are reported in table 13.

Observing table 13, we find that models with an input set size of 5 consistently demonstrate superior performance for LC and PAIP datasets. While slight improvements are observed for larger input set sizes, they are not substantial. Conversely, for the NCT dataset, as presented in table 13, the best performance is attained when the input set size is 10.

## G    CONTRASTIVE AND NON-CONTRASTIVE LEARNING MODELS

Self-supervised learning achieves promising results on multiple visual tasks (Bachman et al., 2019; He et al., 2020; Chen et al., 2020; Grill et al., 2020; Caron et al., 2020; Chen & He, 2021). Contrastive learning method avoid collapse by encouraging the representations to be far apart for views from different images. Henaff (2020); He et al. (2020); Misra & Maaten (2020); Chen et al. (2020) implemented instance discrimination, in which a pair of augmented views from the same image are positive and others are negative. Caron et al. (2020; 2018) contrasted different cluster of positives.

| set size | 2 | 5 | 10 | 15 |
|---|---|---|---|---|
| \multicolumn{5}{c}{1-shot 5-way test on LC dataset} | | | | |
| NC | $75.29 \pm 1.33$ | $\mathbf{77.84 \pm 1.16}$ | $74.88 \pm 1.36$ | $75.25 \pm 1.29$ |
| LR | $73.72 \pm 1.33$ | $\mathbf{77.56 \pm 1.16}$ | $73.84 \pm 1.29$ | $74.00 \pm 1.27$ |
| RC | $74.10 \pm 1.34$ | $\mathbf{77.56 \pm 1.17}$ | $73.42 \pm 1.31$ | $73.42 \pm 1.29$ |
| LR+LA | $75.27 \pm 1.28$ | $\mathbf{79.16 \pm 1.09}$ | $74.41 \pm 1.31$ | $74.92 \pm 1.26$ |
| RC+LA | $74.36 \pm 1.33$ | $\mathbf{77.38 \pm 1.18}$ | $72.60 \pm 1.34$ | $73.16 \pm 1.32$ |
| \multicolumn{5}{c}{5-shot 5-way test on LC dataset} | | | | |
| NC | $90.62 \pm 0.56$ | $90.35 \pm 0.50$ | $90.62 \pm 0.57$ | $\mathbf{90.83 \pm 0.55}$ |
| LR | $89.41 \pm 0.63$ | $\mathbf{90.91 \pm 0.47}$ | $89.80 \pm 0.59$ | $89.63 \pm 0.60$ |
| RC | $89.11 \pm 0.63$ | $\mathbf{91.54 \pm 0.46}$ | $89.26 \pm 0.61$ | $89.25 \pm 0.60$ |
| LR+LA | $90.46 \pm 0.58$ | $\mathbf{91.68 \pm 0.50}$ | $90.29 \pm 0.57$ | $90.46 \pm 0.56$ |
| RC+LA | $89.64 \pm 0.63$ | $\mathbf{90.97 \pm 0.54}$ | $88.52 \pm 0.66$ | $89.00 \pm 0.64$ |
| NC | $48.95 \pm 1.24$ | $52.04 \pm 1.25$ | $51.72 \pm 1.22$ | $\mathbf{52.46 \pm 1.20}$ |
| LR | $50.55 \pm 1.22$ | $53.27 \pm 1.25$ | $52.33 \pm 1.25$ | $\mathbf{53.38 \pm 1.23}$ |
| RC | $50.14 \pm 1.25$ | $\mathbf{54.19 \pm 1.26}$ | $53.04 \pm 1.24$ | $52.68 \pm 1.25$ |
| LR+LA | $50.12 \pm 1.22$ | $52.66 \pm 1.25$ | $52.96 \pm 1.21$ | $\mathbf{53.41 \pm 1.21}$ |
| RC+LA | $49.91 \pm 1.22$ | $\mathbf{52.79 \pm 1.23}$ | $51.67 \pm 1.17$ | $51.51 \pm 1.20$ |
| \multicolumn{5}{c}{5-shot 3-way test on PAIP dataset} | | | | |
| NC | $66.99 \pm 0.93$ | $\mathbf{69.42 \pm 0.85}$ | $69.10 \pm 0.91$ | $69.08 \pm 0.87$ |
| LR | $68.11 \pm 0.94$ | $69.93 \pm 0.92$ | $\mathbf{70.30 \pm 0.90}$ | $69.28 \pm 0.90$ |
| RC | $68.63 \pm 0.91$ | $\mathbf{70.52 \pm 0.87}$ | $70.45 \pm 0.87$ | $70.12 \pm 0.90$ |
| LR+LA | $69.03 \pm 0.83$ | $69.96 \pm 0.84$ | $\mathbf{70.25 \pm 0.81}$ | $70.00 \pm 0.81$ |
| RC+LA | $67.32 \pm 0.83$ | $\mathbf{68.39 \pm 0.84}$ | $68.35 \pm 0.83$ | $67.70 \pm 0.81$ |
| NC | $66.31 \pm 1.36$ | $68.21 \pm 1.30$ | $\mathbf{72.44 \pm 1.25}$ | $72.05 \pm 1.27$ |
| LR | $68.55 \pm 1.32$ | $68.17 \pm 1.31$ | $\mathbf{72.62 \pm 1.25}$ | $72.14 \pm 1.27$ |
| RC | $68.58 \pm 1.32$ | $68.69 \pm 1.30$ | $\mathbf{72.60 \pm 1.25}$ | $72.04 \pm 1.27$ |
| LR+LA | $67.42 \pm 1.33$ | $69.24 \pm 1.28$ | $\mathbf{72.18 \pm 1.26}$ | $71.92 \pm 1.27$ |
| RC+LA | $65.87 \pm 1.36$ | $68.18 \pm 1.33$ | $\mathbf{69.98 \pm 1.31}$ | $69.88 \pm 1.28$ |
| \multicolumn{5}{c}{5-shot 9-way test on NCT dataset} | | | | |
| NC | $85.28 \pm 0.72$ | $85.18 \pm 0.60$ | $\mathbf{88.25 \pm 0.56}$ | $88.22 \pm 0.57$ |
| LR | $86.39 \pm 0.69$ | $85.87 \pm 0.60$ | $\mathbf{88.80 \pm 0.55}$ | $88.55 \pm 0.55$ |
| RC | $87.03 \pm 0.66$ | $87.11 \pm 0.56$ | $\mathbf{89.25 \pm 0.52}$ | $89.02 \pm 0.54$ |
| LR+LA | $86.85 \pm 0.65$ | $87.06 \pm 0.61$ | $88.52 \pm 0.59$ | $\mathbf{88.93 \pm 0.55}$ |
| RC+LA | $85.60 \pm 0.70$ | $85.81 \pm 0.66$ | $87.40 \pm 0.63$ | $\mathbf{87.74 \pm 0.59}$ |

Table 13: Abation on set size; models tested on LC, PAIP, and NCT dataset; average F1 and CI are reported.

Non-contrastive models (Grill et al., 2020; Richemond et al., 2020; Chen & He, 2021) removed the reliance on negatives. These non-contrastive models achieved strong results in the ImageNet (Deng et al., 2009) pretraining setting. SimSiam (Chen & He, 2021) works with typical batches and does not rely on large-batch training, which makes it preferable for academics and practitioners with low computation resources.

In this section, some contrastive learning and non-contrastive learning models, e.g., SimCLR, Sup-Con, and SimSiam, that are used in this paper are explained. Details of implementation are provided. There are three main components in SimCLR and SupCon framework. We follow the notation of Khosla et al. (2020) in this section to explain SimCLR and SupCon.

- Data augmentation $Aug(\cdot)$. For each input sample $x$, the augmentation module generates two random augmented views, i.e., $\tilde{x} \sim Aug(x)$. The augmentation schedules used in this paper are explained in section D.

- Encoder $Enc(\cdot)$. The encoder extracts a representation vector $r = Enc(\tilde{x})$. The pair of augmented views are separately fed to the same encoder and generate a pair of representations. The $r$ is normalized to the unit hypersphere.

- Projection head $Proj(\cdot)$. It maps $r$ to a vector $z = Proj(r)$. We instantiate $Proj(\cdot)$ as a multi-layer perceptron (MLP) with a single hidden layer of size 512 and output vector size of 512. We also normalize the output to the unit hypersphere.

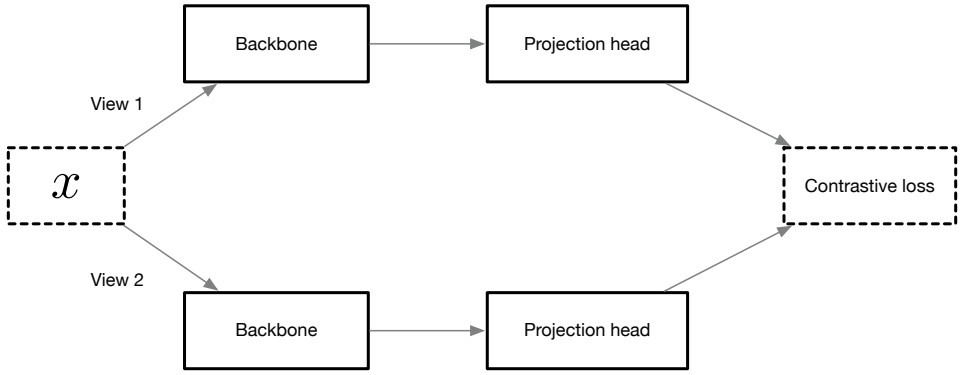

Figure 13: Abstraction of SimCLR structure

For a set of $N$ randomly sampled sample/label pairs, $\{(x_k, y_k)\}_{k=1}^{N}$. The corresponding batch used for training consists of $2N$ pairs, $\{(\tilde{x}_l, \tilde{y}_l)\}_{l=1}^{2N}$, where $\tilde{x}_{2k-1}$ and $\tilde{x}_{2k}$ are two random augmented views of $x_k$ and $\tilde{y}_{2k-1} = \tilde{y}_{2k} = y_k$.

## G.1 SIMCLR

Let $i \in I \equiv \{1 \ldots 2N\}$ be the index of an arbitrary augmented sample and let $j(i)$ be the index of the other augmented sample originating from the same source sample. The abstraction of SimCLR structure can be found in figure 13. In SimCLR, the loss takes the following form.

$$\mathcal{L}^{\text{self}} = \sum_{i \in I} \mathcal{L}_i^{\text{self}} = -\sum_{i \in I} \log \frac{\exp\left(z_i \cdot z_{j(i)}/\tau\right)}{\sum_{a \in A(i)} \exp\left(z_i \cdot z_a/\tau\right)} \tag{2}$$

where $\tau$ is the temperature parameter. $A(i) \equiv I \backslash \{i\}$. The denominator has a total of $2N - 1$ terms.

In this paper, the $\tau$ is always set to $0.07$. The patches are augmented randomly by the augmentation module described in section D. We use an MLP as a projection head with two fully-connected layers, a hidden dimension of 512, and an output dimension of 512.

## G.2 SUPCON

For supervised learning, the contrastive loss in equation 2 cannot handle class discrimination (Khosla et al., 2020). Khosla et al. (2020) proposed two straightforward ways, as shown in equation 3 and equation 4, to generalize equation 2 to incorporate supervison.

$$\mathcal{L}_{\text{out}}^{\text{sup}} = \sum_{i \in I} \mathcal{L}_{\text{out},i}^{\text{sup}} = \sum_{i \in I} \frac{-1}{|P(i)|} \sum_{p \in P(i)} \log \frac{\exp\left(z_i \cdot z_p/\tau\right)}{\sum_{a \in A(i)} \exp\left(z_i \cdot z_a/\tau\right)} \tag{3}$$

$$\mathcal{L}_{in}^{\text{sup}} = \sum_{i \in I} \mathcal{L}_{in,i}^{\text{sup}} = \sum_{i \in I} -\log \left\{ \frac{1}{|P(i)|} \sum_{p \in P(i)} \frac{\exp\left(z_i \cdot z_p/\tau\right)}{\sum_{a \in A(i)} \exp\left(z_i \cdot z_a/\tau\right)} \right\} \tag{4}$$

Here $P(i) \equiv \{p \in A(i) : \tilde{y}_p = \tilde{y}_i\}$ is the set of indices of all positives in the batch distinct from $i$. The authors showed that $\mathcal{L}_{in}^{\text{sup}} \leq \mathcal{L}_{\text{out}}^{\text{sup}}$ and $\mathcal{L}_{\text{out}}^{\text{sup}}$ is the superior supervised loss function. Thus, we use SupCon with equation 3 as the default loss. The $\tau$ is also set to $0.07$.

In our model FACILE-SupCon, the input sample is a set of randomly sampled patches and labels are slide properties, i.e., organs or TCGA projects. Each patch is augmented randomly by the augmentation module described in section D. The feature map $e$ and set function $g$ work as the encoder $Enc(\cdot)$. We also use an MLP as a projection head with two fully-connected layers, a hidden dimension of 512, and an output dimension of 512.

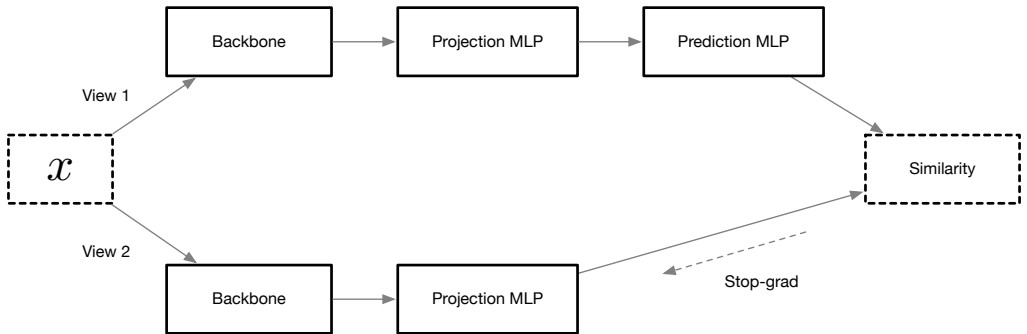

Figure 14: Abstraction of SimSiam structure

When employing set-input data with the SupCon method, the standard practice of augmenting each instance within a set poses significant challenges for the training of SupCon models. These challenges stem from two main aspects: 1) Complexity in maximizing agreement with set-input data: SupCon is traditionally trained to maximize agreement between differently augmented views of the same data point using labeled data. In our application, using set-input data means that we apply conventional data augmentation methods to each instance within a set. This results in an independently augmented set of images, as opposed to augmenting a single instance. This complexity makes it more challenging to achieve the desired maximization of agreement. 2) Constraints on batch sizes due to set inputs: Set-input models take a batch of sets as input instead of a batch of instances. It requires us to use relatively smaller batch sizes when using the same hardware configuration because of the set input. It's important to emphasize that the batch size is a critical factor for the effectiveness of the SupCon model.

We have observed that despite these challenges, the performance of FACILE-SupCon is commendable in contexts involving smaller datasets or less complex models, i.e., CIFAR-100 in section 3.2 and section 3.3 or smaller trainable models as discussed in Appendices B.1 and B.3. We believe that our approach, with its nuanced application of SupCon in a set-input context, offers a valuable contribution to the field and shows the versatility of the FACILE algorithm.

### G.3 SIMSIAM

Non-contrastive models, e.g., SimSiam and BYOL, achieve strong results in typical ImageNet (Deng et al., 2009) pretraining setting (Chen & He, 2021; Grill et al., 2020; Li et al., 2022). Among the non-contrastive models, SimSiam removes the negatives and uses stop-grad to avoid collapse. Besides, it trains faster, requires less GPU memory, and works well with small batch size (Chen & He, 2021; Li et al., 2022), which makes it extremely appealing to academics.

The abstraction of SimSiam structure is shown in figure 14. Given two augmented views $\tilde{x}_1$ and $\tilde{x}_2$ of the same image $x$, SimSiam learns to use $\tilde{x}_1$ to predict the representation of $\tilde{x}_2$. Specifically, $\tilde{x}_1$ is passed into the online backbone network on the upper. The $\tilde{x}_2$ is passed into the target backbone network on the lower. The outputs of the two backbone networks are passed to the projection MLPs and then a prediction MLP is used to predict the projected representation of $\tilde{x}_2$ from the projected representation of $\tilde{x}_1$. SimSiam uses the same network for the online and target backbone and projection networks.

In our paper, the projection MLP has 3 fully-connected layers with a hidden dimension of 512 and an output dimension of 512. It has batch normalization (BN) applied to each fully-connected layer including its output fully-connected layer. The prediction MLP also has BN applied to its hidden fully-connected layer. Its output fully-connected layer does not have BN or ReLU. The prediction MLP has 2 layers.

# H    EXCESS RISK BOUND OF FACILE

Our proof framework follows closely the work of Robinson et al. (2020). We consider the setting where we have some coarse-grained labels of some sets, rather than instances and the downstream classifiers only use the learned embeddings to train and test on the downstream tasks. /Robinson et al. (2020) considers a different setting where each instance has a weak label and a strong label, and the strong label predictor learns to predict the strong labels from the instances and their corresponding embeddings learned with weak labels. The diagram of only using trained embeddings for downstream tasks is more often used in self-supervised learning and representation learning for FSL literature (Du et al., 2020; Yang et al., 2021; Bachman et al., 2019; He et al., 2020; Chen et al., 2020; Grill et al., 2020; Caron et al., 2020; Chen & He, 2021). The coarse-grained data contains useful information, which is characterized by our defined Lipschitzness, to pretrain a instance feature map that can be leveraged for downstream FSL. We include the full proof of our key result as follows.

In order to prove theorem 4, we first split the excess risk by the following proposition.

**Proposition 5.** *Suppose that $f^*$ is $L$-Lipschitz relative to $\mathcal{E}$. The excess risk* $\mathbb{E}\left[\ell^{\mathrm{fg}}_{\hat{f}\circ\hat{e}}(X,Y) - \ell^{\mathrm{fg}}_{f^*\circ e^*}(X,Y)\right]$ *is bounded by,*

$$2L\mathrm{Rate}_m(\ell^{\mathrm{cg}}, P_{S,W}, \mathcal{E}) + \mathrm{Rate}_n(\ell^{\mathrm{fg}}, \hat{P}_{Z,Y}, \mathcal{F})$$

**Proof.**    We split the excess risk into three parts

$$\mathbb{E}_{P_{X,Y}}\left[\ell^{\mathrm{fg}}_{\hat{f}\circ\hat{e}}(X,Y) - \ell^{\mathrm{fg}}_{f^*\circ e^*}(X,Y)\right]$$

$$=\mathbb{E}_{P_{X,Y}}\left[\ell^{\mathrm{fg}}_{\hat{f}\circ\hat{e}}(X,Y) - \ell^{\mathrm{fg}}_{f^*\circ\hat{e}}(X,Y)\right] + \mathbb{E}_{P_{X,Y}}\left[\ell^{\mathrm{fg}}_{f^*\circ\hat{e}}(X,Y) - \ell^{\mathrm{fg}}_{f^*\circ e_0}(X,Y)\right]$$

$$+ \mathbb{E}_{P_{X,Y}}\left[\ell^{\mathrm{fg}}_{f^*\circ e_0}(X,Y) - \ell^{\mathrm{fg}}_{f^*\circ e^*}(X,Y)\right]$$

For the second term and third term, relative Lipschitzness of $f^*$ to $\mathcal{E}$ delivers

$$\mathbb{E}_{P_{X,Y}}\left[\ell^{\mathrm{fg}}_{f^*\circ\hat{e}}(X,Y) - \ell^{\mathrm{fg}}_{f^*\circ e_0}(X,Y)\right] = \mathbb{E}_{P_{X,Y,S,W}}\left[\ell^{\mathrm{fg}}_{f^*\circ\hat{e}}(X,Y) - \ell^{\mathrm{fg}}_{f^*\circ e_0}(X,Y)\right]$$

$$\leq L\mathbb{E}_{P_{X,Y,S,W}}\ell^{\mathrm{cg}}\left(g_{\hat{e}}\circ\hat{e}(S), g_{e_0}\circ e_0(S)\right)$$

$$= L\mathbb{E}_{P_{S,W}}\ell^{\mathrm{cg}}\left(g_{\hat{e}}\circ\hat{e}(S), g_{e_0}\circ e_0(S)\right),$$

$$\mathbb{E}_{P_{X,Y}}\left[\ell^{\mathrm{fg}}_{f^*\circ e_0}(X,Y) - \ell^{\mathrm{fg}}_{f^*\circ e^*}(X,Y)\right] = \mathbb{E}_{P_{X,Y,S,W}}\left[\ell^{\mathrm{fg}}_{f^*\circ e_0}(X,Y) - \ell^{\mathrm{fg}}_{f^*\circ e^*}(X,Y)\right]$$

$$\leq L\mathbb{E}_{P_{X,Y,S,W}}\ell^{\mathrm{cg}}\left(g_{e_0}\circ e_0(S), g_{e^*}\circ e^*(S)\right)$$

$$= L\mathbb{E}_{P_{S,W}}\ell^{\mathrm{cg}}\left(g_{e_0}\circ e_0(S), g_{e^*}\circ e^*(S)\right)$$

Since $e^*$ attains minimal risk and $W = g_{e_0}\circ e_0(S)$, the sum of the two terms can be bounded by,

$$L\mathbb{E}_{P_{S,W}}\ell^{\mathrm{cg}}\left(g_{\hat{e}}\circ\hat{e}(S), g_{e_0}\circ e_0(S)\right) + L\mathbb{E}_{P_{S,W}}\ell^{\mathrm{cg}}\left(g_{e_0}\circ e_0(S), g_{e^*}\circ e^*(S)\right)$$

$$\leq 2L\mathbb{E}_{P_{S,W}}\ell^{\mathrm{cg}}\left(g_{\hat{e}}\circ\hat{e}(S), W\right) \leq 2L\mathrm{Rate}_m(\ell^{\mathrm{cg}}, P_{S,W}, \mathcal{E})$$

By combining the bounds on the three terms we can get the claim.

The central condition is well-known to yield fast rates for supervised learning (Van Erven et al., 2015). It directly implies that we could learn a map $Z \to Y$ with $\widetilde{\mathcal{O}}(1/n)$ excess risk. The difficulty is that at test time we would need access to latent value $Z = e(X)$. To circumnavigate this hurdle, we replace $e_0$ with $\hat{e}$ and solve the supervised learning problem $(\ell^{\mathrm{fg}}, \hat{P}_{Z,Y}, \mathcal{F})$.

It is not clear whether this surrogate problem satisfies the central condition. We show that $(\ell^{\mathrm{fg}}, \hat{P}_{Z,Y}, \mathcal{F})$ indeed satisfies a weak central condition and shows weak central condition still enables strong excess risk guarantees.

Following Robinson et al. (2020); Van Erven et al. (2015), we define the central condition on $\mathcal{F}$.

**Definition 6.** (The central condition). A learning problem $(\ell^{\mathrm{fg}}, P_{Z,Y}, \mathcal{F})$ on $\mathcal{Z} \times \mathcal{Y}$ is said to satisfy the $\epsilon$-weak $\eta$-central condition if there exists an $f^* \in \mathcal{F}$ such that

$$\mathbb{E}_{(Z,Y) \sim P_{Z,Y}} \left[ e^{\eta(\ell_{f^*}^{\mathrm{fg}}(Z,Y) - \ell_f^{\mathrm{fg}}(Z,Y))} \right] \leq e^{\eta \epsilon}$$

for all $f \in \mathcal{F}$. The 0-weak central condition is known as the strong central condition.

**Capturing relatedness of pretraining and downstream task with the central condition.** Intuitively, the strong central condition requires that the minimal risk model $f^*$ attains a higher loss than $f \in \mathcal{F}$ on a set of $Z, Y$ with an exponentially small probability. This is likely to happen when $Z$ is highly predictive of $Y$ so that the probability of $P(Y|Z)$ concentrates in a single location for most $Z$. If $f^*$ in $\mathcal{F}$ such that $f^*(Z)$ maps into this concentration, $\ell_{f^*}^{\mathrm{fg}}(Z, Y)$ will be close to zero most of the time.

We assume that the strong central condition holds for the learning problem $(\ell^{\mathrm{fg}}, P_{Z,Y}, \mathcal{F})$ where $Z = e_0(X)$. Similar to Robinson et al. (2020), we split the learning procedure into two supervised tasks as depicted in Algorithm 1. In the algorithm, we replace $(\ell^{\mathrm{fg}}, P_{Z,Y}, \mathcal{F})$ with $(\ell^{\mathrm{fg}}, \hat{P}_{Z,Y}, \mathcal{F})$.

We will show that $(\ell^{\mathrm{fg}}, \hat{P}_{Z,Y}, \mathcal{F})$ satisfies the weak central condition.

**Proposition 7.** *Assume that $\ell^{\mathrm{cg}}(w, w') = \mathbb{1}\{w \neq w'\}$ and that $\ell^{\mathrm{fg}}$ is bounded by $B > 0$, $\mathcal{F}$ is L-Lipschitz relative to $\mathcal{E}$, and that $(\ell^{\mathrm{fg}}, P_{Z,Y}, \mathcal{F})$ satisfies $\epsilon$-weak central condition. Then $(\ell^{\mathrm{fg}}, \hat{P}_{Z,Y}, \mathcal{F})$ satisfies the $\epsilon + \mathcal{O}\left(\frac{\exp(\eta B)}{\eta} \mathrm{Rate}_m\left(\mathcal{E}, P_{S,W}\right)\right)$-weak central condition with probability at least $1 - \delta$.*

**Proof.** Note that

$$\frac{1}{\eta} \log \mathbb{E}_{\hat{P}_{Z,Y}} \exp\left(\eta(\ell_{f^*}^{\mathrm{fg}} - \ell_f^{\mathrm{fg}})\right) = \frac{1}{\eta} \log \mathbb{E}_{P_{X,Y}} \exp\left(\eta(\ell_{f^* \circ \hat{e}}^{\mathrm{fg}} - \ell_{f \circ \hat{e}}^{\mathrm{fg}})\right)$$

To prove that $(\ell^{\mathrm{fg}}, \hat{P}_{Z,Y}, \mathcal{F})$ satisfies the central condition we therefore need to bound $\frac{1}{\eta} \log \mathbb{E}_{P_{X,Y}} \exp\left(\eta(\ell_{f^* \circ \hat{e}}^{\mathrm{fg}} - \ell_{f \circ \hat{e}}^{\mathrm{fg}})\right)$ by some constant.

$$
\begin{aligned}
&\frac{1}{\eta} \log \mathbb{E}_{P_{X,Y}} \exp\left(\eta(\ell_{f^* \circ \hat{e}}^{\mathrm{fg}} - \ell_{f \circ \hat{e}}^{\mathrm{fg}})\right) \\
=&\frac{1}{\eta} \log \mathbb{E}_{P_{X,Y,S,W}} \exp\left(\eta(\ell_{f^* \circ \hat{e}}^{\mathrm{fg}} - \ell_{f \circ \hat{e}}^{\mathrm{fg}})\right) \\
=&\frac{1}{\eta} \log \mathbb{E}_{P_{X,Y,S,W}} \left[\exp(\eta(\ell_{f^* \circ \hat{e}}^{\mathrm{fg}} - \ell_{f \circ \hat{e}}^{\mathrm{fg}})) \mathbb{1}\{\hat{g}_{\hat{e}} \circ \hat{e}(S) = W\}\right] + \\
&\quad\quad \frac{1}{\eta} \log \mathbb{E}_{P_{X,Y,S,W}} \left[\exp(\eta(\ell_{f^* \circ \hat{e}}^{\mathrm{fg}} - \ell_{f \circ \hat{e}}^{\mathrm{fg}})) \mathbb{1}\{\hat{g}_{\hat{e}} \circ \hat{e}(S) \neq W\}\right] \\
=&\underbrace{\frac{1}{\eta} \log \mathbb{E}_{P_{X,Y,S,W}} \left[\exp(\eta(\ell_{f^* \circ e_0}^{\mathrm{fg}} - \ell_{f \circ e_0}^{\mathrm{fg}})) \mathbb{1}\{\hat{g}_{\hat{e}} \circ \hat{e}(S) = W\}\right]}_{\text{first term}} + \\
&\underbrace{\frac{1}{\eta} \log \mathbb{E}_{P_{X,Y,S,W}} \left[\exp(\eta(\ell_{f^* \circ \hat{e}}^{\mathrm{fg}} - \ell_{f \circ \hat{e}}^{\mathrm{fg}})) \mathbb{1}\{\hat{g}_{\hat{e}} \circ \hat{e}(S) \neq W\}\right]}_{\text{second term}}
\end{aligned}
$$

The third line follows from the fact that for any $f$ in the event $\{\hat{g}_{\hat{e}} \circ \hat{e}(S) = W\}$ we have $\ell_{f \circ \hat{g}}^{\mathrm{fg}} = \ell_{f \circ g_0}^{\mathrm{fg}}$.

This is because $|\ell_{f \circ \hat{e}}^{\mathrm{fg}}(X, Y) - \ell_{f \circ e_0}^{\mathrm{fg}}(X, Y)| \leq L\ell^{\mathrm{cg}}(g_{\hat{e}} \circ \hat{e}(S), g_{e_0} \circ e_0(S)) = L\ell^{\mathrm{cg}}(W, W) = 0$.

We get $\frac{1}{\eta} \log \mathbb{E}_{P_{X,Y,S,W}} \left[\exp(\eta(\ell_{f^* \circ e_0}^{\mathrm{fg}} - \ell_{f \circ e_0}^{\mathrm{fg}}))\right]$ after we drop the $\mathbb{1}\{\hat{g}_{\hat{e}} \circ \hat{e}(S) = W\}$. It is bounded by $\epsilon$ with the weak central condition. The second term is bounded by

$$\frac{1}{\eta} \log \mathbb{E}_{P_{X,Y,S,W}} \left[ \exp(\eta(\ell^{\mathrm{fg}}_{f^* \circ \hat{e}} - \ell^{\mathrm{fg}}_{f \circ \hat{e}})) \mathbb{1}\{\hat{g}_{\hat{e}} \circ \hat{e}(S) \neq W\} \right]$$

$$\leq \frac{1}{\eta} \log \mathbb{E}_{P_{X,Y,S,W}} \left[ \exp(\eta B) \mathbb{1}\{\hat{g}_{\hat{e}} \circ \hat{e}(S) \neq W\} \right]$$

$$= \frac{1}{\eta} \log \mathbb{E}_{P_{S,W}} \left[ \exp(\eta B) \mathbb{1}\{\hat{g}_{\hat{e}} \circ \hat{e}(S) \neq W\} \right]$$

$$< \frac{1}{\eta} \mathbb{E}_{P_{S,W}} \left[ \exp(\eta B) \mathbb{1}\{\hat{g}_{\hat{e}} \circ \hat{e}(S) \neq W\} \right]$$

$$= \frac{\exp(\eta B)}{\eta} P_{S,W}(\hat{g}_{\hat{e}} \circ \hat{e}(S) \neq W)$$

$$= \frac{\exp(\eta B)}{\eta} \mathrm{Rate}_m(\ell^{\mathrm{cg}}, P_{S,W}, \mathcal{E})$$

The first inequality uses the fact that $\ell^{\mathrm{fg}}$ is bounded by $B$. The forth line is because that $\log x < x$. By combining this bound with $\epsilon$ bound on the first term we can get the claimed result of proposition 7.

The proof of the main theorem further relies on a proposition provided by Robinson et al. (2020), as we show below:

**Proposition 8.** *Robinson et al. (2020) Suppose $(\ell^{\mathrm{fg}}, Q_{Z,Y}, \mathcal{F})$ satisfies the $\epsilon$-weak central condition, $\ell^{\mathrm{fg}}$ is bounded by $B > 0$, $\mathcal{F}$ is $L'$-Lipschitz in its $d$-dimensional parameters in the $l_2$ norm, $\mathcal{F}$ is contained in Euclidean ball of radius $R$, and $\mathcal{Y}$ is compact. Then when $\mathcal{A}_n(\ell^{\mathrm{fg}}, Q_{Z,Y}, \mathcal{F})$ is ERM, the excess risk $\mathbb{E}_{Z,Y \sim Q_{Z,Y}} \left[ \ell^{\mathrm{fg}}_{\hat{f}}(Z,Y) - \ell^{\mathrm{fg}}_{f^*}(Z,Y) \right]$ is bounded by,*

$$\mathcal{O}\left( V \frac{d \log \frac{RL'}{\epsilon} + \log \frac{1}{\delta}}{n} + V\epsilon \right)$$

*with probability at least $1 - \delta$, where $V = B + \epsilon$.*

**Proof of the main theorem:** If $m = \Omega(n^\beta)$, the $\mathrm{Rate}_m(\ell^{\mathrm{cg}}, P_{S,W}, \mathcal{E}) = \mathcal{O}(\frac{1}{m^\alpha}) = \mathcal{O}(\frac{1}{n^{\alpha\beta}})$. proposition 7 concludes that $(\ell^{\mathrm{fg}}, \hat{P}_{Z,Y}, \mathcal{F})$ satisfies the $\mathcal{O}(\frac{1}{n^{\alpha\beta}})$-weak central condition with probability at least $1 - \delta$. Thus by proposition 8, we can get $\mathrm{Rate}_n(\ell^{\mathrm{fg}}, \hat{P}_{Z,Y}, \mathcal{F}) = \mathcal{O}\left( \frac{d\alpha\beta \log RL'n + \log \frac{1}{\delta}}{n} + \frac{B}{n^{\alpha\beta}} \right)$. Combining bounds with proposition 5 we conclude that

$$\mathbb{E}\left[ \ell^{\mathrm{fg}}_{\hat{f} \circ \hat{e}}(X,Y) - \ell^{\mathrm{fg}}_{f^* \circ e^*}(X,Y) \right] \leq 2L\mathrm{Rate}_m(\ell^{\mathrm{cg}}, P_{S,W}, \mathcal{E}) + \mathrm{Rate}_n(\ell^{\mathrm{fg}}, \hat{P}_{Z,Y}, \mathcal{F})$$

$$\leq \mathcal{O}\left( \frac{d\alpha\beta \log RL'n + \log \frac{1}{\delta}}{n} + \frac{B}{n^{\alpha\beta}} + \frac{2L}{n^{\alpha\beta}} \right)$$

