# 2 FINE-GRAINED REPRESENTATION LEARNING FROM COARSE-GRAINED LABELS

**Notation** Our model pretrains on a collection of samples, denoted by $\{(s_i, w_i)\}_{i=1}^m$. Each $s_i$ is an input set of instances $\{x_j\}_{j=1}^a$, where $a$ represents the variable input set size. $\{w_i\}$ are the set-level coarse-grained labels. The space of all instances is $\mathcal{X}$ and the space of all instance labels, which we call fine-grained labels, is $\mathcal{Y}$. The space of pretraining data is $\mathcal{S} \times \mathcal{W}$, where $\mathcal{S} = \{\{x_1, \ldots, x_a\} : x_j \in \mathcal{X} \text{ for } \forall j \in [a]\}$ and $\mathcal{W}$ denotes the space of coarse-grained labels. We receive $(X, Y)$ from product space $\mathcal{X} \times \mathcal{Y}$ and corresponding $(S, W)$ from product space $\mathcal{S} \times \mathcal{W}$. The goal is to predict the strong labels $y \in \mathcal{Y}$ from the instance features $x \in \mathcal{X}$.

## 2.1 THE FACILE ALGORITHM

We study the model in an FSL setting where we have three datasets: (1) pretraining coarse-grained datasets $\mathcal{D}_m^{\mathrm{cg}} = \{(s_i, w_i)\}_{i=1}^m$ sampled i.i.d. from $P_{S,W}$ (2) fine-grained support dataset $\mathcal{D}_n^{\mathrm{fg}} = \{(x_i, y_i)\}_{i=1}^n$ sampled i.i.d., from $P_{X,Y}$, and (3) query set $\mathcal{D}^{\mathrm{query}}$. The support set $\mathcal{D}_n^{\mathrm{fg}}$ contains $c$ classes and $k$ samples $x$ in each class (i.e., $n \equiv kc$). We assume a latent space $\mathcal{Z}$ for embedding $Z$. We define instance feature maps $\mathcal{E} = \{e : \mathcal{X} \to \mathcal{Z}\}$, set-input functions $\mathcal{G} = \{g : \mathcal{M} \to \mathcal{W}\}$ where $\mathcal{M} = \{\{z_1, \ldots, z_a\} : z_j \in \mathcal{Z} \text{ for } j \in [a]\}$, and fine-grained label predictors $\mathcal{F} = \{f : \mathcal{Z} \to \mathcal{Y}\}$. The corresponding set-input feature map of an instance feature map $e$ is defined as $\phi^e : \mathcal{S} \to \mathcal{M}$. We assume the class of $f$ is parameterized and identify $f$ with parameter vectors for theoretical analysis. We then learn feature map $e$, fine-grained label predictor $f$, and predict fine-grained label with $f \circ e$. The schema of our model is illustrated in figure 3.

We assume two loss functions: $\ell^{\mathrm{fg}} : \mathcal{Y} \times \mathcal{Y} \to \mathbb{R}$ for fine-grained label prediction and $\ell^{\mathrm{cg}} : \mathcal{W} \times \mathcal{W} \to \mathbb{R}$ for coarse-grained label prediction. $\ell^{\mathrm{fg}}$ measures the loss of the fine-grained label predictor. We assume this loss is differentiable in its first argument. $\ell^{\mathrm{cg}}$ measures the loss of pretraining with coarse-grained labels. For theoretical analysis, we are interested in two particular cases of $\ell^{\mathrm{cg}}$: i) $\ell^{\mathrm{cg}}(w, w') = \mathbb{1}\{w \neq w'\}$ when $\mathcal{W}$ is a categorical space; and ii) $\ell^{\mathrm{cg}}(w, w') = \|w - w'\|$ (for some norm $\|\cdot\|$ on $\mathcal{W}$) when $\mathcal{W}$ is a continuous space. We can also measure the loss of a feature map $e$ by $\ell_e^{\mathrm{cg}} = \ell^{\mathrm{cg}}(g_e \circ \phi^e(s), w)$, where $g_e \in \arg\min_g \mathbb{E}_{P_{S,W}} \ell^{\mathrm{cg}}(g \circ \phi^e(S), W)$. We assume there is an unknown "good" embedding $M = \phi^{e_0}(S) \in \mathcal{M}$, by which a set-input function $g_{e_0}$ can determine $W$, i.e., $g_{e_0}(M) = g_{e_0} \circ \phi^{e_0}(S) = W$. The strict assumption of equality can be relaxed by incorporating an additive error term into our risk bounds of $g_{e_0} \circ \phi^{e_0}$.

Our primary goal is to learn an instance label predictor or fine-grained label predictor $\hat{f} \circ \hat{e}$ that achieves low risk $\mathbb{E}_{P_{X,Y}}[\ell^{\mathrm{fg}}(\hat{f} \circ \hat{e}(X), Y)]$ and we can bound the excess risk:

$$\mathbb{E}_{P_{X,Y}}[\ell^{\mathrm{fg}}(\hat{f} \circ \hat{e}(X), Y) - \ell^{\mathrm{fg}}(f^* \circ e^*(X), Y)] \tag{1}$$

where $e^* \in \arg\min_{e \in \mathcal{E}} \mathbb{E}_{P_{S,W}} \ell_e^{\mathrm{cg}}(S, W)$ and $f^* \in \arg\min_{f \in \mathcal{F}} \mathbb{E}_{P_{X,Y}}[\ell^{\mathrm{fg}}(f \circ e^*(X), Y)]$.

The pseudocode for FACILE is provided in Algorithm 1, and we further illustrate the FACILE algorithm in figure 2. Given an input set $s_i$ comprising instances $x_1, \ldots, x_a$, the feature map $e$ is

---

**Algorithm 1** FACILE algorithm

---

1: **Input:** loss functions $\ell^{\text{fg}}$, $\ell^{\text{cg}}$, predictors $\mathcal{E}$, $\mathcal{G}$, $\mathcal{F}$, datasets $\mathcal{D}_m^{\text{cg}}$ and $\mathcal{D}_n^{\text{fg}}$
2: obtain feature map $\hat{e} \leftarrow \mathcal{A}(\ell^{\text{cg}}, \mathcal{D}_m^{\text{cg}}, \mathcal{E})$
3: create dataset $\mathcal{D}_n^{\text{fg,aug}} = \{(z_i, y_i) : z_i = \hat{e}(x_i), (x_i, y_i) \in \mathcal{D}_n^{\text{fg}}\}_{i=1}^n$
4: obtain fine-grained label predictor $\hat{f} \circ \hat{e}$, where $\hat{f} \leftarrow \mathcal{A}(\ell^{\text{fg}}, \mathcal{D}_n^{\text{fg,aug}}, \mathcal{F})$
5: **Return:** $\hat{f} \circ \hat{e}$

---

employed to extract instance-level features for all the instances within the input set. Subsequently, a set-input model $g$ is utilized to generate set-level features based on the instance-level features. Our FACILE model is pretrained on coarse-grained labels in conjunction with either the FSP or the SupCon model (Khosla et al., 2020). During testing, we extract the pretrained feature map $\hat{e}$ and fine-tune a classifier $f$ using the generated embeddings from $\hat{e}$ and the fine-grained labels of the support set. The performance of the classifier $\hat{f}$ is then reported for the query set. Note that Algorithm 1 is generic since the two learning steps can use any supervised learning algorithm. Similar to Robinson et al. (2020), our analysis treats the case where $\mathcal{A}(\ell^{\text{cg}}, \mathcal{D}_m^{\text{cg}}, \mathcal{E})$ is empirical risk minimization (ERM) and is agnostic to the choice of $\mathcal{A}(\ell^{\text{fg}}, \mathcal{D}_n^{\text{fg}}, \mathcal{F})$.

## 2.2 THEORETICAL ANALYSIS

We denote the underlying distribution of $\mathcal{D}_m^{\text{cg}}$ as $P_{S,W}$ and the underlying distribution of $\mathcal{D}_n^{\text{fg}}$ as $P_{X,Y}$. We assume the joint distribution of $Z$ and $Y$ is $P_{Z,Y}$. After we learn the feature map $\hat{e}$, we can define a new distribution $\hat{P}_{Z,Y} = P(Z,Y)\mathbb{1}\{Z = \hat{e}(X)\}$, where $\mathbb{1}$ is the indicator function. The $\mathcal{D}_n^{\text{fg,aug}}$ is i.i.d. samples from $\hat{P}_{Z,Y}$. In order to include the underlying distribution of $\mathcal{D}_m^{\text{cg}}$, and $\mathcal{D}_n^{\text{fg}}$ into analysis, with a slight abuse of notation we use $\mathcal{A}_m(\ell^{\text{cg}}, P_{S,W}, \mathcal{E})$ to denote $\mathcal{A}(\ell^{\text{cg}}, \mathcal{D}_m^{\text{cg}}, \mathcal{E})$ and use $\mathcal{A}_n(\ell^{\text{fg}}, \hat{P}_{Z,Y}, \mathcal{F})$ to denote $\mathcal{A}(\ell^{\text{fg}}, \mathcal{D}_n^{\text{fg,aug}}, \mathcal{F})$.

The two learning algorithms are described as follows.

**Definition 1.** (Coarse-grained learning; pretraining) Let $\text{Rate}_m(\ell^{\text{cg}}, P_{S,W}, \mathcal{E}; \delta)$ (abbreviated to $\text{Rate}_m(\ell^{\text{cg}}, P_{S,W}, \mathcal{E})$) be the rate of $\mathcal{A}_m(\ell^{\text{cg}}, P_{S,W}, \mathcal{E})$ which takes $\ell^{\text{cg}}$, $\mathcal{E}$ and $m$ i.i.d. observations from $P_{S,W}$ as input, and return a feature map $\hat{e} \in \mathcal{E}$ such that

$$\mathbb{E}_{P_{S,W}} \ell_{\hat{e}}^{\text{cg}}(S, W) \leq \text{Rate}_m(\ell^{\text{cg}}, P_{S,W}, \mathcal{E}; \delta)$$

with probability at least $1 - \delta$.

**Definition 2.** (Fine-grained learning; downstream task learning) Let $\text{Rate}_m(\ell^{\text{fg}}, Q_{Z,Y}, \mathcal{F}; \delta)$ (abbreviated to $\text{Rate}_m(\ell^{\text{fg}}, Q_{Z,Y}, \mathcal{F})$) be the excess risk rate of $\mathcal{A}_n(\ell^{\text{fg}}, Q_{Z,Y}, \mathcal{F})$ which take $\ell^{\text{fg}}$, $\mathcal{F}$, and $n$ i.i.d. obser-

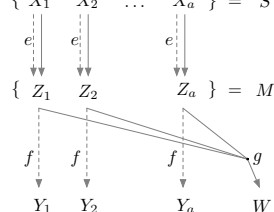

$$\{ X_1 \quad X_2 \quad \dots \quad X_a \} = S$$

$$\{ Z_1 \quad Z_2 \qquad Z_a \} = M$$

$$Y_1 \quad Y_2 \qquad Y_a \qquad W$$

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

FSL methods' performance on LC dataset is shown in table 4. We can see from table 4 that there is no improvement for the LC dataset after set size $a = 5$. More ablation results on set size with LC, PAIP, and NCT dataset can be seen in appendix table 12. We refer interested readers to section F for more ablation studies about training procedures and set-input models.

| set size | 2 | 5 | 10 | 15 |
|---|---|---|---|---|
| 1-shot 5-way test | | | | |
| NC | $75.29 \pm 1.33$ | $\mathbf{77.84 \pm 1.16}$ | $74.88 \pm 1.36$ | $75.25 \pm 1.29$ |
| LR | $73.72 \pm 1.33$ | $\mathbf{77.56 \pm 1.16}$ | $73.84 \pm 1.29$ | $74.00 \pm 1.27$ |
| LR+LA | $75.27 \pm 1.28$ | $\mathbf{79.16 \pm 1.09}$ | $74.41 \pm 1.31$ | $74.92 \pm 1.26$ |
| 5-shot 5-way test | | | | |
| NC | $90.62 \pm 0.56$ | $90.35 \pm 0.50$ | $90.62 \pm 0.57$ | $\mathbf{90.83 \pm 0.55}$ |
| LR | $89.41 \pm 0.63$ | $\mathbf{90.91 \pm 0.47}$ | $89.80 \pm 0.59$ | $89.63 \pm 0.60$ |
| LR+LA | $90.46 \pm 0.58$ | $\mathbf{91.68 \pm 0.50}$ | $90.29 \pm 0.57$ | $90.46 \pm 0.56$ |

Table 4: Abation on input set size; models tested on LC dataset; average F1 and CI are reported.

## 4 RELATED WORK

**Weakly supervised learning** The concept of weakly supervised learning is introduced as a means to alleviate the annotation bottleneck in the training of machine learning models. There has been a large body of existing work in learning with only weak labels. A comprehensive survey about weakly supervised learning is provided in Zhou (2018); Zhang et al. (2022). We study a novel form of weak supervision which is provided by set-level coarse-grained labels. Among weakly supervised learning methods, Robinson et al. (2020) studied the generalization properties of weakly supervised learning and proposed a generic learning algorithm that can learn from weak and strong labels and can be proved to achieve a fast rate. The authors consider a different setting where each instance has a weak label and a strong label, and the strong label predictor learns to predict the strong labels from the instances and their corresponding embeddings learned with weak labels. We consider the setting where we have some coarse-grained labels of some sets, rather than instances and the downstream classifiers only use the learned embeddings to train and test on the downstream tasks.

**Multiple-instance learning for WSIs** WSI classification and regression are formulated based on multiple-instance learning (MIL) (Campanella et al., 2019; Xu et al., 2022; Ilse et al., 2018; Sharma et al., 2021; Hashimoto et al., 2020; Shao et al., 2021; Yao et al., 2020; Lu et al., 2021b;a; Chen et al., 2021b; Li et al., 2021; Chen et al., 2021a; Myronenko et al., 2021; Xiang & Zhang, 2022; Javed et al., 2022). These MIL models employ two procedures: i) feature extraction for patches cropped from a WSI and ii) aggregation of features from the same WSI. ImageNet pretrained backbones, self-supervised backbones pretrained on histopathology images, or backbones fine-tuned during training are used to extract features from patches. Deep attention pooling, graph neural networks, or sequence models, adapted for WSIs, are used for feature aggregation. In this paper, we consider a different problem setting where we enhance patch-level classification with related set-level labels. In the application of histopathology images, line 2 of our generic algorithm can be instantiated with any MIL models that have the backbones with trainable modules to extract patch-level features, e.g., Ilse et al. (2018). A complete comparison of MIL models for WSIs is out of the scope of this paper.

**Learning from coarsely-labeled data** Another related line of research is Phoo & Hariharan (2021), where the authors assume a taxonomy of classes with two levels, i.e., a set of fine-grained classes that are more challenging to annotate and a set of coarse-grained classes that are easier to annotate. In our paper, we do not assume a taxonomy of classes for the coarse-grained and fine-grained labels. The coarse-grained and fine-grained labels are closely related in a more sophisticated way. Besides, the inputs that are fed to models to predict the coarse-grained or fine-grained labels are different, i.e., set input for coarse-grained labels and instances for fine-grained labels.

## 5 CONCLUSION AND DISCUSSION

**Summary** We introduce FACILE, a representation learning framework that leverages coarse-grained labels for model training and enhances model performance for downstream tasks. Our theoretical analysis highlights the significant potential of leveraging set-level coarse-grained labels to benifit the learning process of fine-grained label prediction tasks. To demonstrate the effectiveness of FACILE, we conduct pretraining on CIFAR-100-based datasets and two large public histopathology datasets using coarse-grained labels and evaluate our model on a diverse collection of datasets with fine-grained labels.

**Limitation and future work** In this paper, we consider a novel problem setting where we enhance downstream fine-grained label classification with easily available coarse-grained labels and propose a generic algorithm that contains two supervised learning steps. As an initial step towards addressing this issue, it's important to recognize that the separate utilization of loosely related coarse-grained labels and fine-grained labels can be inefficient. Besides, the pretraining of our proposed algorithm could be expensive given the large amount of coarse-grained data and the nature of set-input models. We are motivated to explore methods for efficient and robust representation learning with coarse-grained labels. Specifically, we are interested in

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

\end{aligned}
$$