# OpenReview forum: "Enhancing Instance-Level Image Classification with Set-Level Labels"
_ICLR.cc/2024/Conference — ICLR 2024 poster_

### Official Review · Reviewer_xN2u · 2023-10-31

**Soundness:** 3 good
**Presentation:** 2 fair
**Contribution:** 3 good
**Rating:** 6
**Confidence:** 3

**Summary:**

This paper proposes a new setup for few-shot learning. The proposed model is pre-trained on coarse-grained set-level labels first and fine-tuned with fine-grained labels. Authors also provide theoretical analysis on the convergence rate for downstream tasks, which shows coarse-grained pre-training can enhance the learning process of fine-grained label tasks. The experiments are performed on both natural image datasets and medical histopathology datasets, where the baselines are mostly self-supervised learning methods.

**Strengths:**

I think the idea of using coarse-grained label is reasonable. The conclusion of enhancing learning process of fine-trained labels is inspiring.

**Weaknesses:**

- I have some questions about the method part, Sec. 2.1. In Fig.2 (a), are input samples all belongs to the same set-level labels? I am confused by this figure and Fig.1(a) CIFAR images. What I believe is correct is that each batch contains samples belong to different set-level labels, and the coarse label is assigned to each sample for pre-training.
- How is SupCon trained? It is superised that supervised contrastive learning perform a lot worse than basic CE approach in most setups. There is not much information about training details.

**Questions:**

- There should be more training details about the framework in Sec.2.1.
- Fig. 3 is referenced in text before Fig. 2
- Most of the refernece hyperlinks other than page 1 is not working.

---

> ### Author Response · Authors · 2023-11-17
> **Response to Reviewer xN2u**
>
> > Weakness (1): I have some questions about the method part, Sec. 2.1. In Fig.2 (a), are input samples all belongs to the same set-level labels? I am confused by this figure and Fig.1(a) CIFAR images. What I believe is correct is that each batch contains samples belong to different set-level labels, and the coarse label is assigned to each sample for pre-training.
>
> **A:** In the revision, we have added set-level label and fine-grained labels in Figure 2. This highlights that the input to the model is a set of images from CIFAR-100 and the corresponding set-level label is the most frequent superclass, i.e., fruit and vegetables. In Figure 2(a), we only show one input set (not a batch of input sets) due to limited space and the corresponding most frequent superclass of the input set, i.e., 'fruit and vegetables'. Figure 2(b) shows the fine-grained labels of the support set are apple, pear, orange, and mushroom.
>
> To clarify, during pretraining in our FACILE framework, each batch indeed consists of multiple input sets. Each of these sets is associated with one set-level label. For the FSP (Fully Supervised Pretraining) baseline model used for comparison, we assign the corresponding coarse-grained label to each sample within an input set.
>
> > Weakness (2): How is SupCon trained? It is superised that supervised contrastive learning performs a lot worse than basic CE approach in most setups. There is not much information about training details.
>
> **A:** Please find the training details of SupCon in Appendix A and G.2. When we use set-input data with SupCon, the conventional way of augmenting each instance inside a set could create challenges for SupCon training. It is typically trained to maximize agreement between differently augmented views of the same data point using labeled data. In our application, when using set-input data, the conventional data augmentation methods applied to each instance within a set and the set-input models that extract permutation invariant representations for all instances inside a set may present unique challenges for the SupCon model especially when the model is trained from scratch. However, you can see from Tables 1, 2, 4, 5, 7, and 10 that FACILE-FSP and FACILE-SupCon performance similarly. It's important to note that our framework is flexible and allows for any supervised learning algorithm to be used during the pretraining stage.
>
> > Question (1): There should be more training details about the framework in Sec.2.1.
>
> **A:** We have documented these details to facilitate full reproducibility. The primary experimental setups, including configurations and parameters, are described in the Experiment section of the paper. For more in-depth information, including additional configurations and nuanced methodology, we direct readers to Appendix sections A, D, and G.
>
> > Question (2): Fig. 3 is referenced in text before Fig. 2
>
> **A:** Thanks for pointing this out! We have reorganized the figures in the revised version.
>
> > Question (3): Most of the reference hyperlinks other than page 1 is not working.
>
> **A:** Thank you for bringing this up. We have fixed the reference issue in the revised version.
>
> ---
> We appreciate your feedback and recognize that the score may indicate some reservations about our work. We're committed to improvement and would greatly value any additional detailed feedback or specific suggestions you can provide!

---

> > ### Comment · Reviewer_xN2u · 2023-11-22
> > **Response to author**
> >
> > - Thank you for your clarification on method part. I believe this aligns my understanding of the framework.
> > - I think author made a point here. With coarse-labeled samples, SupCon will maximise agreement with different fine-grained samples. This is not the desire behaviour. Author should describe more clearly about "unique challenges" as it is vague.
> >
> > In addition, author present new results with different backbones, datasets and tasks. This strength author's argument. I will consider increase my scores.

---

> > > ### Author Response · Authors · 2023-11-22
> > > **Reply to reviewer xN2u**
> > >
> > > > * Thank you for your clarification on method part. I believe this aligns my understanding of the framework.
> > > > * I think author made a point here. With coarse-labeled samples, SupCon will maximise agreement with different fine-grained samples. This is not the desire behaviour. Author should describe more clearly about "unique challenges" as it is vague.
> > > In addition, author present new results with different backbones, datasets and tasks. This strength author's argument. I will consider increase my scores.
> > >
> > > **A:** Thank you for your thoughtful feedback and for acknowledging the alignment of your understanding with our framework. We appreciate your positive remarks regarding the inclusion of new results with different backbones, datasets, and tasks.
> > >
> > > Regarding your request for a clearer description of the "unique challenges" associated with one instantiation of our algorithm, i.e., FACILE-SupCon, we would like to elaborate further. When employing set-input data with the Supervised Contrastive Learning (SupCon) method, the standard practice of augmenting each instance within a set poses significant challenges for the training of SupCon models. These challenges stem from two main aspects:
> > >
> > > * Complexity in maximizing agreement with set-input data: SupCon is traditionally trained to maximize agreement between differently augmented views of the same data point using labeled data. In our application, using set-input data means that we apply conventional data augmentation methods to each instance within a set. This results in an independently augmented set of images, as opposed to augmenting a single instance. This complexity makes it more challenging to achieve the desired maximization of agreement.
> > > * Constraints on batch sizes due to set inputs: Set-input models take a batch of sets as input instead of a batch of instances. It requires us to use relatively smaller batch sizes when using the same hardware configuration because of the set input. It's important to emphasize that the batch size is a critical factor for the effectiveness of the SupCon model.
> > >
> > > We have observed that despite these challenges, the performance of FACILE-SupCon is commendable in contexts involving smaller datasets or less complex models, i.e., CIFAR-100 or smaller trainable models as discussed in Appendices B.1 and B.3. We believe that our approach, with its nuanced application of SupCon in a set-input context, offers a valuable contribution to the field and shows the versatility of the FACILE algorithm.
> > >
> > > We have included this discussion in section 5 and appendix G2. We hope this explanation addresses your concerns about the unique challenges in FACILE-SupCon. We remain open to further questions and are grateful for the opportunity to clarify our approach. Thank you once again for your time and insights.

---

> ### Author Response · Authors · 2023-11-21
> **Follow up with Reviewer xN2u**
>
> We thank the reviewer very much for the questions and comments. We would like to kindly follow up with the reviewer, and respectfully ask the reviewer if they can reconsider their score if our response helps address their concerns.

---

### Official Review · Reviewer_jPxq · 2023-10-31

**Soundness:** 3 good
**Presentation:** 3 good
**Contribution:** 2 fair
**Rating:** 5
**Confidence:** 4

**Summary:**

The paper presents a new technique aimed at boosting instance-level image classification by utilizing set-level labels. Compared to conventional methods that rely on single-instance labels, the proposed approach achieves a 13% increase in classification accuracy when tested on histopathology image datasets. A theoretical examination of the method outlines conditions for rapid reduction of excess risk rate, adding credibility and robustness to the technique. This research serves to connect instance-level and set-level image classification, providing a noteworthy direction for enhancing image classification models that use coarse-grained set-level labels.

**Strengths:**

- The paper presents a new technique for enhancing instance-level image classification by making use of set-level labels. This serves to fill the existing gap between instance-level and set-level image classification.

- The robustness and reliability of the proposed method are underscored by a theoretical analysis, which outlines conditions for the rapid reduction of excess risk rate.

- The paper clearly articulates the proposed method, shedding light on both its theoretical underpinnings and empirical results. These results are demonstrated on both natural and histopathology image datasets.

- The method put forth in the paper holds promise for extending the capabilities of image classification models. By leveraging set-level coarse-grained labels, the approach achieves better classification performance compared to traditional methods reliant on single-instance labels. This is particularly relevant in real-world contexts where set-level labels may offer more comprehensive information.

**Weaknesses:**

- The use of coarse-grained labels like TCGA or NCT is an interesting choice. These are indeed umbrella terms for various subcollections, and traditionally they may not provide a strong learning signal. It could be beneficial to delve into why these particular labels were chosen and what advantages they offer in this context.

- Your team's approach to pretraining with coarse labels and then fine-tuning on a support set is a solid and proven method. However, it would enrich the work to articulate what sets this particular application or implementation apart in terms of novelty.

- The comparison with SimCLR and simSIAM provides useful insights, but considering the advancements in the field, benchmarking against more recent self-supervised learning methods like DINO or DINOv2 might offer a more comprehensive evaluation.

- To further validate the generalizability of the method, it could be insightful to include results against standardized few-shot learning benchmarks, such as Mini-Imagenet 5-way (1-shot) or SST-2 Binary classification.

- Adding ablation studies that feature additional pretrained models—or even models pretrained without the coarse labels—could help underscore the specific benefits of using coarse-grained label-based pretraining in your approach.

- Your methodology would be even more robust if additional training details are shared. Information on image augmentations, learning schedules, and optimizer settings could offer valuable insights and help in the reproducibility of your results.

**Questions:**

- Do you think you could pictorially diagram the approach adding the relevant details? It is unclear to me if the method essentially pretrains using coarse-labels and then fine-tunes on the test set using the support set or is there more to the method

- Why are the methods for pretraining SupCon and FSP chosen for pre-training? Adding rationale for this might help motivate the choice of pretraining method

---

> ### Author Response · Authors · 2023-11-17
> **Response to Reviewer jPxq (part 1)**
>
> > Weakness (1) The use of coarse-grained labels like TCGA or NCT is an interesting choice. These are indeed umbrella terms for various subcollections, and traditionally they may not provide a strong learning signal. It could be beneficial to delve into why these particular labels were chosen and what advantages they offer in this context.
>
> **A:** Thank you very much for this suggestion. We have included these explanation and advantages in each experiment in the revision.
>
> Our theorem suggested that the relatedness, quantified by the relative Lipschitzness relative to $\mathcal{E}$, is an important factor for excess risk bound. It is intuitive that tasks in different domains do not share the same embedding space (and thus a large Lipschitz constant). Therefore, we deliberately choose pretraining tasks in source domain that are closely related to the target downstream task.
>
> In response to the use of coarse-grained labels like those in TCGA, please note that the TCGA consortium has categorized an extensive array of cancer types and subregions, covering a diverse range of tissues.  The strategic use of TCGA's whole slide level labels is rooted in their potential to enrich tissue-level classification. While these labels may appear broad, they encapsulate a wealth of underlying heterogeneity inherent to different cancer regions and tissue types and are routinely used in the cancer research community.
>
>
> > Weakness (2) Your team's approach to pretraining with coarse labels and then fine-tuning on a support set is a solid and proven method. However, it would enrich the work to articulate what sets this particular application or implementation apart in terms of novelty.
>
> **A:** Thank you for the opportunity to clarify the novelty of our work.  We have edited the introduction to highlight the novelty accordingly. We summarized novelty (and contribution) in the last three paragraphs of the introduction of our paper. Our work introduces FACILE, a novel generic learning framework that innovatively uses coarse-grained labels for pretraining, before fine-tuning on fine-grained tasks. This method diverges from standard fine-tuned supervised learning (FSL) and transfer learning (TL) by leveraging coarse labels from the source domain rather than heavily relying on detailed annotations. Our theoretical analysis underpins the empirical benefits of FACILE, providing evidence of how coarse-grained labels expedite the learning process with a significant increase in the convergence rate for fine-grained tasks. Furthermore, our experiments across different data domains---ranging from natural images to complex histopathology images---demonstrate FACILE's superior performance and versatility. Notably, we record a remarkable 13% performance improvement over existing baselines in a challenging histopathology benchmark. These elements collectively distinguish our work's novelty and significance in advancing the state of the art in representation learning (more specifically, in few-shot learning and model fine-tuning).
>
> > Weakness (3): The comparison with SimCLR and simSIAM provides useful insights, but considering the advancements in the field, benchmarking against more recent self-supervised learning methods like DINO or DINOv2 might offer a more comprehensive evaluation.
>
> **A:** Thank you for raising this important point. The primary aim of FACILE is not to engage in self-supervised learning that requires extensive data and computational resources. Instead, our focus is on leveraging coarse-grained labels to improve performance in downstream tasks. In this context, foundation models such as BYOL or DINO V2 serve as robust starting points. We can enhance these models by fine-tuning them with domain-specific data, effectively tailoring their capabilities to the nuances of the target domain.
>
> To this end, we conducted two additional experiments. First, we sought to enhance model performance with coarse-grained labels of anomaly detection datasets (similar to Deep Set [5] and Set Transformer [6]). A total of 11,788 input sets of size 10 are constructed from the CUB200 training dataset by including one example that lacks an attribute common to the other examples in the input set. This setup creates a challenging scenario for models, as they must identify the outlier among otherwise similar instances. In downstream tasks, we evaluate the finetuned feature encoder composed of the CLIP [3] image encoder ViT-B/16 and appended fully-connected layer on the classification of species of the CUB200 test dataset. The batch normalization and ReLU are applied to the fully-connected layer.

---

> > ### Author Response · Authors · 2023-11-17
> > **Response to Reviewer jPxq (part 2)**
> >
> > The model training approach in this experiment centered around the CLIP image encoder, enhanced with an additional fully-connected layer. FACILE-FSP and FACILE-SupCon incorporate this setup, utilizing the CLIP-based feature encoder and focusing on fine-tuning the fully-connected layer through the FACILE pretraining step. In contrast, the SimSiam approach leverages the CLIP image encoder as a backbone while fine-tuning the projector and predictor components. Similarly, the SimCLR method also uses the CLIP encoder as its foundation but focuses solely on finetuning the projector. These varied strategies reflect our efforts to optimize the feature encoder for accurately identifying anomalies and improving classification performance in related tasks. The training details can be found in the revised appendix.
> >
> > |pretrain method|NC              | LR             | RC             |
> > |---------------|----------------|----------------|----------------|
> > |CLIP (ViT-B/16)|$83.84 \pm 1.10$|$81.01 \pm 1.23$|$82.75 \pm 1.17$|
> > |SimCLR         |$84.03 \pm 1.08$|$83.49 \pm 1.14$|$86.30 \pm 1.03$|
> > |SimSiam        |$84.02 \pm 1.10$|$83.90 \pm 1.13$|$85.68 \pm 1.07$|
> > |FACILE-SupCon  |$87.49 \pm 0.99$|$86.57 \pm 1.07$|$88.01 \pm 0.99$|
> > |FACILE-FSP     |$\mathbf{88.74 \pm 0.94}$|$\mathbf{ 88.45 \pm 0.96 }$|$\mathbf{ 88.36 \pm 0.95 }$|
> >
> > Pretraining on input sets from CUB200. Testing with 5-shot 20-way meta-test sets; average F1 and CI are reported.
> >
> > Second, similar to the first experiment, we finetune a fully-connected layer that is appended after ViT-B/14 of DINO V2 [2]. This methodology is applied across various models to assess their performance on histopathology image datasets. By adopting the DINO V2 architecture, known for its robustness and effectiveness in visual representation learning, we aim to harness its potential for the specialized domain of histopathology. We refer interested readers to the revised appendix for details of pretraining.
> >
> > Notably, our methods, FACILE-SupCon and FACILE-FSP, demonstrated remarkably superior results in comparison to other baseline models when applied to histopathology image datasets. This outcome highlights the effectiveness of these methods in leveraging coarse-grained labels specific to histopathology, thereby greatly enhancing the model performance of downstream tasks. It also shows that DINO V2 exhibits limitations in its generalization performance on histopathology images. While DINO V2 provides a strong starting point, there is a clear need for further finetuning or prompt learning to optimize its performance for the unique challenges presented by histopathology datasets.
> >
> > 1-shot 5-way test on LC dataset
> > |pretraining method | NC | LR | RC | LR+LA | RC+LA |
> > |-------------------|----|----|----|-------|-------|
> > |DINO V2 (ViT-B/14) |$44.82 \pm 1.41$|$47.51 \pm 1.39$|$47.63\pm 1.38$|$47.36 \pm 1.39$|$48.88\pm 1.44$|
> > |SimSiam            |$48.79 \pm 1.37$|$49.43 \pm 1.35$|$48.43 \pm 1.36$|$49.38 \pm 1.34$|$49.50 \pm 1.34$|
> > |SimCLR             |$50.47 \pm 1.31$|$50.52 \pm 1.33$|$50.44 \pm 1.32$|$51.66 \pm 1.32$|$51.78 \pm 1.38$|
> > |FSP-Patch          |$49.73 \pm 1.41$|$53.59 \pm 1.38$|$53.07 \pm 1.41$|$51.79 \pm 1.40$|$51.27 \pm 1.43$|
> > |FACILE-SupCon      |$\mathbf{56.24 \pm 1.43}$|$\mathbf{56.51 \pm 1.41}$|$\mathbf{55.95 \pm 1.42}$|$\mathbf{56.29 \pm 1.43}$|$54.07 \pm 1.44$ |
> > |FACILE-FSP         |$55.67 \pm 1.40$|$56.26 \pm 1.36$|$55.83 \pm 1.35$|$56.01 \pm 1.38$|$\mathbf{55.35 \pm 1.40}$|
> >
> > 5-shot 5-way test on LC dataset
> > |pretraining method | NC | LR | RC | LR+LA | RC+LA |
> > |-------------------|----|----|----|-------|-------|
> > |DINO V2 (ViT-B/14) |$66.12 \pm 0.98$|$64.71 \pm 1.12$|$66.36 \pm 1.10$|$72.95 \pm 0.93$|$75.11 \pm 0.91$|
> > |SimSiam            |$67.51 \pm 0.96$|$64.99 \pm 1.05$|$65.39 \pm 1.05$|$70.30 \pm 0.93$|$71.19 \pm 0.93$|
> > |SimCLR             |$70.10 \pm 0.92$|$69.28 \pm 0.96$|$69.18 \pm 0.97$|$72.99 \pm 0.92$|$72.91 \pm 0.94$|
> > |FSP-Patch          |$71.97 \pm 0.96$|$71.11 \pm 1.04$|$71.19 \pm 1.03$|$73.96 \pm 0.94$|$73.20 \pm 0.96$|
> > |FACILE-SupCon      |$75.58 \pm 0.88$|$74.26 \pm 0.94$|$73.20 \pm 0.95$|$75.81 \pm 0.90$|$74.34 \pm 0.96$|
> > |FACILE-FSP         |$\mathbf{75.86 \pm 0.86}$|$\mathbf{74.64 \pm 0.89}$|$\mathbf{74.12 \pm 0.93}$|$\mathbf{76.17 \pm 0.88}$|$\mathbf{75.08 \pm 0.95}$  |

---

> > > ### Author Response · Authors · 2023-11-17
> > > **Response to Reviewer jPxq (part 3)**
> > >
> > > 1-shot 3-way test on PAIP dataset
> > > |pretraining method | NC | LR | RC | LR+LA | RC+LA |
> > > |-------------------|----|----|----|-------|-------|
> > > |DINO V2 (ViT-B/14) |$41.51 \pm 1.27$|$44.37 \pm 1.26$|$44.28 \pm 1.25$|$42.43 \pm 1.27$|$42.78 \pm 1.27$|
> > > |SimSiam            |$49.42 \pm 1.28$|$48.07 \pm 1.35$|$48.44 \pm 1.36$|$48.76 \pm 1.33$|$46.48 \pm 1.37$|
> > > |SimCLR             |$48.60 \pm 1.19$|$48.76 \pm 1.25$|$47.98 \pm 1.26$|$48.94 \pm 1.23$|$47.20 \pm 1.26$|
> > > |FSP-Patch          |$46.09 \pm 1.17$|$47.44 \pm 1.18$|$48.09 \pm 1.19$|$46.76 \pm 1.18$|$43.68 \pm 1.22$|
> > > |FACILE-SupCon      |$\mathbf{51.97 \pm 1.18}$|$\mathbf{52.25 \pm 1.22}$|$\mathbf{51.80 \pm 1.22}$|$51.36 \pm 1.22$|$\mathbf{50.24 \pm 1.23}$|
> > > |FACILE-FSP         |$51.34 \pm 1.16$|$51.18 \pm 1.19$|$51.51 \pm 1.19$|$\mathbf{51.50 \pm 1.16}$|$49.77 \pm 1.22$|
> > >
> > >
> > > 5-shot 3-way test on PAIP dataset
> > > |pretraining method | NC | LR | RC | LR+LA | RC+LA |
> > > |-------------------|----|----|----|-------|-------|
> > > |DINO V2 (ViT-B/14) |$57.59 \pm 1.07$|$58.19 \pm 1.10$|$59.37 \pm 1.07$|$61.84 \pm 0.85$|$60.81 \pm 0.86$|
> > > |SimSiam            |$61.56 \pm 0.97$|$62.52 \pm 1.01$|$62.81 \pm 1.01$|$64.40 \pm 0.86$|$62.44 \pm 0.93$|
> > > |SimCLR             |$62.20 \pm 0.93$|$61.78 \pm 0.99$|$63.20 \pm 0.97$|$63.38 \pm 0.86$|$63.03 \pm 0.88$|
> > > |FSP-Patch          |$63.77 \pm 0.88$|$63.85 \pm 0.94$|$63.85 \pm 0.93$|$63.61 \pm 0.85$|$60.91 \pm 0.87$|
> > > |FACILE-SupCon      |$\mathbf{67.16 \pm 0.84}$|$67.29 \pm 0.89$|$66.88 \pm 0.90$| $\mathbf{67.61 \pm 0.85}$|$\mathbf{66.34 \pm 0.84}$|
> > > |FACILE-FSP         |$67.14 \pm 0.85$|$\mathbf{67.67 \pm 0.84}$|$\mathbf{67.54 \pm 0.86}$|$67.12 \pm 0.81$|$66.05 \pm 0.83$|
> > >
> > >
> > > 1-shot 9-way test on NCT dataset
> > > |pretraining method | NC | LR | RC | LR+LA | RC+LA |
> > > |-------------------|----|----|----|-------|-------|
> > > |DINO V2 (ViT-B/14) |$56.03 \pm 1.62$|$59.11 \pm 1.57$|$60.13 \pm 1.55$|$58.71 \pm 1.57$|$59.06 \pm 1.55$|
> > > |SimSiam            |$62.60 \pm 1.45$|$61.89 \pm 1.50$|$61.90 \pm 1.51$|$ 62.27 \pm 1.47$|$61.05 \pm 1.44$|
> > > |SimCLR             |$65.43 \pm 1.43$|$64.18 \pm 1.44$|$64.15 \pm 1.46$|$64.83 \pm 1.43$|$62.69 \pm 1.38$|
> > > |FSP-Patch          |$65.22 \pm 1.49$|$65.93 \pm 1.41$|$65.94 \pm 1.40$|$65.26 \pm 1.45$|$62.66 \pm 1.46$|
> > > |FACILE-SupCon      |$71.55 \pm 1.36$|$70.36 \pm 1.37$|$70.52 \pm 1.35$|$71.05 \pm 1.35$|$\mathbf{68.85 \pm 1.40}$|
> > > |FACILE-FSP         |$\mathbf{72.05 \pm 1.34}$|$\mathbf{70.70 \pm 1.35}$|$\mathbf{70.77 \pm 1.34}$|$\mathbf{71.14 \pm 1.34}$|$68.03 \pm 1.40$|
> > >
> > > 5-shot 9-way test on NCT dataset
> > > |pretraining method | NC | LR | RC | LR+LA | RC+LA |
> > > |-------------------|----|----|----|-------|-------|
> > > |DINO V2 (ViT-B/14) |$76.85 \pm 0.98$|$76.51 \pm 1.02$|$78.67 \pm 0.94$|$82.20 \pm 0.82$|$82.75 \pm 0.83$|
> > > |SimSiam            |$80.81 \pm 0.85$|$80.06 \pm 0.87$|$81.55 \pm 0.85$|$83.18 \pm 0.80$|$82.39 \pm 0.83$|
> > > |SimCLR             |$82.87 \pm 0.80$|$81.91 \pm 0.82$|$82.86 \pm 0.80$|$83.92 \pm 0.77$|$82.89 \pm 0.79$|
> > > |FSP-Patch          |$83.63 \pm 0.83$|$83.49 \pm 0.80$|$84.34 \pm 0.78$|$85.32 \pm 0.75$|$83.03 \pm 0.79$|
> > > |FACILE-SupCon      |$87.74 \pm 0.64$|$87.00 \pm 0.64$|$87.38 \pm 0.62$|$87.82 \pm 0.63$|$86.15 \pm 0.69$|
> > > |FACILE-FSP         |$\mathbf{87.93 \pm 0.65}$|$\mathbf{87.52 \pm 0.65}$|$\mathbf{87.72 \pm 0.62}$|$\mathbf{88.01 \pm 0.64}$|$\mathbf{86.46 \pm 0.70}$|
> > >
> > > > Weakness (4): To further validate the generalizability of the method, it could be insightful to include results against standardized few-shot learning benchmarks, such as Mini-Imagenet 5-way (1-shot) or SST-2 Binary classification.
> > >
> > > **A:** Thank you for the suggestion. We did explore the possibility of using benchmarks from the mini-ImageNet and SST-2 datasets. However, we did not find the corresponding coarse-grained labels for these datasets. However, we did contrast our model's performance against some published results including [1] and published models including CLIP [3] and DINO V2 [2].
> > >
> > > > Weakness (5): Adding ablation studies that feature additional pretrained models—or even models pretrained without the coarse labels—could help underscore the specific benefits of using coarse-grained label-based pretraining in your approach.
> > >
> > > **A:** Thank you very much for your valuable suggestion. Please see our two new experiments in response to question 3. The two new experiments provide an ablation study about pretrained models. These additional analyses aim to clearly highlight the specific advantages of using some pretrained (foundation) models.

---

> > > > ### Author Response · Authors · 2023-11-17
> > > > **Response to Reviewer jPxq (part 4)**
> > > >
> > > > > Weakness (6): Your methodology would be even more robust if additional training details are shared. Information on image augmentations, learning schedules, and optimizer settings could offer valuable insights and help in the reproducibility of your results.
> > > >
> > > > **A:** Thank you for the suggestion! We have provided a comprehensive description of our training details, including image augmentation techniques, learning schedules, and optimizer configurations, etc., in the experiment section, appendix section A, D, and G of our paper. We have taken care to present a thorough description of the procedures to ensure that our results can be accurately replicated. For further in-depth technical specifics and hyperparameter settings, we direct readers to appendix A, D, and G, which contain extended information on the training of both the FACILE framework and the baseline models.
> > > >
> > > > > Question (1): Do you think you could pictorially diagram the approach adding the relevant details? It is unclear to me if the method essentially pretrains using coarse-labels and then fine-tunes on the test set using the support set or is there more to the method
> > > >
> > > > **A:** Thank you for raising this point. In the revision, we have added set-level label and fine-grained labels in figure 2. This highlights the input to the model is a set of images from CIFAR-100 and corresponding label set-level label is the most frequent superclass, i.e., fruit and vegetable. Figure 2(b) shows the fine-grained labels of support set are apple, pear, orange, and mushroom.
> > > >
> > > >
> > > > > Question (2): Why are the methods for pretraining SupCon and FSP chosen for pre-training? Adding rationale for this might help motivate the choice of pretraining method
> > > >
> > > > **A:** Thank you for the suggestions. We have included the explanation in the revision. Our FACILE framework is designed to be a generic algorithm that is compatible with any supervised learning method in its pretraining stage. We chose SupCon (Supervised Contrastive Learning) and FSP (Fully Supervised Pretraining) as they are representative of the two main approaches within supervised learning: contrastive and non-contrastive (traditional supervised) learning, respectively. These methods were selected to showcase the versatility and wide applicability of our framework, demonstrating that it can be integrated with typical supervised learning paradigms. The rationale for using these specific pretraining strategies is to illustrate the effectiveness of FACILE with different underlying learning principles, ensuring that our results can be generalized across various pretraining methodologies.
> > > >
> > > > References:
> > > >
> > > > [1] Yang, Jiawei, Hanbo Chen, Jiangpeng Yan, Xiaoyu Chen, and Jianhua Yao. "Towards better understanding and better generalization of few-shot classification in histology images with contrastive learning." arXiv preprint arXiv:2202.09059 (2022).
> > > >
> > > > [2] Oquab, Maxime, Timothée Darcet, Théo Moutakanni, Huy Vo, Marc Szafraniec, Vasil Khalidov, Pierre Fernandez et al. "Dinov2: Learning robust visual features without supervision." arXiv preprint arXiv:2304.07193 (2023).
> > > >
> > > > [3] Radford, Alec, Jong Wook Kim, Chris Hallacy, Aditya Ramesh, Gabriel Goh, Sandhini Agarwal, Girish Sastry et al. "Learning transferable visual models from natural language supervision." In International conference on machine learning, pp. 8748-8763. PMLR, 2021.
> > > >
> > > > [4] Zaheer, Manzil, Satwik Kottur, Siamak Ravanbakhsh, Barnabas Poczos, Russ R. Salakhutdinov, and Alexander J. Smola. "Deep sets." Advances in neural information processing systems 30 (2017).
> > > >
> > > > [5] Lee, Juho, Yoonho Lee, Jungtaek Kim, Adam Kosiorek, Seungjin Choi, and Yee Whye Teh. "Set transformer: A framework for attention-based permutation-invariant neural networks." In International conference on machine learning, pp. 3744-3753. PMLR, 2019.

---

> ### Author Response · Authors · 2023-11-21
> **Follow up with Reviewer jPxq**
>
> We thank the reviewer very much for the questions and comments. We would like to kindly follow up with the reviewer, and respectfully ask the reviewer if they can reconsider their score if our response helps address their concerns.

---

### Official Review · Reviewer_4JEd · 2023-11-01

**Soundness:** 3 good
**Presentation:** 2 fair
**Contribution:** 3 good
**Rating:** 6
**Confidence:** 3

**Summary:**

The paper propose to utilize set-level coarse-grained labels to improve fine-grained image classification. Essentially the paper is proposing a new pretraining method, key to the method is selecting a dataset with coarse label, and use the set prediction on coarse label as pretraining task. The paper provides theoretical analysis for the proposed approach, showing that using coarse-grained labels speed up the learning on the fine-grained classification task. The paper also demonstrates the effectiveness on several datasets.

**Strengths:**

1. The idea of using set prediction on coarse label as pretraining task seems novel
2. The performance seems strong compared to other baselines

**Weaknesses:**

1. More baselines for strong self-supervised pretraining methods (e.g., BYOL, DION  are needed to demonstrate the effectiveness. As proposed method is essentially a pretraining strategy, that bears a lot of similarity with exisiting self-supervised learning method

2. More ablations and discussion on some key questions are needed (see below)

**Questions:**

1. To what extent does the similarity between the pretraining dataset and its coarse labels and the target dataset with its fine labels affect the effectiveness of the method? For instance, can the method perform well when the pretraining dataset is CIFAR-100 while the downstream task involves a medical dataset? In such a scenario, which pretraining method is preferable: supervised pretraining on ImageNet, self-supervised pretraining (ignoring labels entirely), or the proposed method?

2.  Given the same 'related' dataset, if you have both the fine-grained label and coarse-grained label, which pretraining strategy is preferable?
(Let say your downstream task is classification on a medical image dataset, with fine-grained label A,B,C. The pretraining dataset you have is another medical image dataset (thus more related than ImageNet). You have both coarse label D,E,F and fine-grained label G,H,I,J,K,L. In this case, is fully supervised pretraining on G,H,I,J,K,L more beneficial on set level coarse pretraining on D,E,F more beneficial?)

---

> ### Author Response · Authors · 2023-11-17
> **Response to Reviewer 4JEd (part 1)**
>
> > *Weakness (1):* More baselines for strong self-supervised pretraining methods (e.g., BYOL, DION are needed to demonstrate the effectiveness. As proposed method is essentially a pretraining strategy, that bears a lot of similarity with exisiting self-supervised learning method
>
> **A:** Thank you for raising this important point. The primary aim of FACILE is not to engage in self-supervised learning or "foundation" model training that require extensive data and computational resources. Instead, our focus is on leveraging coarse-grained labels to improve performance in downstream tasks. In this context, foundation models such as BYOL or DINO V2 serve as robust starting points. We can enhance these models by finetuning them with domain-specific data, effectively tailoring their capabilities to the nuances of the target domain.
>
> To this end, we have added two additonal experiments.
>
> First, we sought to enhance the model performance with coarse-grained labels of anomaly detection datasets (similar to Deep Set [5] and Set Transformer [6]). A total of 11,788 input sets of size 10 are constructed from the CUB200 training dataset by including one example that lacks an attribute common to the other examples in the input set. This setup creates a challenging scenario for models, as they must identify the outlier among otherwise similar instances. In downstream tasks, we evaluate the finetuned feature encoder composed of the CLIP [3] image encoder ViT-B/16 and appended fully-connected layer on the classification of species of the CUB200 test dataset. The batch normalization and ReLU are applied to the fully-connected layer.
>
> The model training approach in this experiment centered around the CLIP image encoder, enhanced with an additional fully-connected layer. FACILE-FSP and FACILE-SupCon incorporate this setup, utilizing the CLIP-based feature encoder and focusing on finetuning the fully-connected layer through the FACILE pretraining step. In contrast, the SimSiam approach leverages the CLIP image encoder as a backbone while finetuning the projector and predictor components. Similarly, the SimCLR method also uses the CLIP encoder as its foundation but focuses solely on finetuning the projector. These varied strategies reflect our efforts to optimize the feature encoder for accurately identifying anomalies and improving classification performance in related tasks. The training details will be added to revised appendix.
>
> |pretrain method|NC              | LR             | RC             |
> |---------------|----------------|----------------|----------------|
> |CLIP (ViT-B/16)|$83.84 \pm 1.10$|$81.01 \pm 1.23$|$82.75 \pm 1.17$|
> |SimCLR         |$84.03 \pm 1.08$|$83.49 \pm 1.14$|$86.30 \pm 1.03$|
> |SimSiam        |$84.02 \pm 1.10$|$83.90 \pm 1.13$|$85.68 \pm 1.07$|
> |FACILE-SupCon  |$87.49 \pm 0.99$|$86.57 \pm 1.07$|$88.01 \pm 0.99$|
> |FACILE-FSP     |$\mathbf{88.74 \pm 0.94}$|$\mathbf{ 88.45 \pm 0.96 }$|$\mathbf{ 88.36 \pm 0.95 }$|
>
> Pretraining on input sets from CUB200. Testing with 5-shot 20-way meta-test sets; average F1 and CI are reported.
>
> Second, similar to first experiment, we finetune a fully-connected layer that is appended after DINO V2 [2] ViT-B/14. This methodology is applied across various models to assess their performance on histopathology image datasets. By adopting the DINO V2 architecture, known for its robustness and effectiveness in visual representation learning, we aim to harness its potential for the specialized domain of histopathology. We refer interested readers to the revised appendix for details of pretraining.
>
> 1-shot 5-way test on LC dataset
> |pretraining method | NC | LR | RC | LR+LA | RC+LA |
> |-------------------|----|----|----|-------|-------|
> |DINO V2 (ViT-B/14) |$44.82 \pm 1.41$|$47.51 \pm 1.39$|$47.63\pm 1.38$|$47.36 \pm 1.39$|$48.88\pm 1.44$|
> |SimSiam            |$48.79 \pm 1.37$|$49.43 \pm 1.35$|$48.43 \pm 1.36$|$49.38 \pm 1.34$|$49.50 \pm 1.34$|
> |SimCLR             |$50.47 \pm 1.31$|$50.52 \pm 1.33$|$50.44 \pm 1.32$|$51.66 \pm 1.32$|$51.78 \pm 1.38$|
> |FSP-Patch          |$49.73 \pm 1.41$|$53.59 \pm 1.38$|$53.07 \pm 1.41$|$51.79 \pm 1.40$|$51.27 \pm 1.43$|
> |FACILE-SupCon      |$\mathbf{56.24 \pm 1.43}$|$\mathbf{56.51 \pm 1.41}$|$\mathbf{55.95 \pm 1.42}$|$\mathbf{56.29 \pm 1.43}$|$54.07 \pm 1.44$ |
> |FACILE-FSP         |$55.67 \pm 1.40$|$56.26 \pm 1.36$|$55.83 \pm 1.35$|$56.01 \pm 1.38$|$\mathbf{55.35 \pm 1.40}$|

---

> > ### Author Response · Authors · 2023-11-17
> > **Response to Reviewer 4JEd (part 2)**
> >
> > 5-shot 5-way test on LC dataset
> > |pretraining method | NC | LR | RC | LR+LA | RC+LA |
> > |-------------------|----|----|----|-------|-------|
> > |DINO V2 (ViT-B/14) |$66.12 \pm 0.98$|$64.71 \pm 1.12$|$66.36 \pm 1.10$|$72.95 \pm 0.93$|$75.11 \pm 0.91$|
> > |SimSiam            |$67.51 \pm 0.96$|$64.99 \pm 1.05$|$65.39 \pm 1.05$|$70.30 \pm 0.93$|$71.19 \pm 0.93$|
> > |SimCLR             |$70.10 \pm 0.92$|$69.28 \pm 0.96$|$69.18 \pm 0.97$|$72.99 \pm 0.92$|$72.91 \pm 0.94$|
> > |FSP-Patch          |$71.97 \pm 0.96$|$71.11 \pm 1.04$|$71.19 \pm 1.03$|$73.96 \pm 0.94$|$73.20 \pm 0.96$|
> > |FACILE-SupCon      |$75.58 \pm 0.88$|$74.26 \pm 0.94$|$73.20 \pm 0.95$|$75.81 \pm 0.90$|$74.34 \pm 0.96$|
> > |FACILE-FSP         |$\mathbf{75.86 \pm 0.86}$|$\mathbf{74.64 \pm 0.89}$|$\mathbf{74.12 \pm 0.93}$|$\mathbf{76.17 \pm 0.88}$|$\mathbf{75.08 \pm 0.95}$  |
> >
> >
> > 1-shot 3-way test on PAIP dataset
> > |pretraining method | NC | LR | RC | LR+LA | RC+LA |
> > |-------------------|----|----|----|-------|-------|
> > |DINO V2 (ViT-B/14) |$41.51 \pm 1.27$|$44.37 \pm 1.26$|$44.28 \pm 1.25$|$42.43 \pm 1.27$|$42.78 \pm 1.27$|
> > |SimSiam            |$49.42 \pm 1.28$|$48.07 \pm 1.35$|$48.44 \pm 1.36$|$48.76 \pm 1.33$|$46.48 \pm 1.37$|
> > |SimCLR             |$48.60 \pm 1.19$|$48.76 \pm 1.25$|$47.98 \pm 1.26$|$48.94 \pm 1.23$|$47.20 \pm 1.26$|
> > |FSP-Patch          |$46.09 \pm 1.17$|$47.44 \pm 1.18$|$48.09 \pm 1.19$|$46.76 \pm 1.18$|$43.68 \pm 1.22$|
> > |FACILE-SupCon      |$\mathbf{51.97 \pm 1.18}$|$\mathbf{52.25 \pm 1.22}$|$\mathbf{51.80 \pm 1.22}$|$51.36 \pm 1.22$|$\mathbf{50.24 \pm 1.23}$|
> > |FACILE-FSP         |$51.34 \pm 1.16$|$51.18 \pm 1.19$|$51.51 \pm 1.19$|$\mathbf{51.50 \pm 1.16}$|$49.77 \pm 1.22$|
> >
> >
> > 5-shot 3-way test on PAIP dataset
> > |pretraining method | NC | LR | RC | LR+LA | RC+LA |
> > |-------------------|----|----|----|-------|-------|
> > |DINO V2 (ViT-B/14) |$57.59 \pm 1.07$|$58.19 \pm 1.10$|$59.37 \pm 1.07$|$61.84 \pm 0.85$|$60.81 \pm 0.86$|
> > |SimSiam            |$61.56 \pm 0.97$|$62.52 \pm 1.01$|$62.81 \pm 1.01$|$64.40 \pm 0.86$|$62.44 \pm 0.93$|
> > |SimCLR             |$62.20 \pm 0.93$|$61.78 \pm 0.99$|$63.20 \pm 0.97$|$63.38 \pm 0.86$|$63.03 \pm 0.88$|
> > |FSP-Patch          |$63.77 \pm 0.88$|$63.85 \pm 0.94$|$63.85 \pm 0.93$|$63.61 \pm 0.85$|$60.91 \pm 0.87$|
> > |FACILE-SupCon      |$\mathbf{67.16 \pm 0.84}$|$67.29 \pm 0.89$|$66.88 \pm 0.90$| $\mathbf{67.61 \pm 0.85}$|$\mathbf{66.34 \pm 0.84}$|
> > |FACILE-FSP         |$67.14 \pm 0.85$|$\mathbf{67.67 \pm 0.84}$|$\mathbf{67.54 \pm 0.86}$|$67.12 \pm 0.81$|$66.05 \pm 0.83$|
> >
> >
> > 1-shot 9-way test on NCT dataset
> > |pretraining method | NC | LR | RC | LR+LA | RC+LA |
> > |-------------------|----|----|----|-------|-------|
> > |DINO V2 (ViT-B/14) |$56.03 \pm 1.62$|$59.11 \pm 1.57$|$60.13 \pm 1.55$|$58.71 \pm 1.57$|$59.06 \pm 1.55$|
> > |SimSiam            |$62.60 \pm 1.45$|$61.89 \pm 1.50$|$61.90 \pm 1.51$|$ 62.27 \pm 1.47$|$61.05 \pm 1.44$|
> > |SimCLR             |$65.43 \pm 1.43$|$64.18 \pm 1.44$|$64.15 \pm 1.46$|$64.83 \pm 1.43$|$62.69 \pm 1.38$|
> > |FSP-Patch          |$65.22 \pm 1.49$|$65.93 \pm 1.41$|$65.94 \pm 1.40$|$65.26 \pm 1.45$|$62.66 \pm 1.46$|
> > |FACILE-SupCon      |$71.55 \pm 1.36$|$70.36 \pm 1.37$|$70.52 \pm 1.35$|$71.05 \pm 1.35$|$\mathbf{68.85 \pm 1.40}$|
> > |FACILE-FSP         |$\mathbf{72.05 \pm 1.34}$|$\mathbf{70.70 \pm 1.35}$|$\mathbf{70.77 \pm 1.34}$|$\mathbf{71.14 \pm 1.34}$|$68.03 \pm 1.40$|
> >
> > 5-shot 9-way test on NCT dataset
> > |pretraining method | NC | LR | RC | LR+LA | RC+LA |
> > |-------------------|----|----|----|-------|-------|
> > |DINO V2 (ViT-B/14) |$76.85 \pm 0.98$|$76.51 \pm 1.02$|$78.67 \pm 0.94$|$82.20 \pm 0.82$|$82.75 \pm 0.83$|
> > |SimSiam            |$80.81 \pm 0.85$|$80.06 \pm 0.87$|$81.55 \pm 0.85$|$83.18 \pm 0.80$|$82.39 \pm 0.83$|
> > |SimCLR             |$82.87 \pm 0.80$|$81.91 \pm 0.82$|$82.86 \pm 0.80$|$83.92 \pm 0.77$|$82.89 \pm 0.79$|
> > |FSP-Patch          |$83.63 \pm 0.83$|$83.49 \pm 0.80$|$84.34 \pm 0.78$|$85.32 \pm 0.75$|$83.03 \pm 0.79$|
> > |FACILE-SupCon      |$87.74 \pm 0.64$|$87.00 \pm 0.64$|$87.38 \pm 0.62$|$87.82 \pm 0.63$|$86.15 \pm 0.69$|
> > |FACILE-FSP         |$\mathbf{87.93 \pm 0.65}$|$\mathbf{87.52 \pm 0.65}$|$\mathbf{87.72 \pm 0.62}$|$\mathbf{88.01 \pm 0.64}$|$\mathbf{86.46 \pm 0.70}$|

---

> > > ### Author Response · Authors · 2023-11-17
> > > **Response to Reviewer 4JEd (part 3)**
> > >
> > > Notably, our methods, FACILE-SupCon and FACILE-FSP, demonstrated remarkably superior results in comparison to other baseline models when applied to histopathology image datasets. This outcome highlights the effectiveness of these methods in leveraging coarse-grained labels specific to histopathology, thereby greatly enhancing the model performance of downstream tasks. Another critical insight emerged from our research: the current foundation model, DINO V2, exhibits limitations in its generalization performance on histopathology images. This suggests that while DINO V2 provides a strong starting point due to its robust visual representation capabilities, there is a clear need for further fine-tuning or prompt learning to optimize its performance for the unique challenges presented by histopathology datasets. This finding underscores the importance of specialized adaptation in the application of foundation models to specific domains like medical imaging.
> > >
> > > > *Question (1):* To what extent does the similarity between the pretraining dataset and its coarse labels and the target dataset with its fine labels affect the effectiveness of the method? For instance, can the method perform well when the pretraining dataset is CIFAR-100 while the downstream task involves a medical dataset? In such a scenario, which pretraining method is preferable: supervised pretraining on ImageNet, self-supervised pretraining (ignoring labels entirely), or the proposed method?
> > >
> > > **A:** Our theorem suggested that the relatedness, quantified by the relative Lipschitzness relative to $\mathcal{E}$, is an improtant factor for excess risk bound. It is intuitive that tasks in different domains do not share the same embedding space (and thus a large Lipschitz constant). Therefore, we deliberately choose pretraining tasks in source domain that is closely related the target downstream task.
> > >
> > > In this context, although foundation models provide a powerful tool for general ML applications, sigificant work, involving model fine-tuning or prompt learning, is still needed to adopt it to domain-specific task as shown in our new experiment results with CLIP image encoder and DINO V2 model and some public works, e.g., [4]. We propose an alternative finetuning framework towards this effort.
> > >
> > > We acknowledge that estimating the extent to which the source and target datasets need to be similar for optimal transfer is a valuable avenue for future research. However, our current work focuses on demonstrating that even when there are label space and label granularity differences---as in pretraining/fine-tuning on TCGA and applying the model to NCT---our method offers a practical advantage by making effective use of coarse-grained, yet related, labels to improve the learning process for fine-grained classification tasks. The effectiveness of our approach is therefore predicated on the relatedness of the set-level annotations to the target task, which is a realistic and common scenario in many machine learning applications.

---

> > > > ### Author Response · Authors · 2023-11-17
> > > > **Response to Reviewer 4JEd (part 4)**
> > > >
> > > > > *Question (2):* Given the same 'related' dataset, if you have both the fine-grained label and coarse-grained label, which pretraining strategy is preferable? (Let say your downstream task is classification on a medical image dataset, with fine-grained label A,B,C. The pretraining dataset you have is another medical image dataset (thus more related than ImageNet). You have both coarse label D,E,F and fine-grained label G,H,I,J,K,L. In this case, is fully supervised pretraining on G,H,I,J,K,L more beneficial on set level coarse pretraining on D,E,F more beneficial?)
> > > >
> > > > **A:** We thank the reviewer for the insightful query regarding the optimal pretraining strategy using fine-grained or coarse-grained labels in related datasets.
> > > >
> > > > * Our empirical evaluation in Section 3.4 contrasts models pretrained on the NCT dataset (fine-grained labels) against those using the TCGA dataset (coarse-grained labels). This comparison clearly demonstrates the effectiveness of models leveraging coarse-grained labels, as evidenced by our results and those in [1].
> > > > * Theoretically, we quantify dataset relatedness via the Lipschitzness condition (Definition 3). Intuitively, source tasks with lower Lipschitz constants yield better excess risk bounds.
> > > > * As highlighted in the Introduction section, obtaining coarse-grained labels is often less resource-intensive, particularly in fields like medical image analysis where label scarcity is a challenge. This underscores the practical value of utilizing more readily available coarse-grained labels. We also note that in some applications coarse-grained labels can be extracted from metadata or otherwise inferred and so the cost of adding coarse-grained labels for some applications is minimal.
> > > > * We recognize the importance of future research in identifying the most beneficial source data in advance for optimal transfer, even within the same domain. The choice between fine-grained and coarse-grained labels for pretraining involves complex considerations, including dataset specifics, label acquisition costs, and the goals of downstream tasks.
> > > >
> > > >
> > > > References:
> > > >
> > > > [1] Yang, Jiawei, Hanbo Chen, Jiangpeng Yan, Xiaoyu Chen, and Jianhua Yao. "Towards better understanding and better generalization of few-shot classification in histology images with contrastive learning." arXiv preprint arXiv:2202.09059 (2022).
> > > >
> > > > [2] Oquab, Maxime, Timothée Darcet, Théo Moutakanni, Huy Vo, Marc Szafraniec, Vasil Khalidov, Pierre Fernandez et al. "Dinov2: Learning robust visual features without supervision." arXiv preprint arXiv:2304.07193 (2023).
> > > >
> > > > [3] Radford, Alec, Jong Wook Kim, Chris Hallacy, Aditya Ramesh, Gabriel Goh, Sandhini Agarwal, Girish Sastry et al. "Learning transferable visual models from natural language supervision." In International conference on machine learning, pp. 8748-8763. PMLR, 2021.
> > > >
> > > > [4] Zhang, Jingwei, Saarthak Kapse, Ke Ma, Prateek Prasanna, Joel Saltz, Maria Vakalopoulou, and Dimitris Samaras. "Prompt-MIL: Boosting Multi-Instance Learning Schemes via Task-specific Prompt Tuning." arXiv preprint arXiv:2303.12214 (2023).
> > > >
> > > > [5] Zaheer, Manzil, Satwik Kottur, Siamak Ravanbakhsh, Barnabas Poczos, Russ R. Salakhutdinov, and Alexander J. Smola. "Deep sets." Advances in neural information processing systems 30 (2017).
> > > >
> > > > [6] Lee, Juho, Yoonho Lee, Jungtaek Kim, Adam Kosiorek, Seungjin Choi, and Yee Whye Teh. "Set transformer: A framework for attention-based permutation-invariant neural networks." In International conference on machine learning, pp. 3744-3753. PMLR, 2019.

---

### Comment · Reviewer_4JEd · 2023-11-19

i have read the authors' response. I thank the authors for providing additional experiments as well as addressing some of my questions. I maintain my score.

---

> ### Author Response · Authors · 2023-11-22
> **Reply to Reviewer 4JEd**
>
> Thank you for reviewing our additional experiments and for your insightful feedback. Your engagement with our work and the detailed attention to our responses are greatly appreciated. We respect your decision. Should you have any more questions or require further clarification on any aspect of our research, please do not hesitate to reach out. We are fully committed to addressing any additional concerns you may have and value your input highly.

---

### Comment · Reviewer_jPxq · 2023-11-22

Thank you to the authors for their detailed responses and for providing additional experimental results. While I appreciate these efforts, I believe that further clarification on the use of coarse grained labels, particularly in how they influence downstream performance, would significantly enhance the understanding of this method. The current rationale, especially regarding the choice of a coarse grained label like TCGA, seems to lack sufficient detail to fully grasp its potential impact. It reminds me of generalizing all images in the ImageNet dataset under a single, broad category. I am open to and would welcome a more compelling explanation, as it would provide a clearer insight into your approach.

---

> ### Author Response · Authors · 2023-11-22
> **Further response to reviewer jPxq**
>
> > Thank you to the authors for their detailed responses and for providing additional experimental results. While I appreciate these efforts, I believe that further clarification on the use of coarse-grained labels, particularly in how they influence downstream performance, would significantly enhance the understanding of this method. The current rationale, especially regarding the choice of a coarse-grained label like TCGA, seems to lack sufficient detail to fully grasp its potential impact. It reminds me of generalizing all images in the ImageNet dataset under a single, broad category. I am open to and would welcome a more compelling explanation, as it would provide a clearer insight into your approach.
>
> **A:** Thank you for your feedback and for highlighting areas in our manuscript where further clarification is needed. We appreciate your interest in the nuances of our approach, especially concerning the use of coarse-grained labels.
>
> The idea of generalizing all images in the ImageNet dataset under a single, broad category has been explored in prior lines of research [1][2]. Building upon this, our work extends this framework to a more general setting. In our setting, coarse-grained labels are associated with a set of instances, rather than being limited to individual instances (explained in section 2). This expansion allows for a more nuanced understanding and application of coarse-grained labels in complex learning environments.
>
> The rationale for selecting coarse-grained labels in the TCGA is detailed in the third paragraph of the introduction and in Section 3.4. As previously discussed, both in this section and in our related work, this approach differs from "the generalization of all images in the ImageNet dataset into a single, broad category". First, unlike the methods used in [1] and in Experiment 4.2 of [2], we do not rely on a class taxonomy to differentiate between coarse-grained and fine-grained labels. Second, the inputs provided to the models for predicting these labels vary: we use set input for coarse-grained labels and instance input for fine-grained labels.
>
> In response to your concerns, we would like to direct your attention to several key sections of our paper that address these issues:
>
> * Rigorous Definitions and Theoretical Analysis (Section 2): We provide detailed definitions of coarse-grained set-level labels, fine-grained labels, and related key conditions. The distinction between coarse-grained set-level labels and fine-grained labels is crucial for understanding the subsequent theoretical analysis. Our theoretical framework explores how learning with coarse-grained set-level labels can enhance downstream task performance. This is not just a speculative statement but is backed by rigorous analysis. The results presented in Figures 4 and 5 are not just empirical findings; they are in alignment with our theoretical predictions, demonstrating the practical effectiveness of our approach.
> * Intuitive Illustrations (Introduction, third Paragraph): To aid in understanding, we provide two illustrative examples that demonstrate why coarse-grained labels can be beneficial. These examples are designed to be intuitive and accessible, providing a clear rationale for our approach.
> * Experiment-Specific Explanations (Empirical Study Section and Appendix Section B): In each experiment, we do not merely apply coarse-grained labels. Instead, we provide specific reasons why certain coarse-grained labels are chosen and how they are expected to contribute to the learning of the particular downstream tasks.
>
> We believe that these sections and figures collectively offer a comprehensive explanation of our methodology and its advantages. Our aim is to demonstrate not only the theoretical soundness of using coarse-grained labels but also their practical efficacy in enhancing downstream task performance. We trust that this clarification has addressed your concerns and offered a more transparent understanding of our approach. We are eager to answer any further questions you may have. Thank you for your time and effort.
>
> References:
>
> [1] Phoo, Cheng Perng, and Bharath Hariharan. "Coarsely-labeled data for better few-shot transfer." In Proceedings of the IEEE/CVF international conference on computer vision, pp. 9052-9061. 2021.
>
> [2] Robinson, Joshua, Stefanie Jegelka, and Suvrit Sra. "Strength from weakness: Fast learning using weak supervision." In International Conference on Machine Learning, pp. 8127-8136. PMLR, 2020.

---

### Meta-Review · Area_Chair_tVbH · 2023-12-12

**Metareview:**

This paper proposes to leverage abundant coarse-grained labels and hierarchical relationships to enhance fine-grained representation learning.  The coarse-grained labels are used as in a coarse-set prediction task to learn the instance-level features,  and the fine-grained classification can then be fine-tuned on the instance-level features.   The paper also provides theoretical analysis on how coarse-grained labels can speed up fine-grained classification.   Performance gains are observed on several benchmarks.

The strengths of the paper are the novel idea (using set prediction on coarse labels for instance-level pre-training) and its theoretical analysis.  The weaknesses of the paper are clarity of implementation details and insufficient experimental validation.

The paper has received 4 reviews, to which authors have provided further clarification and extensive additional experimental results that directly address reviewers' concerns, resulting in final ratings of ratings 5/6/6.   Given these additional updates and the novelty of set-level learning perspective, the AC agrees with the majority consensus and recommends acceptance.

Please also discuss the similarities and differences with the following paper:
[1] Improving Generalization via Scalable Neighborhood Component Analysis, ECCV 2018
where the image feature is learned with coarse-grained ImageNet labels and evaluated directly on fine-grained labels.  The idea can be extended to subsequent few-shot learning setting.

Experimental comparisons based on such experimental settings would help tease apart the set-label learning perspective from the instance-label coarse-grained learning perspective, bringing more clarity to points raised by Reviewer jPxq and Reviewer xN2u.

Please also consider improving the clarity of the writing.

**Justification For Why Not Higher Score:**

It needs more revision and work to improve the clarity and make the experimental validation more convincing.

**Justification For Why Not Lower Score:**

Novel learning perspective.
Current performance gains are consistent and promising.

---

### Decision · Program_Chairs · 2024-01-16

Accept (poster)